# Peat Oxic and Anoxic Controls of *Sphagnum* Decomposition Rates in the Holocene Peatland Model Decomposition Module Estimated from Litterbag Data

Henning Teickner[1, 2], Edzer Pebesma[2], and Klaus-Holger Knorr[1]

[1]Ecohydrology & Biogeochemistry Group, Institute of Landscape Ecology, University of Münster, 48149, Germany
[2]Spatiotemporal Modelling Lab, Institute for Geoinformatics, University of Münster, 48149, Germany

**Correspondence:** Henning Teickner (henning.teickner@uni-muenster.de)

**Abstract.** The Holocene Peatland Model (HPM) is a widely applied model to understand and predict long-term peat accumulation, but it is difficult to test due to its complexity, measurement errors, and lack of data. Instead of testing the complete model, tests of individual modules may avoid some of these problems. In particular, the HPM decomposition module can be tested with litterbag data, but no such test has been conducted yet.

Here, we estimate parameter values of the HPM decomposition module from available *Sphagnum* litterbag experiments included in the Peatland Decomposition Database and with a litterbag decomposition model that considers initial leaching losses. Using either these estimates or the standard parameter values, we test whether the HPM decomposition module fits decomposition rates ($k_0$) in *Sphagnum* litterbag experiments along a gradient from oxic to anoxic conditions.

Both litterbag data and model versions where HPM decomposition module parameters were estimated suggest a less steep gradient of decomposition rates from oxic to anoxic conditions and larger anaerobic decomposition rates for several species than the standard parameter values. This discrepancy may be caused by ignoring effects of water table fluctuations on aerobic and anaerobic decomposition rates. Moreover, our analysis suggests that maximum possible decomposition rates of individual species ($k_{0,i}$) vary more than suggested by the standard parameter values of the HPM plant functional types. Based on previous sensitivity analyses of the HPM, the estimated differences to the standard parameter values can cause differences in predicted 5000 year C accumulation up to $100 \, \text{kg m}^{-2}$.

The HPM decomposition module with standard parameter values fits $k_0$ estimated from *Sphagnum* litterbag data, but model versions where HPM decomposition module parameters were estimated and differ significantly have an equivalent fit. The reason why models with different parameter values have equivalent fit is that errors in remaining masses and the design of available litterbag experiments support a range of initial leaching loss and $k_0$ estimates. Consequently, applications of the HPM and any other peatland model should consider that a broad range of decomposition module parameter values is compatible with available litterbag experiments.

Improved litterbag experiments are needed for more accurate tests of any peatland decomposition module and for obtaining parameter estimates accurate enough to allow even only approximate predictions of long-term peat accumulation. The modeling approach used here can be combined with different data sources (for example measured degree of saturation) and decomposition modules. In light of the large differences in long-term peat accumulation suggested by the parameter estimates, we conclude

that it is worth to conduct such experiments, not only to improve the decomposition module of the HPM, but to improve peatland models in general.

## 1 Introduction

Decomposition is one of the major controls of how much carbon (C) peatlands can store. Compared to other ecosystems, northern peatlands usually have small decomposition rates because of cold temperatures, high water table levels, acidic pH value, and litter that does not decompose fast because of chemical and physical litter properties (van Breemen, 1995; Rydin et al., 2013). These slow decomposition rates caused northern peatlands to accumulate at least 400 Gt C (Yu, 2012; Nichols and Peteet, 2019) during the Holocene but changes in the controls of decomposition rates may cause them to loose considerable amounts of C to the atmosphere under climate and land use changes (Frolking et al., 2011; Loisel et al., 2017).

Peatland models are used to better understand past C accumulation and to predict future changes in peat C stocks, but because of the long time scales which have to be considered, they are difficult to test. Past studies have compared site-adapted simulations of peat height, age, C and N stocks, macrofossil composition, and water table level predicted by peatland models against peat core data (e.g., Frolking et al., 2010; Tuittila et al., 2013; Treat et al., 2021; Zhao et al., 2022), and have shown that existing peatland models can reproduce observed patterns to some extent. However, these tests suffer from two problems. First, they cannot reliably identify the parameter values or model equations that cause discrepancies between model predictions and measurements because they test entire peatland models against observed data. Second, there often are large uncertainties both in the model being tested and the data used to test the model; peatland models have large uncertainties in parameter values and model structure and these may produce a range of predictions as illustrated by uncertainty analyses (e.g., Quillet et al., 2013a, Quillet et al. (2013b)) and model intercomparisons (e.g., Zhao et al., 2022). Observed data also has uncertainty from measurements, peat dating, or simply missing data, for example for past precipitation. Large uncertainties can make tests inconclusive, no matter how much data we use. As a consequence, there remains large and often not quantified uncertainty about parameter values that control decomposition rates.

An alternative that avoids some of these problems is to test only some part of a model while taking into account relevant uncertainty sources. Estimating values and errors of parameters that directly control decomposition rates could be used to test the decomposition module of a peatland model. For example, in the Holocene Peatland Model (HPM) (Frolking et al., 2010), we only need to know litter species, peat degree of saturation, the depth of the litter below the peat surface, water table depth, and only five parameters to predict decomposition rates. Decomposition rates can also be estimated from litterbag experiments, where a known initial mass of litter is filled into mesh bags, incubated in peat, excavated after some time, and re-weighed to estimate the mass loss due to decomposition. Therefore, predicted decomposition rates can be compared to decomposition rates estimated from litterbag experiments and replicability of any identified discrepancies can be directly tested in future litterbag studies. Admittedly, such a test is restricted to the time ranges covered by available litterbag experiments and therefore not representative for long-term decomposition rates which may differ from that of fresh litter (e.g., Frolking et al., 2001), but

future tests with different scope and applications of the model will benefit from the reduced parameter uncertainties and can consider where the model fails already on short time scales.

A test of decomposition modules is relevant because of the importance of decomposition for long-term C accumulation in peatlands. Previous sensitivity analyses of the HPM and applications to peat cores suggest that relative small changes to the anoxia scale length ($c_2$), the parameter controlling how anaerobic decomposition rates are limited by electron acceptor depletion and accumulation of decomposition products, can result in a doubling of accumulated C, depending on climate conditions (Frolking et al., 2010; Quillet et al., 2013b; Kurnianto et al., 2015). These sensitivity analyses used assumed parameter ranges

that are not informed by litterbag experiments. A test of only the HPM decomposition module can provide better estimates for $c_2$ and may therefore help to reduce uncertainties in predicted C accumulation rates.

Currently, litterbag experiments are not as extensively used for testing peatland models as they could and only a fraction of the information available from litterbag experiments is used to develop models. The HPM derives initial decomposition rates of moss plant functional types from litterbag data, but parameters for environmental controls of decomposition are assumptions

which appear to be informed at most qualitatively by litterbag experiments, and it is not tested whether the HPM decomposition module successfully fits available litterbag data (Frolking et al., 2010). This is also the case for other dynamic peatland models, e.g. Frolking et al. (2001), Bauer (2004), Heijmans et al. (2008), Heinemeyer et al. (2010), Morris et al. (2012), Chaudhary et al. (2018), Bona et al. (2020).

One reason why such tests have been difficult is that suitable litterbag raw data to test peatland models are scarce. Bona et al.

(2018) developed a Peatland Productivity and Decomposition Parameter Database, but it contains only data from studies older than 2010 and no error estimates for remaining masses in litterbag data. Since decomposition rates have been estimated with different litterbag decomposition models in previous studies, their values are not directly comparable. Moreover, initial leaching losses (losses of soluble compounds, which do not originate from microbial depolymerization, due to leaching during the first days to weeks of incubation) can bias decomposition rate estimates if they are not explicitly considered and can vary between

species and experiments (Yu et al., 2001; Teickner et al., 2025a). Therefore, raw data (remaining masses) are necessary for any meaningful test of decomposition modules with litterbag data. The recently published Peatland Decomposition Database (Teickner and Knorr, 2024) contains raw data from available *Sphagnum* litterbag experiments and therefore allows to estimate parameters with any mass loss-based decomposition model and therefore also allows to consider initial leaching losses.

Even though tests of only a part of a model are less uncertain than tests of whole models, there still is a risk that they

are dominated by uncertainties. Remaining masses in litterbag experiments are often very variable, even under controlled environmental conditions (e.g., Bengtsson et al., 2018), and for many litterbag experiments, a range of decomposition rates may produce similar predictions for remaining masses (e.g., Yu et al., 2001), also if a litterbag decomposition model compatible with the HPM, i.e. that uses equation (7) in Frolking et al. (2010) to describe decomposition mass losses, is used (Teickner et al., 2025a). Finally, also only five model parameters, as in the case of the HPM decomposition module, can make predictions

uncertain. These uncertainties have to be taken into account to check whether litterbag data are compatible with the peatland model. A possible way to do this is to combine the HPM decomposition module, a litterbag decomposition model compatible

with this module, and available litterbag experiments into one model and use Bayesian data analysis (Gelman et al., 2014) to estimate uncertainties of data and parameters.

If such a test suggests that decomposition rates predicted by the HPM decomposition module do not fit estimates from litterbag experiments, or only if parameter estimates of the decomposition module differ from the parameter values originally suggested, even if main uncertainty sources are considered, the test has identified a discrepancy worth considering in more detail. We can then analyze whether previous sensitivity analyses of the HPM suggest that these discrepancies may have large effects on the predicted C accumulation, and if this is the case, the discrepancies are worth testing in future litterbag experiments.

Our aim is to test the HPM decomposition module against decomposition rates estimated from available *Sphagnum* litterbag experiments. Specifically, we want to:

1. Test whether the HPM decomposition module can predict litterbag decomposition rates for different *Sphagnum* species along the gradient from oxic to anoxic conditions.

2. Estimate HPM decomposition module parameters from litterbag data and compare them to the originally suggested values (standard parameter values) (Frolking et al., 2010) that are often used when applying the HPM (Tab. 1).

3. If some of the parameter values differ, identify the possible causes why parameter estimates from litterbag data differ to provide guidance for future litterbag experiments.

4. Analyze whether estimated differences in HPM parameter values could imply significant differences in decomposition rates and long-term peat accumulation.

To address these aims, we used the HPM decomposition module to predict decomposition rates in available litterbag experiments and compared these to decomposition rates estimated for the same litterbag experiments with a litterbag decomposition model that considers initial leaching losses (Teickner et al., 2025a) (Fig. 1). These predictions require the peat degree of saturation, which we estimate with the modified Granberg model (Granberg et al., 1999; Kettridge and Baird, 2007) from water table depth data reported in the litterbag studies. Furthermore, some *Sphagnum* litterbag experiments do not report water table depths and therefore cannot be used to test the HPM, but they still provide information on initial leaching losses and decomposition rates and therefore help to constrain parameter estimates. We therefore include these data via Bayesian hierarchical modeling in the litterbag decomposition model. In summary, our approach combines the HPM decomposition module, the modified Granberg model, and a *Sphagnum* litterbag decomposition model that allows to consider initial leaching losses and to pool information across litterbag experiments (Teickner et al., 2025a). While this approach has its limitations, it exploits available data as far as possible, while considering known confounders and propagating relevant uncertainties.

We only test the decomposition module of the HPM, but the decomposition modules of many other peatland models are also parameterized based on litterbag experiments and our modeling approach is flexible enough to be combined with other decomposition modules. Therefore, our test could serve as a blueprint for similar tests of other peatland model decomposition modules. Similarly, the parameter discrepancies identified here suggest future litterbag experiments that would provide novel

insights into oxic and anoxic controls of *Sphagnum* decomposition rates and our study therefore suggests a strategy to improve decomposition modules in general.

**Table 1.** Standard values of parameters of the decomposition module in the Holocene Peatland Model (Frolking et al., 2010).

| HPM parameter | Standard value | Description |
|---|---|---|
| $W_{opt}$ ($\text{L}_{\text{water}}\,\text{L}_{\text{pores}}^{-1}$) | 0.450 | Optimum degree of saturation for aerobic decomposition. |
| $c_1$ (-) | 2.310 | Curvature of the relation of the aerobic decomposition rate to the degree of saturation (larger values imply a steeper decrease of decomposition rates for degrees of saturation diverging from $W_{opt}$). |
| $f_{min}$ ($\text{yr}^{-1}$) | 0.001 | Minimum anaerobic decomposition rate. |
| $c_2$ (m) | 0.300 | Anoxia scale length. Represents limitation of anaerobic decomposition rates with increasing distance below the annual average water table depth due to end product accumulation and limitation of available electron acceptors. Larger values mean that anaerobic decomposition rates decrease less strongly with depth below the average annual water table level. |
| $k_{0,\text{hollow}}$ ($\text{yr}^{-1}$) | 0.130 | Maximum possible decomposition rate for hollow *Sphagnum* species. |
| $k_{0,\text{lawn}}$ ($\text{yr}^{-1}$) | 0.080 | Maximum possible decomposition rate for lawn *Sphagnum* species. |
| $k_{0,\text{hummock}}$ ($\text{yr}^{-1}$) | 0.060 | Maximum possible decomposition rate for hummock *Sphagnum* species. |

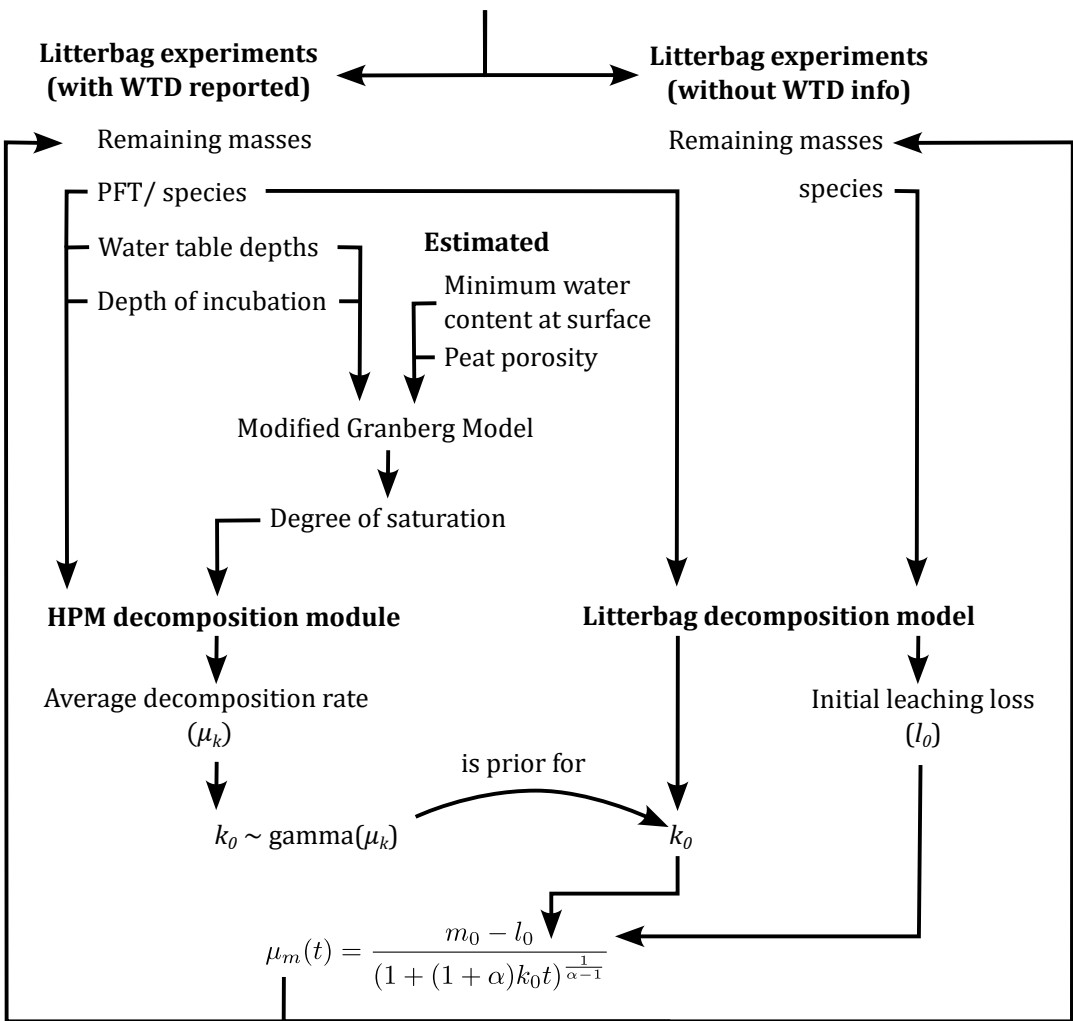

**Figure 1.** Conceptual representation of the modeling approach. Arrows represent flows of information. Litterbag data that have information on water table depths (WTD) and incubation depths are used to estimate average decomposition rates ($\mu_k$) with the HPM decomposition module. The HPM decomposition module needs plant functional type identity, peat degree of saturation, WTD, and incubation depth to predict decomposition rates. The modified Granberg model is used to estimate peat degree of saturation at incubation depths from WTD, minimum water content at the surface, and porosity, of which the latter two are estimated from the remaining masses. The litterbag decomposition model is used to estimate decomposition rates ($k_0$) for all litterbag studies, including those that have information on WTD and those that have not. A gamma distribution with $\mu_k$ as average is used as prior distribution for $k_0$ for the litterbag experiments that have information on WTD (curved arrow). This helps to constrain initial leaching loss and decomposition rate estimates for studies that can be predicted with the HPM decomposition module. The Litterbag decomposition module also estimates initial leaching losses ($l_0$) for all litterbag experiments. The equation at the bottom uses these to estimate remaining masses as reported in the litterbag experiments. The litterbag decomposition model is described in more detail in section 2.2. See the text for further details.

## 2 Methods

### 2.1 *Sphagnum* litterbag data

To test the HPM decomposition module against litterbag data, we used the Peatland Decomposition Database (Teickner and Knorr, 2024). In this study, we use data from Bartsch and Moore (1985), Vitt (1990), Johnson and Damman (1991), Szumigalski and Bayley (1996), Prevost et al. (1997), Scheffer et al. (2001), Thormann et al. (2001), Asada and Warner (2005), Trinder et al. (2008), Breeuwer et al. (2008), Straková et al. (2010), Hagemann and Moroni (2015), Golovatskaya and Nikonova (2017), and Mäkilä et al. (2018) to estimate decomposition rates ($k_0$) using the litterbag decomposition model (Fig. 1). Data from Johnson and Damman (1991), Szumigalski and Bayley (1996), Prevost et al. (1997), Straková et al. (2010), Golovatskaya and Nikonova (2017), and Mäkilä et al. (2018) include water table depths (WTD) and depths below the surface where litterbags were incubated, in addition to remaining masses and taxonomic information, and therefore only these datasets were used to predict $k_0$ also with the HPM decomposition module (Fig. 1). Samples originally classified as *Sphagnum magellanicum* are here classified as *Sphagnum magellanicum aggr.* (Hassel et al., 2018).

### 2.2 Modeling remaining masses and decomposition rates with the litterbag decomposition model

To estimate decomposition rates for available *Sphagnum* litterbag experiments we use the equation from the HPM that computes remaining masses from decomposition rates and decomposition time (Frolking et al., 2001, 2010) as litterbag decomposition model (Fig. 1), with three modifications. The original equation (equation (4) Frolking et al. (2001)) is:

$$m(t) = \frac{m_0}{(1 + (1 + \alpha)k_0 t)^{\frac{1}{\alpha - 1}}}, \tag{1}$$

where $m(t)$ is the fraction of initial mass remaining at time $t$, $m_0$ is the fraction of initial mass remaining at time $t = 0$, $k_0$ is the decomposition rate, and $\alpha$ is a parameter that describes how decomposition slows down as mass is lost, where the HPM assumes $\alpha = 2$ for simplicity (Frolking et al., 2001, 2010).

The modified version we use here is (Fig. 1):

$$\mu_m(t) = \begin{cases} m_0 & \text{if } t = 0 \\ \frac{m_0 - l_0}{(1 + (\alpha - 1)k_0 t)^{\frac{1}{\alpha - 1}}} & \text{if } t > 0 \end{cases}, \tag{2}$$

where $l_0$ is the fraction of mass lost due to initial leaching. The HPM decomposition process does not assume that there are initial leaching losses, but these are commonly observed in litterbag experiments and bias decomposition rate estimates when they are ignored (Yu et al., 2001; Teickner et al., 2025a); therefore, the modification is necessary to allow a sensible test of the HPM decomposition module with litterbag data.

The second modification is that we do not assume $\alpha = 2$, but consider it as an unknown parameter that is estimated from

**Table 2.** Overview on litterbag experiments included for each *Sphagnum* taxon in this study. "HPM microhabitat" is the HPM microhabitat assigned to each taxon. Taxa without value are not considered in Johnson et al. (2015) (see section 2.3.2). "Number of experiments" is the number of litterbag experiments available from the Peatland Decomposition Database (these are either individual replicates or average values of replicates, depending on what data were reported in the studies). "Number of experiments with WTD data" is the number of litterbag experiments that also report water table depths and for which we therefore could make predictions with the HPM decomposition module. "Depth range" are the maximum and minimum depth below the peat surface at which litterbags were placed [cm]. Missing values mean that no study reported depths.

| Taxon | HPM microhabitat | Number of studies | Number of experiments | Number of experiments with WTD data | Depth range |
|---|---|---|---|---|---|
| *Sphagnum* spec. | | 2 | 16 | 10 | 10, 30 |
| *S. angustifolium* | Hummock | 4 | 14 | 8 | 1, 30 |
| *S. auriculatum* | | 1 | 3 | 0 | 0, 6 |
| *S. balticum* | Lawn | 3 | 12 | 3 | 1, 30 |
| *S. cuspidatum* | Hollow | 1 | 5 | 5 | 10, 50 |
| *S. fallax* | Lawn | 1 | 4 | 1 | 1, 1 |
| *S. fuscum* | Hummock | 9 | 32 | 13 | 1, 50 |
| *S. lindbergii* | Lawn | 1 | 2 | 0 | |
| *S. magellanicum aggr.* | Hummock | 3 | 7 | 5 | 1, 50 |
| *S. majus* | Hollow | 1 | 2 | 2 | 10, 30 |
| *S. papillosum* | Lawn | 2 | 6 | 1 | 0, 1 |
| *S. rubellum* | Hummock | 1 | 2 | 2 | 10, 30 |
| *S. russowii* | Hummock | 1 | 3 | 2 | 1, 1 |
| *S. russowii* and *capillifolium* | | 1 | 18 | 0 | 5, 5 |
| *S. squarrosum* | Lawn | 1 | 2 | 0 | 0, 0 |
| *S. teres* | Lawn | 1 | 1 | 1 | 2, 2 |

litterbag data. Since $\alpha = 2$ was chosen for simplicity and attempts to reliably estimate $\alpha$ have failed (e.g., Clymo et al., 1998; Frolking et al., 2001; Teickner et al., 2025a), we estimate $\alpha$ mainly to consider the possible error introduced by this parameter. The third modification is that we change $m(t)$ to $\mu_m(t)$ because we assume that equation (2) describes only the average fraction of the initial mass remaining. For each retrieved litterbag, we assume that the remaining mass can be described with a beta distribution with precision parameter $\phi_m$:

$$m(t) \sim \text{beta}(\mu_m(t)\phi_m, (1 - \mu_m(t))\phi_m), \tag{3}$$

Values for $k_0$ are estimated from remaining masses reported in available litterbag experiments conditional on equation (2) and a hierarchical prior structure (Teickner et al., 2025a):

$$k_0 = \exp(\beta_{k,1} + \beta_{k,2,\text{species}} + \beta_{k,3,\text{species x study}} + \beta_{k,4,\text{sample}}), \tag{4}$$

where $\beta_{k,1}$ is the estimated decomposition rate across all litterbag experiments, $\beta_{k,2,\text{species}}$ describes the difference of the average decomposition rate for the *Sphagnum* species, $\beta_{k,3,\text{species x study}}$ for the study (nested within species), and $\beta_{k,4,\text{sample}}$ for the sample (litterbag experiment). All these parameters have normal distributions as priors. Hierarchical models of the same structure are used to estimate $l_0$ and $\alpha$ from equation (2) and to estimate $\phi_m$ from equation (3).

## 2.3 Prediction of litterbag decomposition rates with the Holocene Peatland Model decomposition module

To predict decomposition rates, the HPM decomposition module needs as inputs the litter type in terms of the HPM plant functional types (PFT), the fraction of mass already lost due to previous decomposition, the depth of the litter below the peat surface, the water table depth, and the peat degree of saturation (Frolking et al., 2010).

Predicting decomposition rates for the available litterbag data is not straightforward because the HPM decomposition module does not consider specific features of available litterbag experiments. The HPM does not specify how to assign species to plant functional types. Moreover, none of the available litterbag studies reported the degree of saturation which therefore needs to be estimated in order to make predictions with the HPM decomposition module. The only variables that can be directly linked are the depth of the litter below the peat surface, and water table depths (both reported in litterbag experiments). All other variables can be estimated from available litterbag experiments only with additional assumptions that are described in the following subsections.

In the following subsection, we give a more detailed description of our modeling approach, in particular of the HPM decomposition module and how it predicts decomposition rates, and how we link the decomposition rates estimated with the litterbag decomposition model to those predicted by the HPM decomposition module. The remaining subsections discuss how we derived or estimated PFT, WTD, and degree of saturation for the litterbag data and additional steps to make the litterbag data compatible with the HPM decomposition module.

### 2.3.1 The Holocene Peatland Model decomposition module

In our study, the decomposition rates estimated from litterbag experiments are compared against decomposition rates predicted by the HPM decomposition module (Frolking et al., 2010) for the same litterbag experiments (Fig. 1). The HPM decomposition module describes how decomposition rates depend on the *Sphagnum* PFT, the degree of saturation and the depth of a litter sample below the water table. Similar to the remaining mass, we here assume that the HPM decomposition module predicts an average decomposition rate, $\mu_k$, instead of the decomposition rate of individual samples (Fig. 1):

$$\mu_k = \begin{cases} k_{0,i} f_1(W) & \text{if } \hat{z} \leq 0 \\ k_{0,i} f_2(\hat{z}) & \text{if } \hat{z} > 0 \end{cases} \tag{5}$$

where $k_{0,i}$ is the PFT-specific maximum possible decomposition rate (Tab. 1), $W$ is the degree of saturation ($L_{\text{water}} \, L_{\text{sample}}^{-1}$), $\hat{z}$ the depth of the sample below the average annual water table ($\hat{z} = z - z_{\text{wt}}$, where $z_{\text{wt}}$ and $z$ are the depth of the water table and

litterbag below the peat surface), and $f_1$ and $f_2$ are modifiers due to $W$ (under oxic conditions) and $\hat{z}$ (under anoxic conditions), respectively. These modifiers are described in equations (8) and (9) in Frolking et al. (2010):

$$f_1(W) = 1 - c_1(W - W_{opt})^2 \tag{6}$$

$$f_2(\hat{z}) = f_{min} + (f_1(1) - f_{min}) \exp\left(\frac{-\hat{z}}{c_2}\right), \tag{7}$$

where all not yet mentioned parameters are defined in Tab. 1.

In our model that combines the HPM decomposition module and the litterbag decomposition model, $k_0$ estimated from the litterbag data for each litterbag experiment with reported WTD (sample) (equation (2)) is assumed to follow a gamma distribution with shape parameter $\alpha_{\mu_k}$ (estimated) and average $\mu_k$ (predicted for each sample with equation (5)):

$$k_0 \sim \text{gamma}\left(\alpha_{\mu_k}, \frac{\alpha_{\mu_k}}{\mu_k}\right), \tag{8}$$

Thus, the decomposition rate predicted by the HPM decomposition module (equation (8)) is a prior for $k_0$ as estimated from the litterbag decomposition model (equation (4)). This forms the link between the litterbag decomposition model and the HPM decomposition module (Fig. 1) and also allows us to estimate parameters of the HPM decomposition module from the litterbag data. The advantage of this modeling approach is that we can consider litterbag experiments without water table depth

to estimate $l_0$ and $k_0$ for individual *Sphagnum* species, which is additional information to constrain estimates of the HPM decomposition module parameters. Moreover, combining the litterbag decomposition model and the HPM decomposition module into one Bayesian model does not only estimate HPM decomposition module parameters from the litterbag data, but it also constrains the decomposition rates estimated from litterbag data by the HPM decomposition module because the HPM decomposition module serves as prior in the combined model which therefore estimates what parameter values are compatible

with the data and the combined model. This is exactly what we want because there is uncertainty both in the remaining masses reported in litterbag experiments and in HPM decomposition module parameters. If HPM decomposition module parameter estimates from the combined model are different from the standard values used in the original model (Tab. 1), even if we consider these uncertainties and use the HPM decomposition module as prior for the litterbag data, this is a discrepancy worth testing in future experiments.

**2.3.2 Assignment of *Sphagnum* species to plant functional types**

The HPM defines maximum possible decomposition rates ($k_{0,i}$) for three *Sphagnum* PFT (hollow, lawn, and hummock species), but not how to assign species to them. We assigned individual *Sphagnum* species to the three PFT by comparing their niche WTD with the optimal WTD for net primary production defined in the HPM. Specifically, we defined fixed average annual WTD intervals for the PFT: hollow ($<5$ cm), lawn ($\geq 5$ cm and $< 15$ cm), hummock ($\geq 15$ cm) based on the HPM (Frolking

et al., 2010). Then, we used niche WTD and standard deviations from Johnson et al. (2015) to assign *Sphagnum* species to these three microhabitats. Using only average values and the microhabitat WTD thresholds resulted in unintuitive assignments, such as assigning *S. fallax* to hummocks. To avoid such obvious misclassifications, we defined rules to assign species to HPM microhabitats based on the probability a species would occur in the three niche WTD intervals. To compute the probabilities, we assumed a normal distribution (Johnson et al., 2015):

1. Species with a probability of occurrence $\geq 15\%$ in the intervals of all three PFT were classified as lawn species.

   2. In all other cases, species were assigned to the PFT for which their probability of occurrence was largest.

Litterbag data from Prevost et al. (1997) are incubations of peat samples where the species is unknown. Based on descriptions in this study, it is likely that the peat was formed by hummock species. Hummock species are assumed to have the smallest decomposition rate among the three *Sphagnum* PFT in the HPM (Frolking et al., 2010) and this is in line with small decomposition
rate estimates for these samples (Teickner et al., 2025a). For these reasons, we assigned these samples to the hummock PFT of the HPM.

When estimating parameters of the HPM decomposition module from the litterbag data (see section 2.4.1), we also estimated the maximum possible decomposition rate ($k_{0,i}$). *Sphagnum* species differ in their decomposition rate and the PFT of the HPM are a simplification that may cause misfits of the HPM decomposition module to litterbag data. We therefore estimated $k_{0,i}$
for individual *Sphagnum* species in models HPM-all, HPM-leaching, and HPM-outlier (see section 2.4.1) and evaluated the variability of these species-specific estimates compared to the standard $k_{0,i}$ values of the HPM *Sphagnum* PFT.

### 2.3.3 Degree of saturation

We estimated the degree of saturation with the modified Granberg model (ModGberg model) (Granberg et al., 1999; Kettridge and Baird, 2007) from minimum water content at the surface ($\theta_{0,\text{min}}$), total porosity ($P$), the water table depth below the peat
surface ($z_{\text{wt}}$), and the depth of the litterbags below the peat surface during the incubation ($z$):

$$
\begin{aligned}
\theta(z) &= \min\left(P, \theta_0 + (P - \theta_0)\left(\frac{z}{z_{\text{wt}}}\right)^2\right) \\
\theta_0 &= \max\left(\theta_{0,\text{min}}, 0.15 z_{\text{wt}}^{-0.28}\right),
\end{aligned}
\tag{9}
$$

where $\theta_0$ is the water content at the surface and $0.15 z_{\text{wt}}^{-0.28}$ is an empirical relation of $\theta_0$ with the WTD (Kettridge and Baird, 2007).

The minimum water content at the surface was not reported in any study and we therefore assumed a minimum water content
at the surface of 0.05 $L_{\text{water}}\ L_{\text{sample}}^{-1}$ with a standard deviation of 0.05 $L_{\text{water}}\ L_{\text{sample}}^{-1}$, based on measurements from Hayward and Clymo (1982). The total porosity was not reported in any study and therefore we assumed an average value of 80% with a standard deviation of 10%, roughly based on values reported for low-density *Sphagnum* peat (Liu and Lennartz, 2019). An improved test of the HPM decomposition module would require litterbag experiments with direct measurements of the degree of saturation at sufficient temporal resolution.

### 2.3.4 Accounting for mass loss before the start of the litterbag experiments

The HPM decomposition module assumes that decomposition rates decrease the more of the initial mass has already been decomposed (Frolking et al., 2001, 2010). Thus, if litter lost some mass due to decomposition before the start of the litterbag experiment, one has to know the magnitude of this mass loss to correctly predict decomposition rates with the HPM decomposition module.

Because of the continuous growth of *Sphagnum* at the top and die-off below, it is difficult to separate living material, assumed to not have lost mass, from dead material, which may have already lost some mass. Based on a visual assessment, the studies that used *Sphagnum* material from the surface, assume that the material did not loose mass prior to the litterbag experiments and we follow this assumption ($m(t = 0) = 1$ in equation (2)).

Samples from Prevost et al. (1997) are *Sphagnum* peat collected from two different depth levels from the same location and these samples probably had already experienced some decomposition, however it is difficult to estimate how much. Apart from knowing the exact mass loss prior to the litterbag experiment, an alternative approach to allow predicting decomposition rates with the HPM with previous mass loss is to define a dummy species for a sample, such that the maximum possible decomposition rate for the sample ($k_{0,i}$) is estimated separately. We therefore estimated $k_{0,i}$ separately for each peat layer in Prevost et al. (1997), implicitly assuming that these are two different PFT with different maximum possible decomposition rates.

### 2.4 Testing the HPM decomposition module against litterbag data

### 2.4.1 Model versions

To test different aspects of the HPM decomposition module and the additional assumptions we make, we computed several models which differ in whether HPM decomposition module parameters were fixed to their standard values or estimated from data, whether peat properties (porosity, water table depth, water content, minimum water content at the surface) are estimated from data or not, and whether the HPM decomposition module was extended to also predict $l_0$ or not (Tab. 3).

The first model (HPM-standard) does not estimate any parameters of the HPM decomposition module (equations (5) to (7)) and does not estimate peat properties from the litterbag data and therefore is the HPM decomposition module with standard parameter values, while propagating prior uncertainties for peat properties. For this model, predictions of $k_0$ equal $\mu_k$ (equation (5)). This version of the HPM decomposition module is completely independent of the litterbag decomposition model, meaning that the HPM decomposition module is not used as prior for the litterbag decomposition model (Fig. 1). This also means that to compare $k_0$ predicted by HPM-standard to $k_0$ estimated from the litterbag decomposition model, we need to estimate the litterbag decomposition model independently, without using the HPM decomposition module as prior. This independent litterbag decomposition model is called LDM-standard (Tab. 3). We use LDM-standard not only to compare $k_0$ estimates to $k_0$ predictions of HPM-standard, but also to analyze how $k_0$ estimates of the litterbag decomposition model changes when we use different versions of the HPM decomposition module as prior in the subsequent models.

Each subsequent model combines the HPM decomposition module and the litterbag decomposition model into one Bayesian model via equation (8). Each of these models estimates an additional set of parameters from the litterbag data relative to the previous model (Tab. 3). First, only the peat properties (HPM-peat) are estimated, and second all HPM parameters ($k_{0,i}$, $c_1$, $W_{opt}$, $f_{min}$, $c_2$) (HPM-all). Finally, HPM-leaching extends HPM-all by adding formulas to model how $l_0$ depends on the degree of saturation, similar to how the HPM decomposition module predicts $k_0$ with equation (6).

HPM-peat tested whether the HPM decomposition module can fit available litterbag data when the HPM decomposition module and the litterbag decomposition model are combined and when peat properties are estimated from data.

HPM-all estimates what HPM decomposition module parameter values are compatible with available litterbag data and therefore allows to test whether the standard parameter values are extreme relative to these estimates. Values of $k_{0,i}$ were estimated for each species separately, as described in section 2.3.2.

HPM-leaching was computed because decomposition rates estimated from available litterbag experiments are sensitive to initial leaching losses (Yu et al., 2001; Lind et al., 2022; Teickner et al., 2025a). It is therefore interesting to see whether litterbag decomposition rates are estimated differently in HPM-leaching, when initial leaching losses are constrained by adding formulas to model how $l_0$ depends on the degree of saturation, compared to HPM-all, when initial leaching loss estimates are constrained only by the litterbag decomposition model. Based on previous experiments with tea bags it is reasonable to assume that there is some relation between initial leaching losses and the degree of saturation (Lind et al., 2022). Specifically, we use the following logistic regression model to describe an average initial leaching loss per sample, in dependency of the degree of saturation:

$$
\begin{aligned}
\mu_l &= \quad \text{logit}^{-1}(\beta_{l,1} + \beta_{l,2}W) \\
l_0 &\sim \quad \text{beta}(\mu_l \phi_l, (1 - \mu_l)\phi_l),
\end{aligned}
\tag{10}
$$

where $\mu_l$ is the average initial leaching loss for a sample, $\beta_{l,1}$ is the (hypothetical) average initial leaching loss at a degree of saturation 0 for each taxon, $\beta_{l,2}$ is the coefficient that describes the relation to the degree of saturation ($W$), and $\phi_l$ transforms $\mu_l$ and $(1 - \mu_l)$ into the shape and rate parameters of a beta distribution. This beta distribution has the same function as the gamma distribution (equation (8)) for $k_0$ (compare also with Fig. 1): it is a prior for $l_0$ estimated with the litterbag decomposition model, where the average of this prior is $\mu_l$.

To check whether outliers in the litterbag data could influence our results, we computed one additional model, HPM-outlier, with the same structure as HPM-leaching, but estimated without litterbag experiments identified as outliers. Litterbag experiments were defined as outliers if the reported average remaining mass of any litterbag (batch) during the experiment had a posterior probability $> 99\%$ to be different from the remaining mass predicted by the litterbag decomposition model alone. This procedure identified experiments as outliers where remaining masses increased over time, where litterbags collected at intermediate time points had unexpectedly low remaining masses, or where initial leaching losses were retarded to later time points, presumably because of freezing after the start of the experiment (Teickner et al., 2025a). In total, 5 litterbag experiments were identified as outliers. Results for HPM-outlier are shown in supporting information S8 and HPM decomposition module parameter estimates agree with estimates of HPM-leaching and HPM-all.

Strictly, we do not test the decomposition module in the HPM, but the combination of the HPM decomposition module and the modified Granberg model, assuming that uncertainties in water table depths are negligible and that we accounted sufficiently for uncertainties in total porosity. This ambiguity has to be accepted when combining heterogeneous litterbag data where some variables have to be estimated. Litterbag experiments where water table depths and the degree of saturation are measured at sufficient temporal resolution are needed to avoid this ambiguity in future studies and to improve any test of the HPM decomposition module.

**Table 3.** Overview of HPM decomposition module modifications computed in this study.

| Model | Description |
|---|---|
| LDM-standard | The litterbag decomposition model without the HPM decomposition module as prior. This is model 1-4 from Teickner et al. (2025a). |
| HPM-standard | The Holocene Peatland Model decomposition module with standard parameter values (Frolking et al., 2010). The model is run with peat water contents estimated with the modified Granberg model, using water table depths and litterbag depths reported from the litterbag studies, and assuming a fixed peat porosity, and minimum peat water content at the surface. |
| HPM-peat | The same as HPM-standard, but combined with LDM-standard into one Bayesian model, where the HPM decomposition module is a prior for the litterbag decomposition model (Fig. 1). Water table depths, peat porosity, and minimum peat water content at the surface are estimated from data. |
| HPM-all | The same as HPM-peat, but now also parameters from the HPM decomposition module ($k_{0,i}$, $W_{opt}$, $f_{min}$, $c_1$, $c_2$) are estimated from the litterbag data. |
| HPM-leaching | The same as HPM-all, but now also an average initial leaching loss for each species and, across all species, a factor, by which this average leaching loss increases or decreases as the peat degree of saturation increases, are estimated (equation (10)). |
| HPM-outlier | The same as HPM-leaching, but computed without litterbag experiments that were identified as outliers (see the text for details). |

### 2.4.2 Bayesian data analysis

All models listed in Tab. 3 were computed with Bayesian statistics to account for relevant uncertainty sources and include relevant prior knowledge (for example that *Sphagnum* decomposition rates are unlikely to be larger than 0.5 yr$^{-1}$). Bayesian

computations were performed using Markov Chain Monte Carlo (MCMC) sampling with Stan (2.32.2) (Stan Development Team, 2021a) in R (4.2.0) (R Core Team, 2022) via the rstan package (2.32.5) (Stan Development Team, 2021b) using the NUTS sampler (Hoffman and Gelman, 2014), with four chains, 4000 total iterations per chain, and 2000 warmup iterations per chain. None of the models had divergent transitions, the minimum bulk effective sample size was larger than 400, and the largest rank-normalized $\hat{R}$ was 1.01, indicating that all chains converged (Vehtari et al., 2021). All models used the same priors for the same parameters and prior choices are listed and justified in supporting Tab. S1. Results of prior and posterior predictive checks are shown in supporting section S3.

We used power-scaling of the prior and likelihood distributions as implemented in the priorsense package (0.0.0.9000) (Kallioinen et al., 2024) to analyze the relative sensitivity of the posterior distribution to small perturbations of the prior and likelihood in HPM-leaching for HPM decomposition module parameters and peat properties. This is a computationally nonexpensive way to check whether the data provide information about a parameter and where prior and data may provide conflicting information (Kallioinen et al., 2024). Results of this analysis and further information on the data analysis are shown in supporting information S2.

### 2.4.3 Fit of model predictions to estimated decomposition rates and observed remaining masses

To analyze how well the models fit remaining masses observed in the litterbag experiments, we plotted reported remaining masses versus remaining masses estimated by the litterbag decomposition model in HPM-peat, HPM-all, and HPM-leaching. HPM-standard is not linked to the litterbag decomposition model and therefore does not predict remaining masses.

To analyze how well all HPM decomposition module versions fit $k_0$ estimated by the respective litterbag decomposition model, we created a similar plot for $k_0$. Here, we compared predictions of HPM-standard (equation (8)) against estimates of LDM-standard (equation (4)). We also computed the average difference of $k_0$ predicted by the HPM decomposition module and estimated from the litterbag data. We then computed the posterior probability that this average difference is different from zero. A large probability indicates a misfit of the model to available litterbag data. We also tested the same difference for specific species because graphical checks indicated that the decomposition rate prediction skill of the HPM decomposition module depends on species.

To test whether HPM-leaching has not only a better fit to available litterbag data, but also a better predictive accuracy for novel data than the model with standard parameter values (HPM-standard), we compared how well both can predict $k_0$ from litterbag experiments. HPM decomposition module parameters of HPM-standard are not estimated from data and therefore we could compute the root mean square error of prediction ($\text{RMSE}_{\text{test}}$) directly with $k_0$ predicted by HPM-standard and estimated with LDM-standard. HPM decomposition module parameters of HPM-leaching are estimated from the litterbag data and we therefore used cross-validation (CV) to estimate $\text{RMSE}_{\text{test}}$. Since decomposition rates from the same species and study usually are not independent, we defined blocks which were used as CV-folds. Each fold represents the data from one study, but only if there were still data for the same *Sphagnum* species left in the remaining data (we want to estimate the predictive accuracy not for new species). Species with data from one study only were always used for model training and not part of the testing folds. This procedure resulted in 5 folds. HPM-standard and HPM-leaching were tested against the same data. In the text, $\text{RMSE}_{\text{train}}$

is the RMSE computed with the data a model was estimated with (for HPM-standard, the data the litterbag decomposition model was estimated with), and $RMSE_{test}$ is the RMSE computed with independent test data.

### 2.4.4 Changes in $k_0$ and $l_0$ estimates of the litterbag decomposition models compared to LDM-standard

To analyze how parameter values of the litterbag decomposition model change when it is combined with different versions of the HPM decomposition module as prior, we estimated the average difference of $k_0$ and $l_0$ estimates of each model version to $k_0$ and $l_0$ estimates of LDM-standard. In particular, this allowed us to analyze whether there is any change in the relative magnitude of $l_0$ and $k_0$ because the litterbag decomposition module would constrain these parameter values to fit the respective HPM decomposition module prior and still fit the observed remaining masses.

### 2.4.5 Magnitudes of $k_0$ along the gradient from oxic to anoxic conditions

To analyze how $k_0$ changes along the gradient from oxic to anoxic conditions, we plotted $k_0$ estimated by LDM-standard versus the water table depth below the litterbags. To this plot, we added $k_0$ predicted by HPM-standard. To analyze how the relation of $k_0$ changes for the HPM decomposition module modifications compared to HPM-standard, we computed differences between $k_0$ estimated by HPM-peat, HPM-all, and HPM-leaching, respectively, and HPM-standard, and plotted these differences versus the water table depth below the litterbags.

### 2.4.6 Difference between values of $k_{0,i}$, $c_1$, $W_{opt}$, $c_2$, $f_{min}$ estimated from litterbag data to the standard parameter values

For HPM-all and HPM-leaching, we computed the posterior probability that the HPM decomposition module parameter values estimated from litterbag data ($k_{0,i}$, $c_1$, $W_{opt}$, $c_2$, $f_{min}$) differ from the standard parameter values (Tab. 1). This way, we could identify discrepancies between standard parameter values and parameter values estimated from available litterbag data.

For HPM-leaching, we conducted in addition a sensitivity analysis, where we simulated decomposition of *S. fuscum* incubated at different depths in a peatland with water table depth of 40 cm below the surface, a porosity of 0.7 $L_{pores}$ $L_{sample}^{-1}$, and a minimum water content at the surface of 0.05 $g_{water}$ $g_{sample}^{-1}$. With these settings, we predicted five sets of average $k_0$: (1) with HPM-leaching ($k_{0,modified}$(HPM-leaching)), and the remaining with (2) $c_1$, (3) $W_{opt}$, (4) $f_{min}$, and (5) $c_2$ set to their standard value ($k_{0,standard}$(HPM-leaching)). We then computed the difference of $k_0$ from set (1) and (2) to analyze the effect of the new $c_1$ estimate, from set (1) and (3) to analyze the effect of the new $W_{opt}$ estimate, and so on for sets (4) and (5). This gives the difference in decomposition rates of HPM-leaching if we would set individual HPM decomposition module parameters to their standard values. This way, we could analyze what HPM decomposition module parameters contribute to a change in $k_0$ predictions along the gradient from oxic to anoxic conditions.

## 3   Results

### 3.1   Fit and predictive accuracy of the different versions of the HPM decomposition module to available litterbag data

In each model, the litterbag decomposition model fitted the observed remaining masses similarly well (Fig. 2 (a) and supporting Fig. S2), no matter whether the HPM decomposition module was used as prior or not, and whether its parameters were estimated from data (HPM-all, HPM-leaching) or not (HPM-peat). Thus, remaining masses do not indicate large differences between the model versions.

For $k_0$, the picture is more nuanced: When the HPM decomposition module is not used as prior (HPM-standard), it fitted $k_0$ estimated by the litterbag decomposition model on average worse than when it was used as prior (all other model versions) (Fig. 2, Tab. 4). For example, HPM-standard had an average $\text{RMSE}_{\text{train}}$ of 0.11 $\text{yr}^{-1}$, whereas HPM-leaching had an average $\text{RMSE}_{\text{train}}$ of 0.02 $\text{yr}^{-1}$. However, the cross-validation indicates that when applied to novel samples, both HPM-standard and HPM-leaching would perform similarly well if one considers the large uncertainty of the $\text{RMSE}_{\text{test}}$ estimates (Tab. 4). Interestingly, all model versions where the HPM decomposition module was used as prior had comparable fits ($\text{RMSE}_{\text{train}}$) (Tab. 4), even the version that still has standard parameter values for the HPM decomposition module (HPM-peat). This indicates that a change in HPM decomposition module standard parameter values is not required to make the HPM decomposition module fit $k_0$ values estimated from available litterbag data via the litterbag decomposition model, under the assumptions we made. Instead, the results indicate that parameter values of the litterbag decomposition model can be adjusted to fit predictions of this HPM decomposition module prior.

**Table 4.** Training and testing RMSE for decomposition rates as predicted by different versions of the decomposition module of the Holocene Peatland Model (see Tab. 3 for a description of the models) and number of misfits. $\text{RMSE}_{\text{train}}(k_0)$ is the root mean square error of model predictions for litterbag replicates used during model computation. $\text{RMSE}_{\text{test}}(k_0)$ is the RMSE for litterbag replicates used in blocked cross-validation. Where no $\text{RMSE}_{\text{test}}(k_0)$ is given, it was not computed for these models. Values are averages and lower and upper bounds of central 95% posterior intervals ($\text{yr}^{-1}$). Misfits counts the number of litterbag experiments for which $k_0$ predicted by the HPM decomposition module modification differed from $k_0$ as estimated from the litterbag decomposition model with a posterior probability of at least 99%. In total, $k_0$ was predicted with the HPM decomposition module modifications for 53 litterbag experiments ($\text{RMSE}_{\text{train}}(k_0)$) or 29 ($\text{RMSE}_{\text{test}}(k_0)$).

| Model | $\text{RMSE}_{\text{train}}(k_0)$ | $\text{RMSE}_{\text{test}}(k_0)$ | Misfits |
|---|---|---|---|
| HPM-standard | 0.105 (0.051, 0.191) | 0.136 (0.06, 0.252) | 13 |
| HPM-peat | 0.02 (0.013, 0.029) | | 0 |
| HPM-all | 0.014 (0.008, 0.021) | | 0 |
| HPM-leaching | 0.022 (0.012, 0.039) | 0.088 (0.038, 0.179) | 0 |
| HPM-outlier | 0.021 (0.013, 0.032) | | 0 |

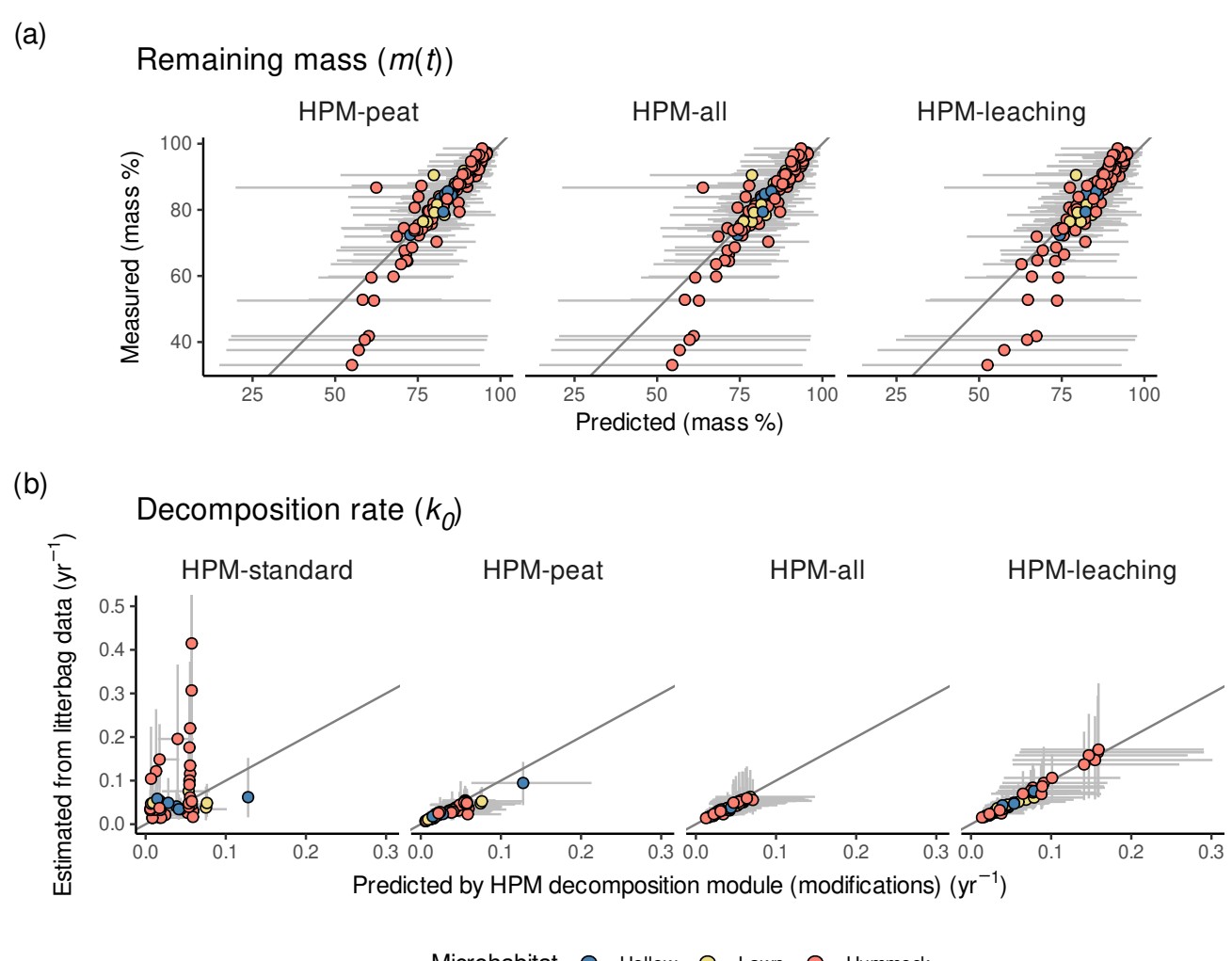

**Figure 2.** (a) Measured remaining masses versus remaining masses predicted by the litterbag decomposition model combined with each HPM decomposition module version. Values are shown for litterbag experiments with reported water table data. For HPM-standard no values are shown because it was not combined with a litterbag decomposition model. (b) $k_0$ estimated by the litterbag decomposition model versus $k_0$ predicted by different modifications of the HPM decomposition module (Tab. 3). For HPM-standard, y-axis values are $k_0$ estimates of LDM-standard. For all other model versions, y-axis values are $k_0$ estimates of the litterbag decomposition module with the respective HPM decomposition module version as prior. Points represent average estimates and error bars 95% posterior intervals. Points are colored according to the microhabitat classification of *Sphagnum species* (see the Methods section for details). In (b), error bars exceeding 0.5 yr$^{-1}$ are clipped.

## 3.2 How are parameter values of the litterbag decomposition model adjusted when different versions of the HPM decomposition module are used as prior?

To understand how using the HPM decomposition module as prior changes $k_0$ and $l_0$ estimates of the litterbag decomposition model, we compared $k_0$ and $l_0$ estimates of the litterbag decomposition model of each model version with the $k_0$ and $l_0$ estimates of LDM-standard. We computed the average difference of $k_0$ estimates by the litterbag decomposition model for all models compared to the $k_0$ estimates of LDM-standard (using only litterbag experiments with reported WTD). Average differences compared to LDM-standard are in the order HPM-peat < HPM-all < HPM-leaching (average and 95% confidence interval: -0.04 (-0.06, -0.02) < -0.03 (-0.06, -0.01) < -0.01 (-0.04, 0.01) $\mathrm{yr}^{-1}$). The magnitude (mean absolute difference) of adjustments of $k_0$ estimates is different for different species (species with at least 3 samples): The largest average absolute differences across all models were made for *S. angustifolium* (0.15 (0.06, 0.27) $\mathrm{yr}^{-1}$) and the smallest for *Sphagnum* spec. (0.01 (0.01, 0.02) $\mathrm{yr}^{-1}$). This indicates that for some species $k_0$ estimates of the litterbag decomposition model are forced to smaller values for HPM-peat and HPM-all, whereas differences are smaller for HPM-leaching.

With these changes in $k_0$ estimates, a similar fit to remaining masses as observed for all models (see the previous subsection) is only possible when $l_0$ estimates are changed in the opposite direction. To check this, we computed the average difference of $l_0$ estimates by the litterbag decomposition model for all model versions compared to the $l_0$ estimates of LDM-standard. Differences compared to LDM-standard are in the order HPM-leaching < HPM-all < HPM-peat (average and 95% confidence interval: 0.1 (-1.9, 2.2) < 2.8 (0.7, 4.8) < 3.3 (1.6, 5) mass %). Again, the magnitude (mean absolute difference) of adjustments of $l_0$ estimates is different for different species (species with at least 3 samples): The largest average absolute differences across all models were made for *S. angustifolium* (11.4 (7, 16.6) mass %) and the smallest for *Sphagnum* spec. (1.43 (0.86, 2.39) mass %). Thus, the smaller $k_0$ estimates are indeed compensated by larger $l_0$ estimates for HPM-peat and HPM-all, whereas the difference to LDM-standard is smaller for HPM-leaching.

Overall, this analysis indicates that errors in remaining masses observed in available litterbag experiments are large enough to support a range of $k_0$ and $l_0$ estimates. The equivalent fit of the different model versions is therefore caused by adjusting $k_0$ to the HPM prior, and adjusting $l_0$ as needed to fit observed masses.

## 3.3 How do HPM decomposition module parameter estimates differ to the standard values?

Two model versions estimated HPM decomposition module parameters ($c_1$, $W_{opt}$, $f_{min}$, $c_2$, $k_{0,i}$): HPM-all and HPM-leaching. These models indicate larger values for $c_2$ and $W_{opt}$ than the standard values. Figure 3 shows marginal posterior densities of the HPM decomposition module parameters for HPM-all, with standard parameter values as defined in Frolking et al. (2010) indicated by vertical lines. For both HPM-all and HPM-leaching, there are large posterior probabilities that $c_2$ ($P_{\mathrm{HPM\text{-}all}}(c_2 > 0.3 \mathrm{~m}) = 1$ and $P_{\mathrm{HPM\text{-}leaching}}(c_2 > 0.3 \mathrm{~m}) = 1$) and $W_{opt}$ ($P_{\mathrm{HPM\text{-}all}}(W_{opt} > 0.45 \mathrm{~L_{water}~L_{pores}^{-1}}) = 1$ and $P_{\mathrm{HPM\text{-}leaching}}(W_{opt} > 0.45 \mathrm{~L_{water}~L_{pores}^{-1}}) = 0.98$) have larger values than the standard parameter values, indicating a discrepancy between the HPM decomposition module and available litterbag data (Fig. 3 and supporting Fig. S11). In contrast, estimates for $f_{min}$ do not differ much to the prior value and the power-scaling sensitivity analysis indicates a weak influence of the data (supporting information

S2) and therefore that currently available litterbag data provide only little information about minimum decomposition rates under anoxic conditions. HPM-all and HPM-leaching suggest a large variability of $k_{0,i}$ for individual species: Both models estimate a large posterior probability ($> 95\%$) that *S. russowii* and *S. rubellum* have a larger, and that *S. cuspidatum* has a smaller maximum possible decomposition rate ($k_{0,i}$) than the standard values for the respective PFT (Fig. 3 (b) and supporting Fig. S11). However, estimates for $k_{0,i}$ were very variable for the same species when different subsets of the litterbag data

were used to estimate the model in the cross-validation. This indicates that samples of the same species from different studies have a large variability in $k_{0,i}$ values. In summary, when HPM decomposition module parameters are estimated from available litterbag data, estimates for $W_{opt}$ and $c_2$ are larger than the standard values, differences to the $c_1$ and $f_{min}$ standard value cannot be detected, and estimates for $k_{0,i}$ are variable and have large errors for different species.

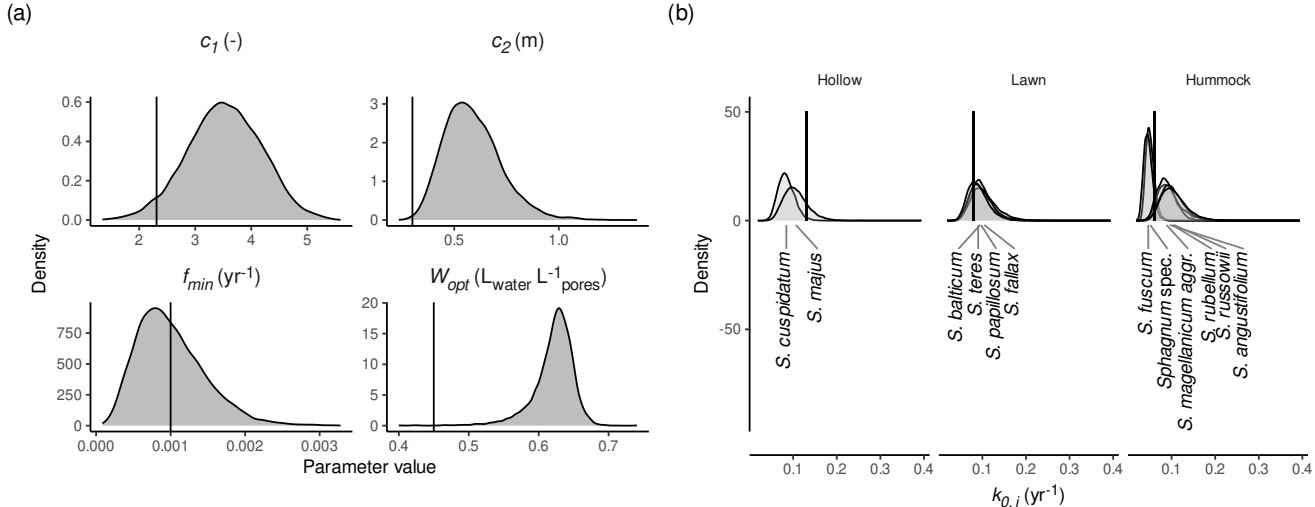

**Figure 3.** Marginal posterior distributions of HPM decomposition module parameters (see Tab. 1) as estimated by HPM-all. (a) Marginal posterior distributions for $c_1$, $W_{opt}$, $f_{min}$, and $c_2$. (b) Marginal posterior distributions for $k_{0,i}$ (maximum possible decomposition rate for species $i$). Species were assigned to HPM microhabitats as described in section 2.2.2. Vertical black lines are the standard parameter values from Frolking et al. (2010). *Sphagnum* spec. are samples that have been identified only to the genus level.

### 3.4 Magnitude and change of decomposition rates along the gradient from oxic to anoxic conditions

A comparison of $k_0$ estimates of LDM-standard and $k_0$ estimates of HPM-standard shows that the HPM decomposition module with standard parameter values implies a steeper decrease of decomposition rates from oxic to anoxic conditions than LDM-standard and, for some species, smaller anaerobic decomposition rates. Figure 4 (a) shows $k_0$ estimated by LDM-standard and $k_0$ predicted by HPM-standard versus water table depths below the litterbags reported in the studies for species with at least three litterbag experiments. Regression lines were fitted to both sets of $k_0$ values and they indicate an on average steeper slope for HPM-standard than for LDM-standard for many species (with large uncertainties). Moreover, under anoxic conditions

(negative water table depth), $k_0$ estimates by LDM-standard are larger on average for many of the litterbag experiments than what HPM-standard predicts (Fig. 4 (b)).

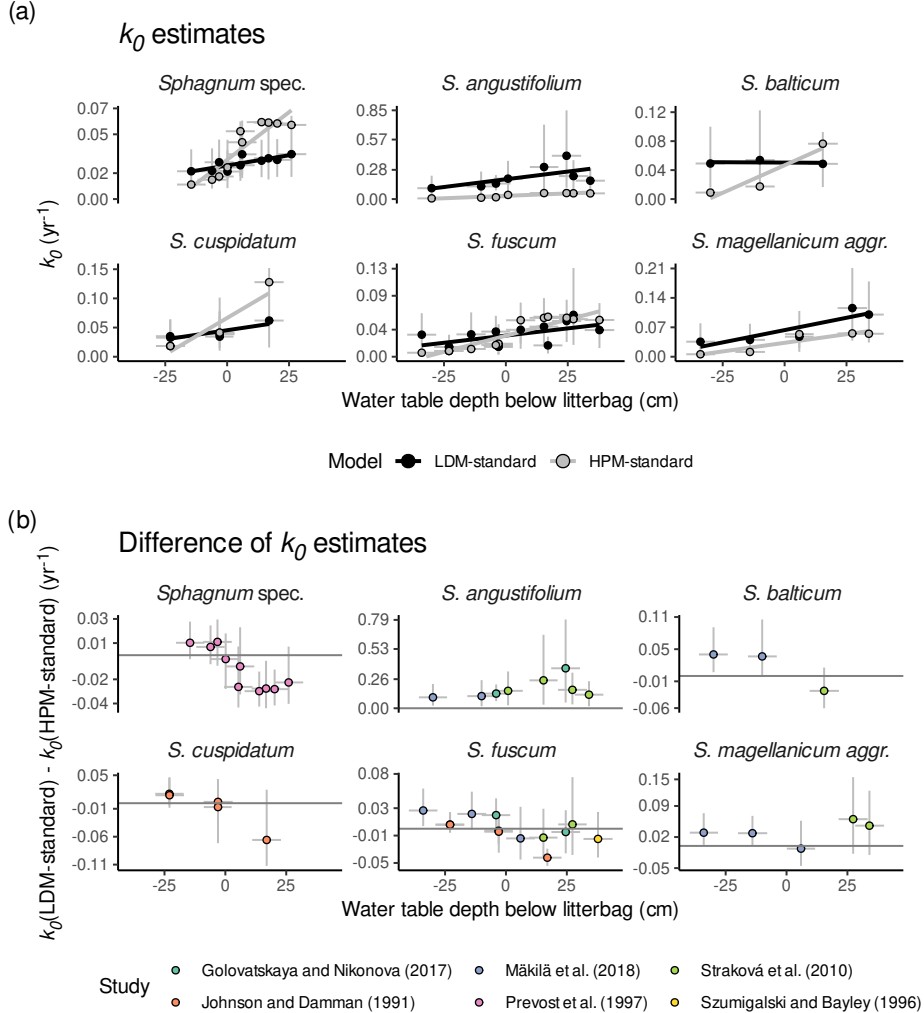

**Figure 4.** Comparison of $k_0$ estimates of HPM-standard and LDM-standard (Tab. 3) for species with at least three litterbag experiments. (a) $k_0$ estimates of HPM-standard (grey) and $k_0$ estimates of LDM-standard (black) versus reported average water table depths below the litterbags (negative values represent litterbags placed below the water table, positive values represent litterbags placed above the water table in the unsaturated zone). (b) $k_0$ estimates of LDM-standard minus $k_0$ estimates of HPM-standard versus reported average water table depths below the litterbags (i.e., the difference of the values shown in (a)). Grey horizontal lines indicate a difference in $k_0$ of 0 $yr^{-1}$. Points represent average estimates and error bars 95% posterior intervals. Lines are predictions of linear models fitted to the average estimates. *Sphagnum* spec. are samples that have been identified only to the genus level.

A comparison of $k_0$ estimates of HPM-standard and the other modifications of the HPM decomposition module suggests that when HPM decomposition module parameters are estimated, larger anaerobic decomposition rates and a less steep decrease of decomposition rates from oxic to anoxic conditions are predicted, similar to LDM-standard. We computed the difference of $k_0$ predicted by HPM-standard and the other HPM decomposition module versions (Fig. 5). When the HPM decomposition module with standard parameter values is used as prior for the litterbag decomposition module (HPM-peat), it predicts $k_0$ nearly identical to HPM-standard. In contrast, both model versions where HPM decomposition module parameters were estimated predict larger anaerobic decomposition rates and less of an increase under oxic conditions relative to anoxic conditions than HPM-standard. Thus, the HPM decomposition module with standard parameter values predicts a steeper decrease of decomposition rates from oxic to anoxic conditions and overall smaller anaerobic decomposition rates than LDM-standard and the models that estimate HPM decomposition module parameters from available litterbag data.

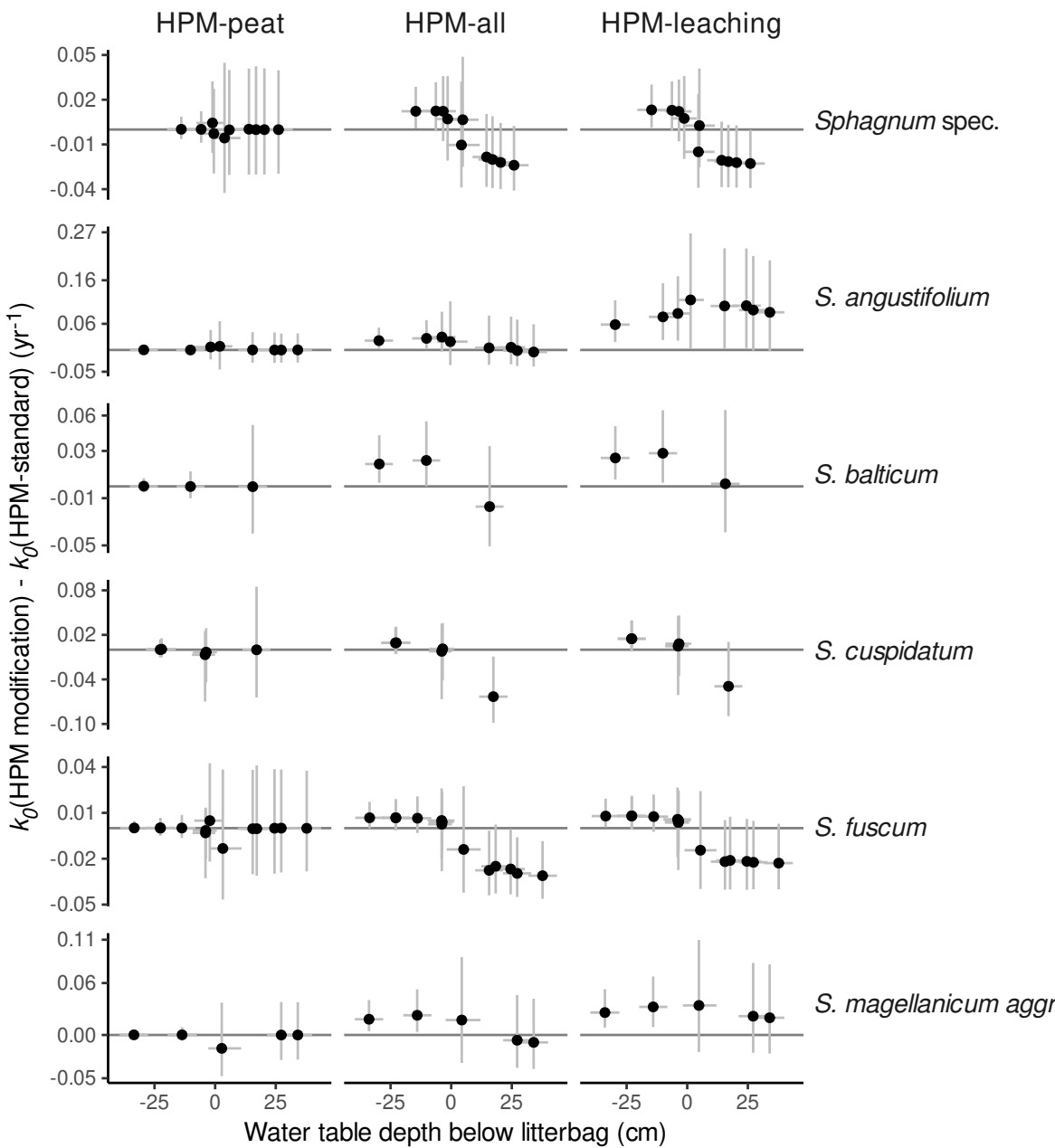

**Figure 5.** $k_0$ predicted by HPM decomposition module modifications (either HPM-peat, HPM-all, or HPM-leaching) minus $k_0$ predicted by the HPM decomposition module with standard parameter values (HPM-standard) versus estimated average water table depths below the litterbags (negative values represent litterbags placed below the water table, positive values represent litterbags placed above the water table in the unsaturated zone). Points represent average estimates and error bars 95% posterior intervals. *Sphagnum* spec. are samples which that been identified only to the genus level. Only data for species with at least three replicates are shown.

### 3.5 HPM decomposition module parameters that are responsible for the less steep gradient in decomposition rates from oxic to anoxic conditions

To analyze which of the HPM decomposition module parameters ($c_1$, $W_{opt}$, $f_{min}$, $c_2$) cause the less steep gradient in decomposition rates from oxic to anoxic conditions, we conducted a sensitivity analysis, where we made predictions with HPM-leaching for the same species and the same gradient from oxic to anoxic conditions, each time setting one of the four parameters to their standard values (four sets of predictions in total). We then computed the difference of the predicted $k_0$ values to predictions of HPM-leaching (with no parameter value set to its standard value). This difference is plotted versus the depth of the water table

below the litter, as shown in Fig. 6. This analysis suggests that $W_{opt}$ and $c_2$ cause the less steep gradient in decomposition rates from oxic to anoxic conditions, whereas the other two parameters have no qualitative influence.

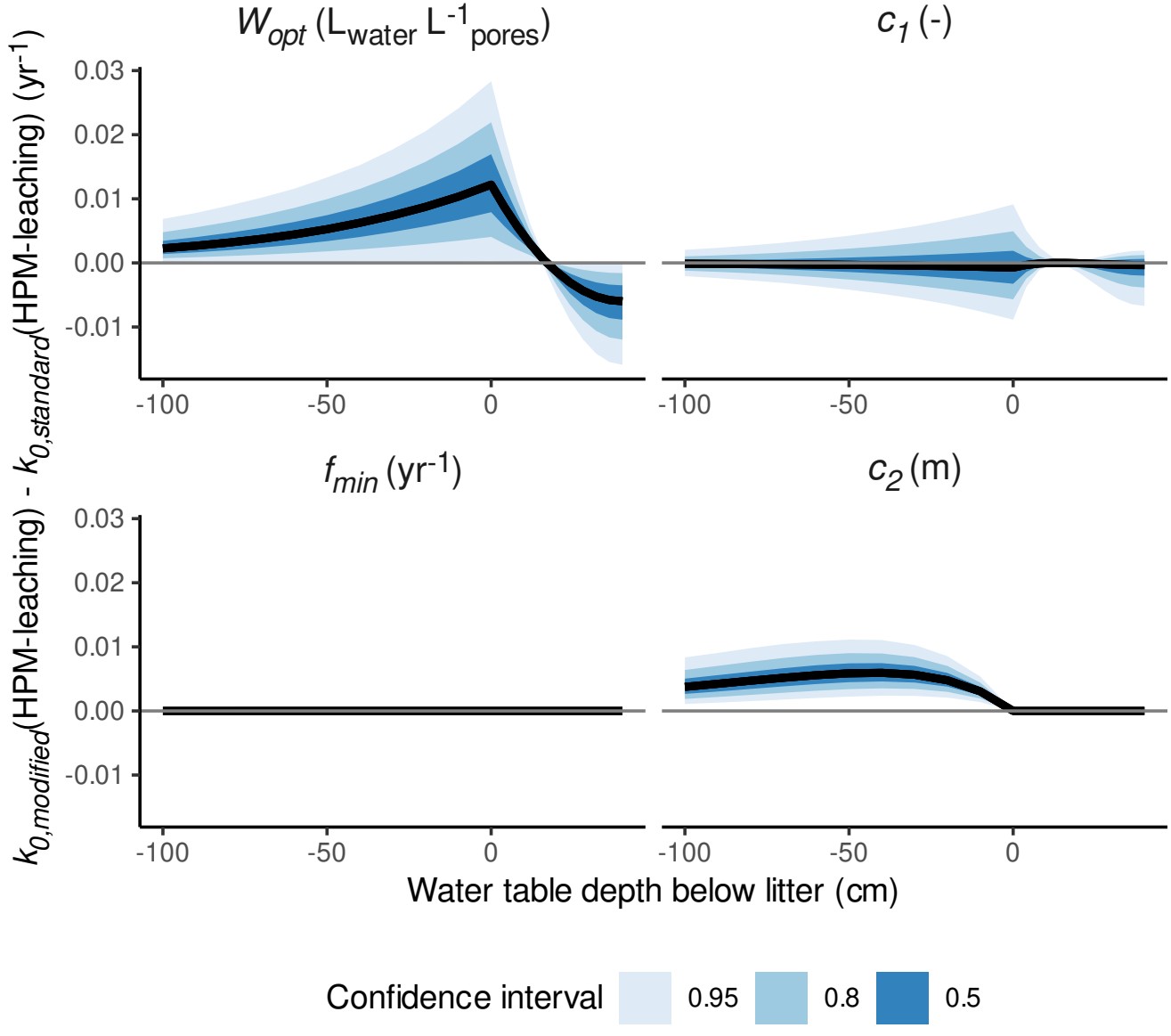

**Figure 6.** Difference between decomposition rates for *S. fuscum* predicted with parameter values estimated by HPM-leaching ($k_{0,\text{modified}}$(HPM-leaching)), and when setting the HPM decomposition module parameter in the panel title to its standard value ($k_{0,\text{standard}}$(HPM-leaching)), versus the water table depth below the litter (negative values represent litter placed below the water table, positive values represent litter placed above the water table in the unsaturated zone). Panels show results when different parameters are set to their standard values. Positive $k_{0,\text{modified}}$(HPM-leaching) $- k_{0,\text{standard}}$(HPM-leaching) means that decomposition rates are larger when using the estimated parameter value compared to using the standard parameter value. Shaded areas are central confidence intervals with probabilities given in the figure legend.

### 3.6 Relation of $l_0$ to the degree of saturation

In model HPM-leaching, we included a logistic regression model that estimates the relation between $l_0$ and the degree of saturation. The parameter estimates suggest that both positive and negative relations of $l_0$ to the degree of saturation are compatible with available litterbag data (95% confidence intervals for the slope (logit scale): (-0.28, 0.15)). Thus, available litterbag data do not allow to conclude whether $l_0$ are positively related to the degree of saturation or not.

### 4 Discussion

Our aims were to test whether the HPM decomposition module fits decomposition rates estimated from available litterbag experiments, to estimate HPM decomposition module parameters from available litterbag experiments, to understand what factors could cause differences in parameter estimates to the standard values, and to check whether the estimates from litterbag data could imply significant differences in peat accumulation predicted by the HPM compared to the standard parameter values.

The parameter estimates derived from available litterbag data suggest differences in the control of decomposition rates compared to the standard parameter values: the HPM decomposition module with standard parameter values predicts a steeper decrease of decomposition rates from oxic to anoxic conditions and smaller anaerobic decomposition rates for several species than estimated from LDM-standard and the models that estimate HPM decomposition module parameters from available litterbag data. These differences imply larger estimates for $W_{opt}$, the degree of saturation where decomposition rates are maximal, and $c_2$, the anoxia scale length (the parameter that controls how strong decomposition rates decrease below the water table depth). We will show here, by comparing parameter estimates to results from sensitivity analyses of the HPM, that the new parameter estimates can cause large differences in long-term peat accumulation predicted by the HPM.

Our analysis suggests that the HPM decomposition module with standard parameter values fits available litterbag data, but our modifications, where $W_{opt}$, $c_2$, and (for some species) $k_{0,i}$ estimates significantly differ from the standard values, have equivalent fit. This can be explained by two mechanisms: first, the litterbag decomposition model explains mass loss by initial leaching and decomposition. Thus, remaining masses reported in a litterbag experiment can be fitted either by assuming a larger $l_0$ and smaller $k_0$, or by assuming a smaller $l_0$ and larger $k_0$. By this first mechanism, the litterbag decomposition model can first estimate $k_0$ to agree with the HPM decomposition module and then adjust $l_0$ to fit the remaining masses of the litterbag experiments. The second mechanism is the impact of the design of available *Sphagnum* litterbag experiments on the accuracy of $l_0$ and $k_0$ estimates: initial leaching losses can explain mass losses only at the start of the experiment (equation (2)), but decomposition explains a continuous mass loss. It is therefore possible to estimate $l_0$ and $k_0$ accurately when remaining masses shortly after the start of the experiment are recorded, but the majority of litterbag experiments collects the first litterbags only after half a year or later (Teickner et al., 2025a). This causes large errors in $l_0$ and $k_0$ estimates and therefore allows the model to adjust $l_0$ and $k_0$ by the first mechanism, such that all model versions have equivalent fit to remaining masses while also fitting decomposition rates suggested by different HPM decomposition module priors. Improved litterbag experiments are needed for more accurate tests of any peatland decomposition module and for obtaining parameter estimates accurate enough to allow

even only approximate predictions of long-term peat accumulation. Applications of the HPM should consider this variability in parameter estimates compatible with available litterbag experiments.

In the next subsections, we first evaluate the reliability of our test. We discuss whether the identified parameter value differences could be an artifact of using heterogeneous litterbag data, and we discuss how compatible the new HPM decomposition module parameter estimates are with other studies that analyzed how decomposition rates differ in dependency of water availability or that estimated $c_2$ from peat core data. Second, we address the remaining aims: we discuss what factors could cause the larger anaerobic decomposition rates and, in some cases, smaller aerobic decomposition rates estimated by the litterbag decomposition model, and we discuss what implications the differences between estimated and standard parameter values have for peat accumulation predicted by the HPM. Finally, we give recommendations for improving tests of peat decomposition modules.

## 4.1 Reliability of the identified discrepancies

Before analyzing potential causes of the discrepancies found for $c_2$ and $W_{opt}$ we first ask if combining different litterbag experiments is reliable evidence for the less steep gradient in decomposition rates from oxic to anoxic conditions.

If we take a look at the misfits of the standard HPM decomposition module (HPM-standard) shown in Fig. 4, many, but not all underestimations of aerobic decomposition rates could have been caused by other factors: for example for *S. balticum* the difference may have been caused by differences in the two litterbag experiments from which we collected the data because the replicate with positive water table depth is from Straková et al. (2010), whereas the two others are from Mäkilä et al. (2018) (Fig. 4). The less pronounced gradient in measured decomposition rates above the water table depth is, however, also visible for *S. fuscum* replicates within the same study and in addition similar across independent studies (Fig. 4, supporting information S6) (Johnson and Damman, 1991; Golovatskaya and Nikonova, 2017; Mäkilä et al., 2018), indicating that this pattern cannot be explained in all cases by differences between studies. In addition, during the cross-validation, we removed data from individual studies from the model and the remaining subsets still resulted in similar estimates for $c_2$ and $W_{opt}$ (supporting Fig. S12). Finally, numerous previous studies suggest that water table depth is an important control of decomposition rates (e.g., Blodau et al., 2004) and one may therefore expect that also between different studies decomposition rate differences should be controlled to a large degree by differences in water table depths. Thus, even with the heterogeneous litterbag data currently available, a less steep gradient of decomposition rates from oxic to anoxic conditions appears to be replicable between studies and species. To fully rule out that this pattern may be biased by heterogeneous litterbag data and biases of the litterbag decomposition model, controlled litterbag experiments that systematically estimate decomposition rates along the gradient from oxic to anoxic conditions are needed.

The $W_{opt}$ estimate suggested by HPM-leaching is near the average optimum of heterotrophic respiration estimated across a range of mineral soils (Moyano et al., 2013). The estimate is also in line with a study where the largest decomposition rates of the same litter type were observed at or just above the average water table level in hummocks (Belyea, 1996), and with maximum $CO_2$ production rates around 13 cm above the water table level in a mesocosm study (Blodau et al., 2004). According to the ModGberg model the degree of saturation at this depth is near the $W_{opt}$ estimate suggested by HPM-all and

HPM-leaching. For example, for our simulation analysis used to produce Fig. 6, the average $W_{opt}$ estimated by model HPM-leaching (0.57 $L_{water}\ L_{pores}^{-1}$) is reached around 16 cm above the water table level, as shown in Fig. 7 (b). At shallower depths, the degree of saturation decreases below the $W_{opt}$ estimate and this would decrease decomposition rates as observed in Belyea (1996). In contrast, according to the the ModGberg model, a degree of saturation corresponding to the standard $W_{opt}$ value (0.45 $L_{water}\ L_{pores}^{-1}$) is reached at shallower depths and in the same simulation with this standard $W_{opt}$ value, no pronounced sub-surface peak in decomposition rates is observed (Fig. 7 (a)). In hollows, the optimum degree of saturation suggested by HPM-leaching is reached near the surface for either $W_{opt}$ value (supporting Fig. S15). Thus, a larger value for $W_{opt}$ would be compatible with results from several previous studies.

Larger and smaller $c_2$ than the standard value have been estimated for several peatland cores with the HPM and a modified version with monthly time step (Quillet et al., 2015; Treat et al., 2021, 2022). Smaller values have been estimated for tropical peatlands (Kurnianto et al., 2015). To our knowledge, no litterbag experiment directly estimated $c_2$. A difficulty is that available litterbag experiments cover only a comparatively small depth range below the water table level (at most around 30 cm, Fig. 4) and therefore gradients in anaerobic decomposition rates across larger depths below the water table currently cannot be estimated with available litterbag data.

The estimates for the maximum possible decomposition rate ($k_{0,i}$) have large errors and removal of data during the cross-validation caused larger relative differences in $k_{0,i}$ estimates compared to $W_{opt}$ and $c_2$ (supporting Fig. S12). On the one hand, this variability indicates that available litterbag data are not sufficient to estimate $k_{0,i}$ accurately and that our assignment of *Sphagnum* species to HPM PFT may not be optimal, but on the other hand, this variability may also indicate that categorizing *Sphagnum* species into three PFT may not accurately describe the variability of maximum possible decomposition rates. Several studies suggest that diverse aspects of litter chemistry may increase $k_{0,i}$ (Turetsky et al., 2008; Bengtsson et al., 2018). However, we are not aware of studies that systematically analyze what factors control $k_{0,i}$ within the same species.

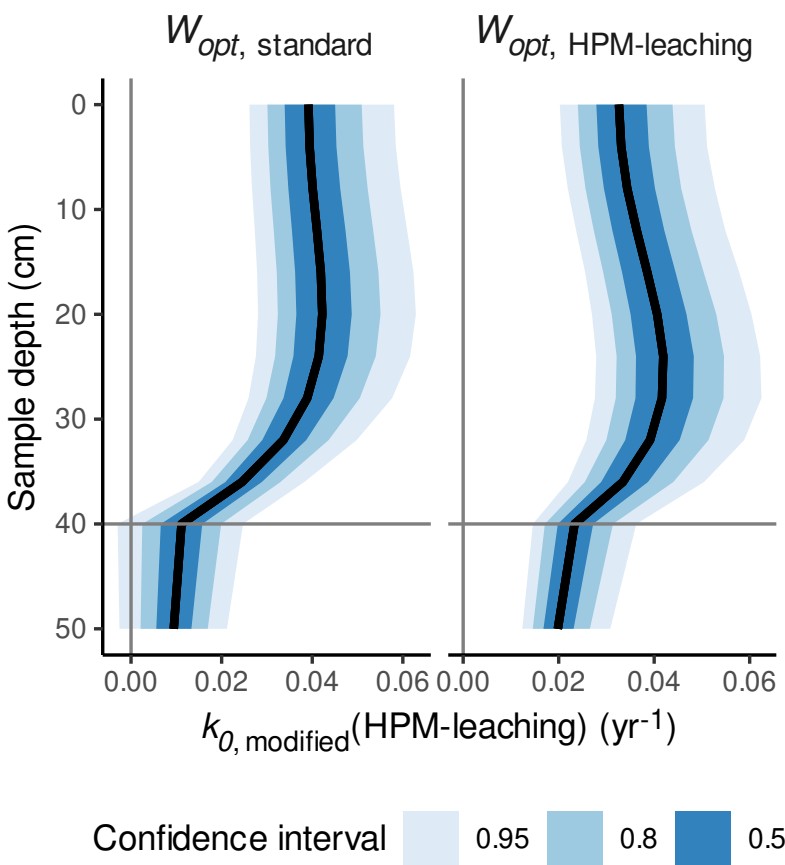

**Figure 7.** Decomposition rates predicted with HPM-leaching ($k_{0,\text{modified}}$(HPM-leaching)) for *S. fuscum* (hummocks), using either the standard value for $W_{opt}$ or the $W_{opt}$ value estimated by HPM-leaching versus depth of the litter below the peat surface. The horizontal line is the average water table depth. Shaded areas are central confidence intervals with probabilities given in the figure legend.

### 4.2 Water table fluctuations may explain the discrepancies in $c_2$ and $W_{opt}$ and larger anaerobic and smaller aerobic decomposition rates.

The HPM decomposition module predicts decomposition rates based on average annual water table depths (Frolking et al., 2010) and ignores water table fluctuations. Our evaluation of the HPM decomposition module also assumed an average water table depth during the litterbag experiments and the HPM decomposition module translated this into a clear pronounced transition between anaerobic and aerobic decomposition rates (Fig. 4). In reality, water table levels fluctuate and this causes transient and nonlinear changes in decomposition rates due to variations in the availability of oxygen and other electron acceptors, flushing of end products of anaerobic decomposition, and possibly other factors (Siegel et al., 1995; Blodau and Moore, 2003; Blodau et al., 2004; Beer and Blodau, 2007; Knorr and Blodau, 2009; Walpen et al., 2018; Campeau et al.,

2021; Kim et al., 2021; Treat et al., 2022; Obradović et al., 2023). A possible explanation why the gradient in decomposition rates from oxic to anoxic decomposition is less steep across litterbag experiments, on average, than suggested by the standard

HPM decomposition module could therefore be that an averaging effect of fluctuating water table levels on both aerobic and anaerobic decomposition rates is neglected by the HPM decomposition module. An additional factor may be that litterbags cover a depth range and therefore the decomposition rate estimate is an average over the depth covered by the litterbag. If moisture conditions vary over this depth, the decomposition rate estimate also averages over moisture conditions, with similar effects as the temporal average caused by water table fluctuations.

If this is the case, $c_2$ would have to be re-interpreted as transition parameter that accounts for both limitation of anaerobic decomposition under anoxic conditions and the effects of periodically oxic conditions. Similarly, $W_{opt}$ would have to be re-interpreted as the optimum average degree of saturation for decomposition under water table level variations and its value would be necessarily different from the optimum degree of saturation for depolymerization under static degree of saturation.

Adjusting the HPM decomposition module parameters as implied by our modified models may be an easy way to account

for the effect of sub-annual variation in water table levels on decomposition rates, if the discrepancies are caused by fluctuating water tables and if the model is representative for different effects variations in water table level may have on decomposition rates (e.g., short-term fluctuations compared to seasonal water table variations compared to prolonged droughts). What we have not considered due to limited data is that $c_2$ can be expected to depend on long-term changes in groundwater flow (e.g., Siegel et al., 1995) or site-specific differences in hydrology and other factors (e.g., Frolking et al., 2010; Treat et al., 2021, 2022).

Therefore, $c_2$ can be expected to differ between litterbag studies and our data only indicate that $c_2$ is larger on average, whereas more research is necessary to estimate and understand site-specific controls of $c_2$ and how a change in hydrology controls $c_2$. Similarly, $W_{opt}$ may differ between sites and over time. It would be interesting to know whether litterbag experiments can quantify these controls and whether $c_2$ estimated from litterbag experiments is generally larger in peatlands with larger water table fluctuations.

It is also worth mentioning that a modification of the HPM, HPM-Arctic (Treat et al., 2021), has a seasonally dynamic WTD and this modification may account for at least a part of the discrepancies we observed here. Unfortunately, most available litterbag data do not report WTD at sufficient temporal resolution to test whether standard HPM parameter values are more compatible with litterbag data when such seasonal variations in WTD are considered.

### 4.3  Implications of the discrepancies in $W_{opt}$, $c_2$, and $k_{0,i}$ for long-term C accumulation

A larger $c_2$ implies larger anaerobic decomposition and may thus indicate that the HPM decomposition module underestimates anaerobic decomposition rates. Previous global and local sensitivity analyses, where HPM parameter values and environmental conditions were varied in broad ranges, identified $c_2$ as influential for C accumulation in the HPM (Quillet et al., 2013a, b).

If $c_2$ is varied within the range from the standard value (0.3 m) to the average posterior estimate from HPM-leaching (0.64 m), this would cause differences in predicted C accumulation of a maximum of ca. 20% in the sensitivity experiment of Quillet

et al. (2013a) (depending on precipitation, Fig. 1 c in Quillet et al. (2013a)). If values are changed across the complete posterior

range compatible with litterbag data and if other HPM parameters would also be varied, the effect would be even larger (Fig. 2 c in Quillet et al. (2013a)).

Due to parameter interactions and feedbacks, an increase in anaerobic decomposition rates can result in smaller or larger C accumulation of the HPM, depending on environmental conditions (Quillet et al., 2013a). Small anaerobic decomposition may cause too rapid C accumulation resulting in a low water table level, a thick aerobic zone, and thus smaller overall C accumulation after a longer time. Larger anaerobic decomposition may result in higher water table levels and this can increase C accumulation in the long-term. Too large anaerobic decomposition decreases C accumulation (Quillet et al., 2013a).

A larger $W_{opt}$ implies that the largest aerobic decomposition rates are reached under more saturated conditions. $W_{opt}$ has not been identified as influential in a sensitivity analysis of the HPM (Quillet et al., 2013a), but as shown above, it contributes to the less steep decrease of decomposition rates from oxic to anoxic conditions. Importantly, since the HPM does not have a seasonally resolved water table depth, the two sensitivity analyses did not consider how seasonal variations of the water table depth may control long-term C accumulation, and consequently the re-interpreted $W_{opt}$ may be more important to long-term C accumulation than previously assumed. In addition, HPM-leaching suggests an average $W_{opt}$ value of 0.57 $L_{water}\ L_{pores}^{-1}$, which is outside the range of values tested in Quillet et al. (2013a) (0.3 to 0.5 $L_{water}\ L_{pores}^{-1}$). This implies that the sensitivity of long-term C accumulation to $W_{opt}$ has been evaluated over a too small range.

A larger $k_{0,i}$ increases decomposition rates for a species and *Sphagnum* $k_{0,i}$ are particularly relevant for many peatlands because the bulk of the peat is *Sphagnum* peat. In the sensitivity analysis in Quillet et al. (2013b), $k_{0,\text{hummock}}$ had large interaction effects with other parameters of the HPM and therefore could either cause larger or smaller peat accumulation, depending on environmental conditions, other parameters, and what vegetation shifts occur in a specific case. Similar to $W_{opt}$, our $k_{0,i}$ estimates have errors that are larger than the range of values tested in Quillet et al. (2013b). For example, for hummock *Sphagnum* species, $k_{0,i}$ was varied from 0.04 to 0.06 $yr^{-1}$, whereas average estimates for $k_{0,i}$ of HPM-leaching for species assigned to the hummock PFT range from 0.04 to 0.19 $yr^{-1}$. As mentioned above, this range of $k_{0,i}$ estimates may be biased because of the difficulty to assign *Sphagnum* species to HPM PFT, but from a different perspective, this is an additional error source for $k_{0,i}$ estimates that should be considered in sensitivity analyses unless more evidence becomes available to define PFT and their maximum possible decomposition rates.

A further aspect that needs to be considered is that HPM-all and HPM-leaching estimate parameter distributions based on available data, whereas existing studies defined fixed parameter values or ranges of parameter values based on expert knowledge. Based on Quillet et al. (2013a), the uncertainties would cause non-negligible differences in predicted long-term C accumulation. For example, values within the uncertainty range of $c_2$ estimated by HPM-leaching ((0.4, 0.97), 95% confidence interval), would imply differences up to 100 kg m$^{-2}$ of accumulated C over 5000 years in some simulations (Fig. 1 (c) in Quillet et al. (2013a), with a maximum total accumulation of ca. 430 kg$_C$ m$^{-2}$). Simulations of remaining masses for different *Sphagnum* species under different conditions also indicate large uncertainties in predicted remaining masses (supporting info S9). This implies that more work is required to estimate parameters accurately enough to detect even relative large differences among peatland models and between model predictions and peat cores.

Summarized, based on existing sensitivity analyses of the HPM the parameter discrepancies suggested by HPM-all and HPM-leaching can translate into non-negligible differences in long-term C accumulation rates. They also imply gaps in previous sensitivity analyses of the HPM, namely that $W_{opt}$ and possibly $k_{0,i}$ (for some species) have been analyzed over a too restricted value range and may play a more important role if water table fluctuations are taken into account.

### 4.4 How can we improve tests of peatland decomposition modules?

We suggest the following steps to improve accuracy when estimating peatland decomposition module parameters:

1. High temporal resolution measurements of WTD: For many available litterbag studies, it is not clear whether reported WTD estimates are unbiased estimates of average WTD (i.e., are derived from high-resolution measurements during the incubation) or biased (due to a too small temporal resolution or coverage). This limitation could be reduced by reporting high temporal resolution WTD measurements along litterbag experiments. Such data are also necessary to investigate
whether HPM decomposition module parameters are controlled by WTD fluctuations.

2. Eliminate the need of auxiliary models to estimate the degree of saturation: There is a lack of data on the degree of saturation (or porosity and volumetric water content, from which the degree of saturation could be computed) for available litterbag experiments. For this reason, we used the modified Granberg model to estimate the degree of saturation based on reported WTD and an assumed peat porosity. The modified Granberg model, reported WTD, and our assumed
peat porosity are error sources for our test. This limitation could be reduced by measurements of peat porosity and high temporal resolution measurements of volumetric water content during litterbag experiments.

3. Implementing a standard for how to assign *Sphagnum* species to model PFT: The HPM does not specify how to assign *Sphagnum* species to PFT (Frolking et al., 2010), which makes it difficult to compare litterbag experiments to parameters for HPM PFT. Ideally, peatland models should provide lists of species they assign to certain PFT to facilitate tests.
Moreover, available niche data used here to assign species to PFT may be biased by short term measurements during summer that are not in line with average niches defined in peatland models, similar to how transfer model for testate amoebae are suggested to be biased (Swindles et al., 2015).

4. Decreasing errors in $k_0$ and $l_0$ estimates from litterbag experiments: Our analysis suggests that a comparatively large range of $c_2$, $W_{opt}$, and $k_{0,i}$ estimates in the HPM decomposition module are compatible with available litterbag data
because errors in remaining masses are large enough to support a range of $k_0$ and $l_0$ estimates and because of deficiencies in the design of the litterbag experiments. As a consequence, $k_0$ estimates of the litterbag decomposition model can be adjusted to fit predictions of the HPM decomposition module for a range of HPM decomposition module parameter values. We also assume that because of these large errors and a large variability of initial leaching losses due to differences in litter handling (Teickner et al., 2025a), we could not detect an expected positive relation of $l_0$ to the degree of saturation
(Lind et al., 2022). Future litterbag experiments that aim to improve peatland models should reduce errors of $k_0$ and $l_0$ estimates. A first step would for example be to modify litterbag experiments as described in Teickner et al. (2025a).

5. Systematic litterbag experiments along the gradient from oxic to anoxic conditions: There are few litterbag experiments available that systematically analyze how decomposition rates differ along the gradient from oxic to anoxic conditions. Problems are that many studies test only few conditions and do not cover depth ranges large enough to estimate the minimum decomposition rate ($f_{min}$) and $c_2$. An ideal study would use litter material of the same species and origin (thus making sure $k_{0,i}$ would be the same for all replicates) and systematically record remaining masses under different degrees of saturation in the same peat material to accurately estimate $W_{opt}$ and $c_1$. Another ideal study would systematically record remaining masses at many depth levels, and deeper than 30 cm below the average annual WTD to allow accurate estimation of $c_2$. Similar experiments could be used to estimate how WTD fluctuations affect decomposition rates along the gradient from oxic to anoxic conditions and how this would change estimates for $W_{opt}$ and $c_2$.

6. Understanding the controls of $k_{0,i}$: Values of $k_{0,i}$ can be assumed to be controlled, among other factors, by litter chemistry. Even though there are studies that analyze how litter chemistry controls decomposition rates (e.g., Turetsky et al., 2008), there are few that do this systematically (e.g., Bengtsson et al., 2018) and these do not consider initial leaching losses and thus may confound initial leaching and decomposition, both of which may depend on initial litter chemistry. Studies that systematically change litter chemistry within species would be required to estimate $k_{0,i}$ (e.g., Siegenthaler et al., 2010; Straková et al., 2010). These estimates would also be useful to define PFT for decomposition modules.

7. Understanding how $c_2$ and $W_{opt}$ vary between sites and in dependency of peat characteristics: Too few litterbag experiments with too few replicates are available to estimate $c_2$ and $W_{opt}$ separately for individual sites (or how they may vary over time). Systematic litterbag experiments are needed to estimate how environmental conditions control the magnitude of these parameters, for example due to temporal variations in water and oxygen availability or differences in availability of alternative electron acceptors under anoxic conditions.

Systematic and high-quality litterbag experiments that are designed specifically to test peatland decomposition modules are required to achieve these improvements. To support the design of such experiments, we created an R package (hpmdpredict, supporting information S10) that allows to make predictions with HPM-leaching for hypothetical litterbag experiments and that also allows to change parameter values (Teickner and Knorr, 2025). This could for example be useful to estimate the sample sizes that are required to detect specific differences in remaining masses, to test to what extent litterbag experiments are compatible with HPM-leaching, or to analyze the effect of changing HPM decomposition module parameter values from the standard values or our estimates.

## 5    Conclusions

Based on the litterbag data, our estimates for the degree of saturation where decomposition is optimal ($W_{opt}$) and the anoxia scale length ($c_2$) are significantly larger than the standard parameter values. Moreover, maximum possible decomposition rates ($k_{0,i}$) for individual species are overall more variable than implied by the standard HPM decomposition module parameter

values. According to previous sensitivity analyses, these parameter estimates imply differences in predicted C accumulation rates of up to 100 $kg_C$ m$^{-2}$ over 5000 years (with a maximum total C accumulation of ca. 430 $kg_C$ m$^{-2}$) when compared to the standard parameter values. The differences in HPM parameter estimates imply larger anaerobic decomposition rates for several species and a less steep gradient of decomposition rates from oxic to anoxic conditions. This pattern may be caused by water table fluctuations, differences in groundwater flow, or spatial averaging in litterbag experiments; factors that are currently not explicitly considered both in the HPM decomposition module and available litterbag experiments.

Our analysis suggests that the HPM decomposition module with standard parameter values fits available *Sphagnum* litterbag data, but model versions where HPM decomposition module parameters were estimated from available litterbag data have an equivalent fit. This is caused by two mechanisms: First, remaining masses in litterbag experiments can be explained by initial leaching losses and decomposition. If remaining masses are reported only some time after the initial leaching loss has happened, they can be explained either by small initial leaching losses and a large decomposition rate or by large initial leaching losses and a smaller decomposition rate. Second, the majority of available *Sphagnum* litterbag experiments reports remaining masses only a long time after the initial leaching loss happened. Taken together, this means that available litterbag data are compatible with a broad range of decomposition rates suggested by HPM decomposition module versions with large differences in parameter values. Improved litterbag experiments are needed for more accurate tests of any peatland decomposition module and for obtaining parameter estimates accurate enough to allow even only approximate predictions of long-term peat accumulation. Applications of the HPM and any other peatland model that relies on litterbag data to parameterize its decomposition process should consider that a broad range of decomposition module parameter values is compatible with available litterbag experiments.

The modeling approach used here can be combined with different data sources and peatland decomposition modules and therefore may serve as blueprint for future tests and to obtain more accurate parameter estimates once improved litterbag experiments are available. In light of the large differences in long-term peat accumulation suggested by the parameter estimates, we conclude that it is worth to conduct such litterbag experiments, not only to improve the decomposition module of the HPM, but to improve peatland models in general.

. Data and code to reproduce this manuscript are available from Teickner et al. (2025b). The data used in this study are derived from Teickner and Knorr (2024). An R package to make predictions for litterbag experiments with model HPM-leaching is available from Teickner and Knorr (2025).

. HT: Conceptualization, methodology, software, validation, formal analysis, investigation, data curation, writing - original draft, visualization, project administration. EP: supervision, funding acquisition, writing - review & editing. KHK: supervision, funding acquisition, writing - review & editing.

. The authors declare no competing interests.

. This study was funded by the Deutsche Forschungsgemeinschaft (DFG, German Research Foundation) grant no. KN 929/23-1 to Klaus-Holger Knorr and grant no. PE 1632/18-1 to Edzer Pebesma.

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
