# Peer review of "Peat Oxic and Anoxic Controls of *Sphagnum* Decomposition Rates in the Holocene Peatland Model Decomposition Module Estimated from Litterbag Data"

_EGUsphere, 2024_

## Author Comment (AC1)

**Reply to Anonymous Referee #1 for "Underestimation of Anaerobic Decomposition Rates in Sphagnum Litterbag Experiments by the Holocene Peatland Model Depends on Initial Leaching Losses"**

Henning Teickner[1,2,*]     Edzer Pebesma[2]     Klaus-Holger Knorr[1]

19 December, 2024

[1] ILÖK, Ecohydrology & Biogeochemistry Group, Institute of Landscape Ecology, University of Münster, 48149, Germany
[2] IfGI, Spatiotemporal Modelling Lab, Institute for Geoinformatics, University of Münster, 48149, Germany

[*] Correspondence: Henning Teickner <henning.teickner@uni-muenster.de>

Comments made by the reviewer start with a bold **Q** while our reply starts with a bold **A**. In section "Additional changes" we list additional changes we would like to incorporate in an updated version of the manuscript.

**1 Reply to comments**

1. **Q:** The manuscript entitled "Underestimation of Anaerobic Decomposition Rates in Sphagnum Litterbag Experiments by the Holocene Peatland Model Depends on Initial Leaching Losses" by Teickner et al tested decomposition rates from the Holocene Peat Model against litterbags decomposition experiments.

   The approach is interesting as it combines litterbag data and a Bayesian approach to improve the parametrisation of a model. Improving the use of litterbag data into peat decomposition models is timely and useful to further advance peat decomposition models. The manuscript is well structured. Unfortunately, major concerns regarding the scope, methods and some assumptions should be addressed before considering this manuscript for publication.

   **A:** We thank the reviewer for their comments and questions that are useful to clarify some points we make in the manuscript. We hope that our comments below and the suggested changes in the manuscript address the concerns.
   Please note that due to the extnsive reviewer comments, we made extensive changes

to the manuscript. Instead of listing all these changes out of context, we therefore attached a version of the manuscript that includes all suggested changes at the end of this document.

**1.1 General comments:**

2. **Q:** The motivation, overall approach and scope of the study are not clearly presented. Was the purpose of the study to identify an improved range of values for Wopt and c2?

   **A:** The aims of the study are twofold (compare with our manuscript, l. 77 to 82):

   1. We wanted to test whether the HPM decomposition module can predict decomposition rates estimated from available *Sphagnum* litterbag experiments using the standard parameter values reported in Tab. 1 and Tab. 2 in the original publication (Frolking et al., 2010).

   2. We wanted to estimate parameters of the HPM decomposition module from available litterbag data and analyze whether they differ from the standard parameter values defined in Frolking et al. (2010), what implication this has for the decomposition process, and whether they are negligible in terms of long-term peat accumulation as predicted by the HPM.

   Due to limitations of available litterbag experiments and limited information reported in these experiments discussed in our manuscript, it is not clear yet whether our estimates are an improvement and therefore we do not mention this as aim, even though it is of course a long-term goal.

   To better describe our aims, we suggest to change l. 79 to 82 to:

   1. "Test whether the HPM decomposition module can predict litterbag decomposition rates for different *Sphagnum* species along the gradient from oxic to anoxic conditions.

   2. Estimate HPM decomposition module parameters from litterbag data and compare them to the originally suggested values (standard parameter values) (Frolking et al., 2010) that are often used when applying the HPM (Tab. 1).

   3. If some of the parameter values differ, identify the possible causes why parameter estimates from litterbag data differ to provide guidance for future litterbag experiments.

   4. Analyze whether estimated differences in HPM parameter values could imply significant differences in decomposition rates and long-term peat accumulation."

   We are not sure whether the comment of the reviewer is due to a misunderstanding of our aims (i.e., different expectations of what is a good test of a model, please see also comments 8 and 13 by reviewer 1) or our failure to communicate our aims appropriately. We would be thankful if the reviewer could clarify whether he or she

still thinks that the aims of the study are not appropriately communicated, so that we can try to further improve our manuscript if necessary.

3. **Q:** It is obvious that some interesting experiment has been done. Unfortunately, it is extremely difficult to judge from the current version of the paper if it is sound and whether is constitutes a significant contribution to the field.

   **A:** We thank the reviewer for the comment and apologize for the missing information in our manuscript. We suggest to make extensive changes to the methods section to describe in detail the formulas of the HPM decomposition module, of the litterbag decomposition model, and of the link between both.
   In particular:

   1. Reviewer 3 suggested to include a conceptual figure of our modeling approach and we will do so in the next version (Fig. 1 in the attached manuscript).

   2. We suggest to update the methods section to include the missing information. Please see our replies to comments 13, 14, 18 - 20, 22, 23, 30, 31 of reviewer 1.

4. **Q:** Also, it would be helpful to clarify what the difference is, in terms of contribution, between this paper and "A Synthesis of Sphagnum Litterbag Experiments: Initial Leaching Losses Bias Decomposition Rate Estimates" by Teickner et al., as the conclusions seem quite similar.

   **A:** The only similarity in the conclusions of both papers is that errors in initial leaching loss ($l_0$) estimates are an important control of bias and errors in decomposition rate ($k_0$) estimates. Even though our manuscript here builds on top of the *Sphagnum* decomposition model in Teickner et al. (2024a), both manuscripts have different scopes, address different questions and perform completely different analyses.
   Teickner et al. (2024a) tries to understand how initial leaching losses control bias and errors in decomposition rate estimates directly derived from *Sphagnum* litterbag experiments.
   In our manuscript here, we test to what extent the HPM decomposition module can predict decomposition rates in the same litterbag experiments from plant functional type identity, water table depth and degree of saturation. Moreover, we estimate parameters of the HPM decomposition module from the litterbag data and how they differ from the parameter values defined in the original publication of the HPM (Frolking et al., 2010). We also discuss what these discrepancies imply for the peat decomposition process modeled by the HPM and for long-term peat accumulation modeled by the HPM (please also see our reply to comment 2 of reiewer 1). We suggest to re-write large parts of the Introduction section to reduce dependency of this manuscript to our previous study and to better explain how both are connected.
   We apologize that we did not provide sufficient details on the HPM decomposition module here and how we estimated its parameter values from litterbag data. We think that this may have caused the confusion and we hope that our changes suggested in comments 13, 14, 18 - 20, 22, 23, 30, 31 of reviewer 1 address this issue.

5. **Q:** The experimental design and methods need to be well justified. The experimental design includes a series of assumptions that need to be justified in more detail, so that the reviewer can assess of the quality and soundness of the study. Please see specific comments for details.

   **A:** We thank the reviewer for pointing out several ambiguous sections of our manuscript. We hope that our replies to comments 14, 18 - 20, 22, 23, 30, 31 of reviewer 1 address this issue.

6. **Q:** Methods should include more detail to allow the reader to reproduce the experiment. Some areas of the methods are well covered. However, it remains unclear what has been done with the litterbag data and with the model. More detail on the models, including HPM, as well as the Bayesian tools used are needed.

   **A:** We hope that our replies to comments 14 and 22 of reviewer 1 address this issue. The reviewer does not mention what details on the Bayesian tools used are needed and we think that we provide detailed information on the Bayesian data analysis in section 2.4 (section 2.3.2 in the updated version of the manuscript) and the supporting information. We also emphasize that all code to reproduce our analyses are published in Teickner et al. (2024b). If the reviewer has specific comments on details missing about the Bayesian data analysis, we are thankful for further explanations.

7. **Q:** The experiment should be presented as a stand-alone in the paper. Information from Teickner et al. 2024 and its supplementary material was useful to get some sense of what has been done here but it should not be necessary to read another paper to understand this study. If the paper draws from the other paper's conclusions, then maybe summarize the previous conclusions and explain the link and contribution of this paper.

   **A:** We suggest extensive updates of the Introduction and Methods section, where we de-emphasize the role of our previous publication that describes the litterbag decomposition model, and describe in detail our modeling approach. We hope that our suggested changes to the manuscript that describe the modeling approach in more detail (please see our reply to comments 14 and 22 by reviewer 1) clarify how the litterbag decomposition model from our previous manuscript is included in our model here and provide sufficient information that allows to understand our current study without the need to read our previous manuscript.

8. **Q:** Assessing the validity of the decomposition functions of the model is interesting. It does not constitute a test of the validity of the whole model. There are therefore limitations around the conclusions that can be drawn from this experiment. The limitations of the study should be clearly presented.

   **A:** We agree with the reviewer that testing a module of a model is no test of the whole model. Our aim was not to test the HPM but the decomposition module of the HPM, as stated in the introduction. We agree that our wording was ambiguous in some places and we suggest to correct this (please see our reply to comment 9 of reviewer 1). However, we do not think that it is a limitation that our study only focuses on the HPM

decomposition module. Instead, we consider it a strength that our focus on only one aspect of the HPM allows us to derive explicit aspects that, when considered in future tests, will give novel insights. Please also see our reply to comment 13 of reviewer 1 where we provide additional detailed justification of our approach to test the HPM decomposition module. We suggest to include a separate subsection at the end of the discussion section (section 4.4 in the attached manuscript with the suggested changes) where we summarize limitations of our test and what experiments and data are needed to improve our approach to test decomposition modules of peatland models.

9. **Q:** Terminology: some technical terms are not consistently or not appropriately used throughout the paper, which brings confusion: e.g. model, prediction, equation, confidence level. Please check usage throughout the text. Also, some statements mention that the decomposition module was tested in this study and some statements indicate that this is not what was done here.

   **A:** We suggest to correct and clarify all instances where our statement may imply that we tested the whole HPM or a modified version of the HPM by explicitly stating that we tested the HPM decomposition module or a modified version of the HPM decomposition module.
   We suggest to correct one instance where "confidence interval" instead of "confidence level" may be used (see our reply to comment 42 of reviewer 1).
   We suggest that our usage of "model equation" questioned in comment 44 of reviewer 1 is not inappropriate (see our reply to comment 44 of reviewer 1).
   We are not aware of instances where we used the terms "model" or "prediction" inconsistently or inappropriately or of specific comments by reviewer 1 where such misuses are mentioned. In case the reviewer disagrees, we would be happy to see specific parts/sentences where this applies and would then include changes or clarification.

10. **Q:** Readability: In many places, the statements are difficult to follow, which makes the paper hard to read. For example, sentence L4-6 should be rephrased to improve clarity.

    **A:** We thank the reviewer for this suggestion. We suggest to simplify statements where they were mentioned in specific comments (please see our reply to comment 36 by reviewer 1). In addition, we extensively reviewed the text and identified parts of the text that can be simplified and suggest to do so in the updated version of the manuscript.

11. **Q:** Conclusions: The gradient from oxic to anoxic conditions in HPM is designed for long-term peat decomposition within a dynamic simulation. Is it therefore possible to draw conclusions on its suitability from this experiment?

    **A:** We agree that litterbag experiments cover shorter time periods than are typically modelled with the HPM. It is also correct that changes in litter chemistry over longer periods of decomposition change the decomposition rate and this effect is not covered by litterbag experiments. Apart from this, long-term decomposition is the outcome of the same decomposition process happening on short time scales after a long time. Thus, even though litterbag experiments do not cover all aspects of decomposition, a fact

highlighted both by Frolking et al. (2010) and in the introduction of our manuscript (ll. 41 to 43), there currently is no better option to directly test how accurate the decomposition process is represented in long-term peatland models other than litterbag experiments.

Due to the assumption that the decomposition process is essentially the same, the HPM was explicitly developed based on litterbag studies, and in particular how the effect of degree of saturation (oxygen availability) and depth below the water table are modeled and how parameter values are defined (Frolking et al., 2010). Moreover, the HPM has an annual time step and also models decomposition of fresh litter over time periods covered by litterbag experiments. We therefore think that litterbag experiments are suitable to test the HPM decomposition module and are not aware of other options that have fewer limitations.

Since we mention this limitation in the introduction, we do not suggest to include changes in the manuscript that address this comment, unless the reviewer thinks this point is still not sufficiently addressed.

**1.2   Specific comments:**

12. **Q:** L1: The title needs to be revised as the litterbag experiments were not conducted with the HPM per se. Also, it might be worth considering a title in line with the general scope of the study.

    **A:** We agree that the title is currently ambiguous as we focus on the decomposition module of the HPM. We also agree that the title can be improved to better summarize all important aspects of our study and shorten it. We therefore suggest to change the title of the manuscript to:

    "Peat Oxic and Anoxic Controls of *Sphagnum* Decomposition Rates in the Holocene Peatland Model Decomposition Module Estimated from Litterbag Data"

13. **Q:** L35: Could you please justify and explain this approach? Considering only a section of the model does not seem to be an adequate method to test the validity of a model.

    **A:** We agree that considering only a part of a model is not an adequate method to test the validity of a model and we did not want to imply that we tested the validity of the entire HPM.

    Previous tests of the HPM and other peatland models typically compare predictions for properties of peat cores after hundreds or thousand years of past or simulated peat accumulation (e.g. Frolking et al. (2010), Tuittila et al. (2013), Quillet et al. (2015), Zhao et al. (2022)). These models test predictions conditional on the complete HPM, but they are of limited use when it comes to detecting what process causes discrepancies and what concrete actions are necessary to improve the HPM because of parameter unidentifiability, many omitted processes, and the practical limitation that peat cores with millenia of peat accumulation cannot be produced in experiments. In philosophy of science, the problem that many possible alternative hypotheses (including sets of parameter values) can equally well explain (fit) a phenomenon is known as Duhem-Quine problem and has been widely discussed (see e.g., Mayo (1996)). A necessary

condition to improve peatland models is to identify what process causes predictions to differ from measured peat properties (e.g. accumulated peat mass) and tests of complete peatland models are not apt for this job because of the Duhem-Quine problem. To overcome the Duhem-Quine problem, scientists conduct piece-meal tests of complex models or research problems (Mayo, 1996). Our study here can be seen as application of some part of this approach to the decomposition module of the HPM. Thus, when we split off the decomposition module from the rest of the HPM, we can test whether the HPM decomposition module correctly predicts decomposition rates for specified plant functional types (PFT), degrees of saturation, and water table depths (WTD), while we do not also have to worry whether the HPM describes litter production, peatland hydrology, root production, changes in bulk density with decomposition, etc. correctly or whether our weather data are correct for the past millenia. This test is possible because suitable experiments exist where we can control (experimentally or statistically) PFT, degree of saturation, and water table depth and from which we can estimate decomposition rates. When the HPM decomposition module fails to predict decomposition rates in litterbag experiments, this certainly indicates a discrepancy between the HPM and a specific aspect of the peat accumulation process and identifies a concrete opportunity to improve one specific aspect of the HPM, instead of just identifying a general problem of the HPM, where the error source is not clearly identified.

We tried to explain this rationale in the introduction while keeping the focus on the problem we try to address (see ll. 25 to 43). We also note that our study has many limitations and because of these our test applies only a fraction of the severe testing approach developed in Mayo (1996) (typically, a series of well designed experiments is necessary to test some subject matter hypothesis severely). However, our study is useful in identifying — at no additional experimental costs — what differences in parameter values may be expected, whether these would have important consequences, and what kind of new litterbag experiments and data are needed to improve our test and estimates.

We therefore think that tests of individual modules typically give more targeted information on what aspects of a model are inadequate and how to improve modules and tests of modules; tests of whole ecosystem models provide insights equivalent to these piece-meal tests only in rare occasions.

We currently think that lines 25 to 43 (old manuscript) justify our approach (also since the other reviewers did not make similar comments), but if the reviewer has specific comments on how our approach may not be well justified, we are interested to hear their suggestions.

14. **Q:** L73: Could you please give more detail for clarification: Are the decomposition rates predicted by the HPM obtained from HPM simulations?

**A:** We thank the reviewer for this very helpful comment and apologize that the first version of the manuscript did not contain a more detailed description of our model. Although these information are also given in supporting information S1, we agree that it is necessary to include a more detailed description of the actual formulas from the HPM we consider as decomposition module and how we link *Sphagnum* litterbag experiments with this decomposition module (compare also with comment 22 of reviewer

1) to avoid any ambiguities.

We suggest a complete re-write of the Methods section to describe our modeling approach in detail, to describe how the HPM decomposition module is used as prior for the litterbag decompostion model, and what model versions we computed. These changes are too extensive to list each of them individually, therefore we attached the manuscript with the planned changes included. Please see this attachment for the suggested changes to the methods section.

15. **Q:** L81: What kind of information does that deliver? Identifying new estimates to be used in future HPM versions would seem a better aim?

    **A:** The long-term goal certainly is to improve the accuracy of parameter estimates to be used in future HPM versions. However, currently there is no information on how accurate the parameter values of the HPM decomposition module are when estimated from data and the first steps (our study provides) therefore are to estimate parameters from data, test whether and how they differ from previous values, and analyze what could cause differences in parameter values and whether there is sufficient evidence that these differences actually are an improvment (or rather require to improve the test).

    Our analysis indicates some differences, but we discuss that these differences could either be caused by deficiencies of current litterbag experiments or indeed represent improved parameter estimates, and our analysis suggests how to improve litterbag experiments to rule out the former. Thus, identifying new estimates to be used in future HPM versions would be a better aim, but first one has to develop a strategy that ensures that parameter estimates derived from litterbag experiments indeed are improvements. This is what our study does.

    We agree that our study does not merely compare parameter values, but also tries to identify possible causes and consequences for peat accumulation. We therefore suggest to modify the description of the second aim and include a third and fourth aim to amphasize this (see our reply to comment 2 of reviewer 1).

16. **Q:** L92: Could you be more specific? It seems that HPM's decomposition module was not tested here.

    **A:** We are not sure why the reviewer has the impression that the HPM decomposition module was not tested here, even though we mention this as explicit aims in previous parts of the introduction.

    We agree that in some places we ambiguously wrote something like "predicted by the HPM" or "HPM modification" throughout the text and we will change these instances to make clear that we refer to the HPM decomposition module.

    We also agree that we do not test the HPM decomposition module alone, but that we need auxiliary models to link it to the litterbag data and to estimate the degree of saturation. However, these limitations are clearly stated in our manuscript (following the reviewer comments, some of these are clarified), and in addition, we suggest to summarize these limitations in a separate subsection in the Discussion section (please see our reply to comment 8 of reviewer 1). Despite these auxiliary models, we consider

our study a test of the HPM decomposition module because it isolates this module from the HPM and tests it against data.

17. **Q:** L115: It is unclear how this can be done while excluding the interactions with the other modules of the model. Could you please give more details?

    **A:** We assume that the reviewer asks how "… to link decomposition rates estimated from litterbag data to the decomposition rates predicted by the HPM …" (l. 115) is possible (or useful?) while excluding the interactions with the other modules of the model. We hope that our reply to comment 14 of reviewer 1 addresses this question and that our modifications to the Methods section mentioned there provide the necessary changes to the manuscript.

18. **Q:** L139: Would it not be more appropriate to reject this study as it brings a lot of uncertainty in the results?

    **A:** The reviewer refers to (l. 139) "Litterbag data from Prevost et al. (1997) are incubations of peat samples where the species is unknown."
    It is true that this study can be be assigned to one of the three *Sphagnum* PFT in the HPM only ambiguously and this potentially introduces errors when testing the HPM. In particular, this mainly introduces errors when estimating $k_{0,i}$.
    However, we note that assigning litterbag experiments where the species was known to one of the three *Sphagnum* PFT in the HPM also is ambiguous (as described in section 2.2.2 of our manuscript) and that all samples required estimating $k_{0,i}$ to make the decomposition rates from the litterbag experiments compatible with predictions of the HPM decomposition module. In particular, our analyses do not indicate that data from this study introduce additional variation which would be different from other studies. At the same time, there are few experiments where WTD were reported and we would therefore lose useful information if these samples were dropped. We agree that there is a need to improve estimates for $k_{0,i}$ in general and this is discussed in section 4.6 of our manuscript (section 4.4 in the updated version).

19. **Q:** L146: Could you please explain why this is needed and how this differs from the approach taken in HPM?

    **A:** We think the reviewer refers to why the degree of saturation had to be estimated and why we did not use the equation for the degree of saturation used in the HPM (equations (16) to (18) in Frolking et al. (2010)), but the ModGBerg model (Granberg et al., 1999; Kettridge and Baird, 2007).
    We indeed do think that a large limitation of available litterbag data — and therefore also our study — is that measurements of the degree of saturation (ideally with high temporal resolution) are lacking. In line with our approach to test the decomposition module of the HPM in isolation of all other modules of the HPM, it makes sense to use an estimate for the average annual degree of saturation as accurate as possible because then discrepancies the test detects cannot be caused by an incorrect representation of the degree of saturation.
    As mentioned in the manuscript (ll. 202 to 206, old version), an improved test of the

HPM decomposition module would use direct measurements of the degree of saturation. Since these data are not available, it makes sense to use a model to estimate the degree of saturation that is accurate and uses only the information available from litterbag data (often, this is only WTD and depth of the sample below the peat surface). The ModGBerg model (Granberg et al., 1999; Kettridge and Baird, 2007) is derived from comparatively accurate laboratory measurements, whereas it is less clear how the equations used in the HPM are exactly derived. We are aware that more sophisticated models exist to model peat water content, but these require additional data not available from available litterbag studies.

Therefore, we do not consider as limitation *which* of the available approximate models for the degree of saturation applicable to available litterbag data to choose, but *that* accurate estimates/measurements for the degree of saturation are currently not available for litterbag experiments. Of course, this limits the severity of our test which is why we emphasize the need to do additional experiments that rule out these limitations in our manuscript (see for example the conclusion section) to replicate our results and to improve our test (ll. 202 to 206, old version).

To even more explicitly state this limitation, we suggest to add in l. 150 (old ersion): "An improved test of the HPM decomposition module would require litterbag experiments with direct measurements of the degree of saturation at sufficient temporal resolution."

20. **Q:** L158: It is unclear why 2 PFTs were assigned. Could you please give more details on your approach?

    **A:** We thank the reviewer for pointing out that we were not explicit enough here. The reason for assigning two dummy PFT to samples from Prevost et al. (1997) is that sample material of two different origins were used in this study and we can therefore assume that at most two different values for $k_{0,i}$ are necessary to predict decomposition rates with the HPM decomposition module.

    To improve our description, we suggest to change ll. 155 to 156 from

    "Prevost et al. (1997) incubated *Sphagnum* peat collected from different depths below the surface and these samples probably have already experienced some decomposition, however it is difficult to estimate how much"

    to

    "Prevost et al. (1997) incubated *Sphagnum* peat collected from two different depth levels from the same location and these samples probably had already experienced some decomposition, however it is difficult to estimate how much."

21. **Q:** L160: Could you please specify what is included or modified in the model versions? How are these model versions linked to the decomposition equations from Frolking et al. 2010?

    **A:** We thank the reviewer for pointing out that we omitted important information here. We hope that our reply to comment 14 and our changes to the manuscript suggested there address this issue.

22. **Q:** L169: Could you please explain how they are combined?

    **A:** We thank the reviewer for this helpful suggestion. We hope that our reply to comment comment 14 by reviewer 1 as well as the suggested changes to the manuscript address this question.

23. **Q:** L203: Could you please explain why uncertainties in water table depths can be considered negligible?

    **A:** The sentence starting in l. 203 is "Strictly, we do not test the decomposition module in the HPM, but the combination of the decomposition model in the HPM and the modified Granberg model, assuming that uncertainties in water table depths are negligible and that we accounted sufficiently for uncertainties in total porosity."
    This sentence does not state that uncertainties in water table depths can be considered negligible but that our analysis assumes this for simplicity because no sufficient information to consider this uncertainty are available. We also want to emphasize that we estimated a standard deviation for each annual WTD estimate and, where multiple WTD measurements were available in a study, we estimated standard deviations for the average annual WTD from these values to consider errors in average annual WTD estimates. However, often WTD values were aggregates or not reported or measured with sufficient temporal resolution and therefore all of these estimates are only approximate.
    To reduce any ambiguity, we suggest to change the sentence starting in l. 205 from

    "Litterbag experiments where the degree of saturation is measured would be needed to avoid this ambiguity."

    to

    "Litterbag experiments where water table depths and the degree of saturation are measured at sufficient temporal resolution are needed to avoid this ambiguity in future studies and to improve any test of the HPM decomposition module."

24. **Q:** L208: Model descriptions need more details. A separate section could be appropriate.

    **A:** We hope that our reply to comments 14 and 22, the additions to the manuscript suggested there, and the information provided in Tab. 2 (a new table that summarizes general information on the littebag studies per species) address this comment.

25. **Q:** L254: Table 3, inaccuracy: different versions of the decomposition module combined with other tools. Also, could you give details as to where details can be found on the 53 litterbag experiments?

    **A:** We suggest to change the first sentence of the caption of Tab. 3 from "Training and testing RMSE for decomposition rates as predicted by different versions of the Holocene Peatland Model …" to "Training and testing RMSE for decomposition rates as predicted by different versions of the decomposition module of the Holocene Peatland Model …" to avoid this ambiguity.
    Details on the 53 litterbag experiments are available from the original publications

explicitly mentioned and cited in l. 103 to 105 (old version) of our manuscript. Moreover, we suggest to include an additional table in the Methods section that lists basic information on the litterbag experiments available for the different *Sphagnum* species (see our reply to comment 24 of reviewer 1).

26. **Q:** Figure 1: 0.3 -0.4 values seem high. Could they be related to S. angustifolium or other species that are seen in a wide range of habitats and might not always be typical of hummocks?

    **A:** As described in section 2.2.2 of our manuscript, we agree that the assignment of individual *Sphagnum* species to *Sphagnum* PFT as defined in the HPM is ambiguous. Future studies should ideally define these PFT and how to assign specific samples to these PFT based on values for the parameter estimates that distinguish the PFT (i.e., $k_{0,i}$, but also all parameters that control the net primary production, see Frolking et al. (2010)).
    We are not sure why the decomposition rates estimated for many *S. angustifolium* are rather large and whether this is a generalizable pattern. However, we want to emphasize that these estimates also depend on the estimated magnitude of initial leaching losses and how one interprets $\alpha$ in equation (2) (updated version) (see also Teickner et al. (2024a)). We state that our test is uncertain here and that errors in $k_{0,i}$ estimates are rather large and we think more targeted experiments are required to address such specific questions (section 4.6 — sections 4.3 and 4.4 in the updated manuscript).

27. **Q:** L289: Figure 2, water table depth are more than 25 cm above litterbag in some cases. Could you please explain how it can be the case?

    **A:** The x-axis in Fig. 2 is the depth of the water table below the litterbag. Negative values occur whenever the litterbag was buried below the average annual WTD (as estimated from the information given in the studies). Thus, for some studies some litterbags were buried more than 25 cm below the average annual WTD. We suggest to include the following note in the caption to clarify this:

    "(negative values represent litterbags placed below the water table, positive values represent litterbags placed above the water table in the unsaturated zone)"

28. **Q:** L350: Was this done within this study? If so, could you please add details in the methods section?

    **A:** Yes, we did these computations within this study. The computations are predictions of decomposition rates with different versions of the modified HPM decomposition module under different conditions. We suggest to move the parts of this section appropriate for the Methods section to the Methods section, as suggested.

29. **Q:** L396: Cannot be estimated: do you mean in this study?

    **A:** The sentence ending in l. 396 is: "A difficulty is that available litterbag experiments cover only a comparatively small depth below the water table level (at most around 30 cm, Fig. 2) and therefore gradients in anaerobic decomposition rates across larger

depths below the water table currently cannot be estimated."
Yes, we wanted to say that because available litterbag data comprise litterbags buried at most around 30 cm below the average annual WTD, it is currently not possible to estimate along gradients in anaerobic decomposition rates covering larger depths below the water table.
To avoid this ambiguity, we suggest to change "… currently cannot be estimated." to "… currently cannot be estimated with available litterbag data."

30. **Q:** L407 and L443: please refer to more recent versions of HPM

    **A:** We thank the reviewer for pointing out that we should mention here recent extensions of the HPM.
    We suggest to add at then end of l. 426: "It is also worth mentioning that a modification of the HPM, HPM-Arctic (Treat et al., 2021), has a seasonally dynamic WTD and this modification may account for at least a part of the discrepancies we observed here. Unfortunately, most available litterbag data do not report WTD at sufficient temporal resolution to test whether standard HPM parameter values are more compatible with litterbag data when such seasonal variations in WTD are considered."
    One reason why we did not consider basing our analysis on HPM-Arctic is therefore that available litterbag experiments do not report WTD at sufficient temporal resolution to test whether standard HPM parameter values are more compatible with litterbag data when such changes in WTD are considered. Additional reasons are discussed in our reply to the following comment.

31. **Q:** L443: HPM was first published (Frolking et al. 2010) with an annual water balance calculation. This has been modified to a monthly water balance calculation a few years later, e.g. Treat's HPM-Arctic . Please consider looking at the latest available HPM code to ensure your conclusions are in line with the current state of development of the model.

    **A:** In addition to the reason mentioned in our reply to the previous comment, we here focus on the HPM version described in Frolking et al. (2010) for two reasons: First, the HPM version described in Frolking et al. (2010) is better analyzed than HPM-Arctic. In particular, the two sensitivity analyses from Quillet et al. (2013a) and Quillet et al. (2013b) allow us to derive consequences for peat accumulation implied by the dscrepancies in parameter values of the decomposition module we identified here, whereas, to our knowledge, no such sensitivity analyses exist for HPM-Arctic. Second, whereas HPM-Arctic makes crucial modifications to describe peat accumulation under permafrost conditions, we do not think that there is sufficient evidence indicating that this model version generally makes more correct predictions.
    We therefore think that practical limitations that make it impossible to sensibly test relevant innovations in HPM-Arctic and less information available on how exactly the modifications in HPM-Arctic affect the sensitivity of peat accumulation make a test of the HPM version described in Frolking et al. (2010) a useful contribution. That said, we suggest that tests of decomposition modules of other peatland models would also be useful to compare and improve these models, including HPM-Arctic.
    As mentioned in our reply to comment 8 of reviewer 1, we suggest to include a new

subsection in the Discussion section, where we list limitaitions of our test and suggest how it can be improved. We suggest to mention there that litterbag data with WTD and degree of saturation reported at sufficient temporal resolution are needed to analyze how seasonal variations in these variables would change estimates for the HPM decomposition module parameters.

32. **Q:** L428: For each site-specific simulation, c2 is adjusted to better represent site-specific conditions. Which value of c2 do you refer to when making this statement?

    **A:** We think the reviewer may refer to l. 420 and not l. 428, since l. 428 does not mention site-specific conditions. The part starting in l. 420 is: "What we have not considered due to limited data is that $c_2$ can be expected to depend on long-term changes in groundwater flow ... or site-specific differences in hydrology and other factors ... . Therefore, $c_2$ may differ between litterbag studies and our data only indicate that $c_2$ is larger on average, whereas more research is necessary to estimate and understand site-specific controls of $c_2$ and how a change in hydrology controls $c_2$."
    Here, we do not refer to any particular value for $c_2$. What we wanted to say is that $c_2$ is intended to describe a process that is assumed to be different between different sites because it is controlled by site-specific differences in hydrology, availability of terminal electron acceptors, etc. Therefore, it is not assumed that one value for $c_2$ is suitable for all sites, but rather that the value of $c_2$ differs between sites and this is one of the few parameters that is estimatedwith peat core data in studies applying the HPM (e.g. Quillet et al. (2015), Treat et al. (2021), Treat et al. (2022)).
    We are not sure whether this answers the question of the reviewer and would like to hear more detailed recommendations if this is not the case.

33. **Q:** L467: How does S12 support this statement?

    **A:** We think that the reviewer confused "supporting Fig. S12" (as stated in the manuscript) with supporting *section* S12 which is part of our supporting information, but not referenced here.

34. **Q:** L477: As mentioned above S. angustifolium might not be appropriately classified here. However, this brings to light that a more detailed classification could be useful to avoid misinterpretation of the HPM PFTs.

    **A:** We agree with the reviewer here and we discuss this issue in detail in section 4.6 (section 4.4 in the updated version of the manuscript). Since we estimate $k_{0,i}$ for each species separately in the other modifications of the HPM decomposition module and our discussion focuses on species-independent discrepancies implied by the litterbag data, this possible misclassification would not have an effect on our conclusions, considering the other limitations we identified, even if we changed our procedure to assign *Sphagnum* species to the HPM PFT.

35. **Q:** L494: Would some of the results not be beneficial to other users or help enhance wider knowledge?

    **A:** We are not sure what the reviewer would consider as "beenficial to other users" or as "enhance wider knowledge". Perhaps the reviewer expects concrete recommendations

such as that future studies should definitely use the parameter estimates we suggest. As explained in our reply to comment 11 of reviewer 1, we do not think that such specific recommendations can be derived from our analysis and instead we suggest that the knowledge gaps for improved tests of the HPM decomposition module are an important contribution that is of practical relevance of users of the HPM and for improvements of the HPM, even if only in the long run.

**1.3 Technical comments**

35. **Q:** L3: missing blank space

    **A:** We thank the reviewer for pointing this out. We will correct this typo in the next version of the manuscript.

36. **Q:** L4-6: Long and complicated sentence, would benefit from being rephrased.

    **A:** The sentence is "Large uncertainties in available litterbag data allow predictions of the HPM to fit decomposition rates estimated from litterbags by adjusting initial leaching losses and decomposition rates estimated from the litterbag data within the range of their uncertainties."
    We agree that this sentence should be rephrased. Also based on comments by the othre reviewers, we suggest a complete re-write of the abstract. Please see the attached manuscript with the suggested changes for how the abstract section would look like.

37. **Q:** L174-182: Some sentences are unclear.

    **A:** We hope that our reply to comments 14 and 22 of reviewer 1 and the additions to the manuscript suggested there address this issue.

38. **Q:** L214: typo that

    **A:** We will remove the redundant "that" as suggested.

39. **Q:** L223: typo from

    **A:** We will correct "form" to "from" as suggested.

40. **Q:** L254: model behaviour of HPM: unclear, could you rephrase?

    **A:** Based on comments by the other reviewers, we suggest a complete re-write of the Results section and we will also change section titles here. Please see the attached manuscript with the suggested changes.

41. **Q:** L370: references seem to be in the wrong place

    **A:** We suggest to change

    "The less pronounced gradient in measured decomposition rates above the water table depth is, however, also visible for *S. fuscum* replicates within the same study (Johnson and Damman, 1991; Golovatskaya and Nikonova, 2017; Mäkilä et al., 2018) and in addition similar across these (independent) studies (supporting information S8),

indicating that this pattern cannot be explained in all cases by differences between studies."

to

"The less pronounced gradient in measured decomposition rates above the water table depth is, however, also visible for *S. fuscum* replicates within the same study and in addition similar across these (independent) studies (Fig. 4, supporting information S8) (Johnson and Damman, 1991; Golovatskaya and Nikonova, 2017; Mäkilä et al., 2018), indicating that this pattern cannot be explained in all cases by differences between studies."

42. **Q:** L397: Figure7, confidence interval

    **A:** We will change the legend label as suggested.

43. **Q:** L504: The last sentence is unclear. Could you please rephrase it?

    **A:** The sentence is "Future litterbag experiments should improve the accuracy of initial leaching loss and decomposition rate estimates and then test whether the identified parameter discrepancies are reproducible and whether they can be described by known, but not yet fully quantified, controls of decomposition rates in dependency of water table fluctuations."
    We suggest to completely re-write the Conclusions section and hope that, by simplifying some sentences, all parts of this section are now understandable. Please see the attached manuscript with the suggested changes.

44. **Q:** S1: model equations: are they not rather sample distributions?

    **A:** We think that "model equations" comprises all mathmeatical formulas required to describe a mathematical model and this also comprises probability distributions for parameters. The model formulas contain both prior and sampling distributions (likelihood) and also mathematical equations and we therefore think that "model equations" is a good summary for the presented information.
    Please note that we suggest to remove supporting section S1 because we now will include all model formulas in the Methods section of the main text. Please see the attached manuscript with the suggested changes.

**2   Additional changes**

1. We suggest a complete re-write of large parts of the manuscript to address the reviewer comments. Specific aspects of this re-write are listed in the comments of the reviewers, others are too numerous for a list of them to be useful without knowing the context of these changes. Please see the attached manuscript with the suggested changes.

2. We suggest to include Quillet et al. (2015) as reference for studies estimating $c_2$ from peat cores. We suggest the following changes:

   We suggest to change l. 392 to 393 from

"Larger and smaller $c_2$ than the standard value have been estimated for several permafrost peatland cores with a modified version of the HPM with monthly time step (Treat et al., 2021, 2022)."

to

"Larger and smaller $c_2$ than the standard value have been estimated for several peatland cores with the HPM and a modified version with monthly time step (Quillet et al., 2015; Treat et al., 2021, 2022)."

3. Frolking et al. (2010) also mention that peat accumulation as predicted by the HPM is sensitive to $c_2$ and a site-specific parameter. We therefore add Frolking et al. (2010) as reference at l. 47 and 422.

4. In l. 77 we will correct "decmposition" to "decomposition".

**References**

[revised manuscript text omitted]

---

## Author Comment (AC2)

**Reply to Anonymous Referee #2 for "Underestimation of Anaerobic Decomposition Rates in Sphagnum Litterbag Experiments by the Holocene Peatland Model Depends on Initial Leaching Losses"**

Henning Teickner[1,2,*]    Edzer Pebesma[2]    Klaus-Holger Knorr[1]

22 December, 2024

[1] ILÖK, Ecohydrology & Biogeochemistry Group, Institute of Landscape Ecology, University of Münster, 48149, Germany
[2] IfGI, Spatiotemporal Modelling Lab, Institute for Geoinformatics, University of Münster, 48149, Germany

[*] Correspondence: Henning Teickner <henning.teickner@uni-muenster.de>

Comments made by the reviewer start with a bold **Q** while our reply starts with a bold **A**. In section "Additional changes" we list additional changes we would like to incorporate in an updated version of the manuscript.

**1   Reply to comments**

1. **Q:** This manuscript describes a study in which Sphagnum moss litterbag data was used to evaluate whether the decomposition module of the Holocene Peatland Model (HPM) can predict correctly Sphagnum litter decomposition rates. In between the experimental data and HPM, there was a litterbag model (hereafter I'll use my own acronym LBM) that was used to estimate decomposition rates from the data. The LBM is described by the authors in another manuscript that is also under review in Egusphere.

   Several different versions of the model test set-up were used. Firstly, decomposition rates from HPM ran with the standard parameter values were compared against decomposition rates from standard LBM. In addition to this, the decomposition rates were estimated with Bayesian statistics by letting different parameter sets of HPM vary. The authors found that the standard HPM underestimates anaerobic decomposition rates for many moss species and predicts too steep decrease of decomposition rates from oxic to anoxic conditions.

The topic is useful and relevant, and the authors seem to have done lots of work, but the methods and materials should be described in more detail and clarifications are needed in the manuscript. In my opinion, in its present form, it is difficult to follow the text, and some information is missing - although I have to admit I'm not especially familiar with the Bayesian methods. In particular, I find the role of the LBM and the Bayesian modelling unclear. Below are my specific comments and questions. I focused on the Methods section since I think it is important to modify it first.

**A:** We thank the reviewer for their helpful comments. We agree that we should have described in more detail which equations of the HPM are considered as HPM decomposition module, how we modified them, and how we linked them to the LBM. We hope that our changes suggested below address the issues. In particular, we suggest an extensive re-write of all sections of the manuscript to address the reviewer comments and have attached an updated version to show how this manuscript would look like.

2. **Q:** The abstract text does not mention the LBM (=the litterbag model) at all

    **A:** We thank the reviewer for this suggestion. We suggest to change the second sentence of the abstract (l. 2 to 3) from

    "Here, we test whether the HPM decomposition module can predict decomposition of available *Sphagnum* litterbag data along a gradient from oxic to anoxic conditions and estimate parameter values from the litterbag data."

    to

    "Here, we estimate parameter values of the HPM decomposition module from available *Sphagnum* litterbag experiments included in the Peatland Decomosition Database and with a litterbag decomposition model that considers initial leaching losses. Using either these estimates or the standard parameter values, we test whether the HPM decomposition module fits decomposition rates ($k_0$) in *Sphagnum* litterbag experiments along a gradient from oxic to anoxic conditions."

    Please also see our reply to comment 8 of reviewer 2 for more information about the purpose of the LBM in our study here.

3. **Q:** l. 3: Please correct "conditionsand"

    **A:** We will correct this typo as suggested.

4. **Q:** l. 13-14: Is this a correct sentence?

    **A:** We think that this is a correct sentence, but we re-wrote the entire abstract section to consider the comments of all reviewers and this part of the abstract was changed completely and should be more readable now.

5. **Q:** l. 29-30: An unclear sentence (do the equations cause observed discrepancies less reliably, or do the tests identify things less reliably?)

    **A:** We agree that this sentence can be improved. We suggest to change it from

    "First, they test entire peatland models against observed data and thus can identify the parameter values or model equations that cause observed discrepancies less reliably."

to

"First, they cannot reliably identify the parameter values or model equations that cause discrepancies between model predictions and measurements because they test entire peatland models against observed data."

6. **Q:** l. 50: Please clarify what you mean with "HPM uses litterbag data". Were the standard parameters derived from litterbag data?

   **A:** Frolking et al. (2010) states that $k_0$ values in the HPM were defined based on litterbag experiments, but the study does not explicitly mention data sources (except Moore et al. (2007)) or modeling (parameter estimation) approaches.
   To avoid ambiguities, we suggest to replace "The HPM uses litterbag data to define average decomposition rates of moss plant functional types …" by "The HPM derives initial decomposition rates of moss plant functional types from litterbag data …".

   Here are the passages from Frolking et al. (2010) that describe how litterbag data were used: "Litter bag decomposition rates are determined by fitting mass loss data [from litterbag experiments] to an exponential function, $m(t) = m_0 \exp(-kt)$ , under the assumption that $k$ is a constant ($\frac{dm}{dt} = -km$); since HPM represents litter/peat decomposition as an initial rate $k_0$, that declines linearly with mass loss (i.e., $\frac{dm}{dt} = -\frac{m}{m_0}k_0 m$), the litter bag $k$-values need to be modified to $k_0$ values. We have modified these so that, after 5 years of decomposition they have approximately the same mass loss (see Table 1)."

   In the footnote to the referenced table, Frolking et al. (2010) states: "$k_0$ are the tissue initial decomposition rates, determined by $k_0 = k(1+3k)$, where $k$ is first order exponential decay rate, i.e., $m(t) = m_0 \exp(-kt)$ (e.g., Moore et al. (2007)), and $k_0$ is the value that gives similar $m(t)$ at $t \sim 5$ years (i.e., near the end of reliable values from a litter bag field decomposition study) when $m(t) = m_0/(1 + k_0 t)$ (e.g. Frolking et al., 2002), the formulation used in HPM., E.g., if $k = 0.15$ y$^{-1}$, $k_0 = 0.22$ y$^{-1}$, $\exp(-5k) = 0.47$; $1/(1 + 5k0) = 0.48$."

7. **Q:** l. 66-68: Are you referring to your own LBM, or are there many models compatible with HPM? Please clarify.

   **A:** The sentence starting in l. 65 is: "Remaining masses in litterbag experiments are often very variable, even under controlled environmental conditions (e.g. Bengtsson et al. (2018)), and for many litterbag experiments, a range of decomposition rates may produce similar predictions for remaining masses if a litterbag decomposition model compatible with the HPM is used (Teickner et al., 2024)."
   A model that is compatible with the HPM is one that uses equation (7) in Frolking et al. (2010) to describe mass losses in dependency of $k_0$. To our knowledge, the formula for remaining masses used in the HPM (equation (7) in Frolking et al. (2010)) has not been used by any litterbag study so far, except our modified version (equation (1) in our manuscript) that we used in Teickner et al. (2024) to estimate decomposition rates and initial leaching losses from *Sphagnum* litterbag experiments.
   That a "range of decomposition rates may produce similar predictions for remaining masses" is due to large errors in estimates for initial leaching losses that can be derived

from available litterbag data and not because of using equation (7) in Frolking et al. (2010) to analyze litterbag data.

Thus, we just wanted to state that even a test of only the decomposition module has uncertainties because initial leaching losses of not well constrained magnitude cause errors in deocmposition rate estimates and this also happens when the litterbag decomposition is analyzed with the equation for mass loss used in the HPM.

To clarify this sentence, we suggest to change it to:

"Remaining masses in litterbag experiments are often very variable, even under controlled environmental conditions (e.g., Bengtsson et al., 2018), and for many litterbag experiments, a range of decomposition rates may produce similar predictions for remaining masses (e.g., Yu et al., 2001), also if a litterbag decomposition model compatible with the HPM is used (Teickner et al., 2024)."

If the reviewer thinks that this does not address the issue, we would be grateful for further suggestions.

8. **Q:** l. 87-88: In my opinion, you need to tell here, and especially in the Methods section, more about your LBM. It remains unclear what kind of it is and what is its purpose. Why is a separate model needed to derive the decomposition rates from the litterbag data?

   **A:** We thank the reviewer for this suggestion and apologize that we did not provide a more detailed description of the purpose of the LBM.

   Our LBM has two purposes: First, as described in our reply to the previous comment by reviewer 2, it makes equation (7) in Frolking et al. (2010) compatible with litterbag experiments by including initial leaching losses; however this can also be seen as a simple modification of the HPM decomposition module and not a separate LBM. Second, there are many litterbag studies that do not provide the necessary data (water table depth) to make predictions with the HPM decomposition module, but they still provide information on initial leaching losses and average decomposition rates for individual species that are useful to constrain decomposition rate and initial leaching loss estimates. Our LBM is a hierarchical model that pools information of these experiments and experiments that provide data to make predictions with the HPM deocmposition module and therefore helps to constrain estimates of the HPM parameters. We agree that it is necessary to make this clear in the introduction, but we now also think that we gave our LBM a too prominent place in the introduction which may be more confusing than helpful at this place.

   We suggest to rewrite the entire paragraph (ll. 87 to 91) from

   "To address these aims, we developed a model that combines the HPM decomposition module and our previous *Sphagnum* litterbag decomposition model, which estimates decomposition rates in available litterbag experiments while considering initial leaching losses (Teickner et al., 2024). Estimated decomposition rates of this model can be directly compared to decomposition rates predicted by the HPM decompsoition module because the formula to compute remaining masses from decomposition rates is the same."

to

"To address these aims, we used the HPM decomposition module to predict decomposition rates in available litterbag experiments and compared these to decomposition rates estimated for the same litterbag experiments with a litterbag decomposition model that considers initial leaching losses (Teickner et al., 2024) (Fig. 1). These predictions require the peat degree of saturation, which we estimate with the modified Granberg model (Granberg et al., 1999; Kettridge and Baird, 2007) from water table depth data reported in these studies. Furthermore, some *Sphagnum* litterbag experiments do not report water table depths and therefore cannot be used to test the HPM, but they still provide information on initial leaching losses and decomposition rates and therefore help to constrain parameter estimates. We therefore include these data via Bayesian hierarchical modeling in the litterbag decomposition model. In summary, our approach combines the HPM decomposition module, the modified Granberg model, and a *Sphagnum* litterbag decomposition model that allows to consider intitial leaching losses and to pool information across litterbag experiments (Teickner et al., 2024). While this approach has its limitations, it exploits available data as far as possible, while considering known confounders and propagating relevant uncertainties."

Please note that "Fig. 1" refers to the new figure suggested by reviewer 2 in comment 12.

9. **Q:** l. 89: Please define what is "initial leaching loss".

   **A:** We agree that it is a good idea to define "initial leahing loss" and to describe its relevance. We define "initial leaching losses" as losses of soluble compounds during the first days to weeks of decomposition that do not originate from microbial depolymerization due to leaching (Teickner et al., 2024). We suggest to introduce "initial leahing loss" already earlier by changing the part starting in l. 59 from

   "Since decomposition rates have been estimated with different litterbag decomposition models in previous studies, their values are not directly comparable and therefore raw data are necessary to obtain estimates directly comparable to predictions from a certain peatland model (Yu et al., 2001; Teickner et al., 2024). Recently, we used available Sphagnum litterbag data to estimate decomposition rates which can be directly compared to decomposition rates predicted by the HPM (Teickner et al., 2024)."

   to

   "Since decomposition rates have been estimated with different litterbag decomposition models in previous studies, their values are not directly comparable. Moreover, initial leaching losses (losses of soluble compounds, which do not originate from microbial depolymerization, due to leaching during the first days to weeks of incubation) can bias decomposition rate estimates if they are not explicitly considered and can vary between species and experiments (Yu et al., 2001; Teickner et al., 2024). Therefore, raw data (remaining masses) are necessary for any meaningful test of decomposition modules with litterbag data. The recently published Peatland Decomposition Database (Teickner and Knorr, 2024) contains raw data from available *Sphagnum* litterbag experiments

and therefore allows to estimate parameters with any mass loss-based decomposition model and therefore also allows to consider initial leaching losses."

10. **Q:** l. 94-95: What do you mean by "Our test": this current work or an earlier study?

    **A:** We mean the current work. To avoid misunderstandings, we suggest to change

    "Our test identified discrepancies between the HPM and litterbag data that could give novel insights into processes controlling anaerobic decomposition rates in future litterbag experiments"

    to

    "Similarly, the parameter discrepancies identified here suggest future litterbag experiments that would provide novel insights into oxic and anoxic controls of *Sphagnum* decomposition rates and our study therefore suggests a strategy to improve decomposition modules in general."

11. **Q:** Table 1: Can you please add here also the other parameters that you are optimizing, alpha and l0. It would make it easier to follow the text, if all the parameters could be checked from here.

    **A:** We suggest not to add $\alpha$ and $l_0$ to Tab. 1 because we did not aim to estimate $\alpha$ from the litterbag data and because $l_0$ is no parameter of the HPM, but a parameter we defined in our previous study (Teickner and Knorr, 2024) (implicitly, the HPM assumes $l_0 = 0$).
    We understand that it may be useful to have a list of all model parameters with their definitions, but we fear that such a list may be cause confusion about which parameters we intended to test. We therefore think that it is more useful after all to only show parameters of the HPM decomposition module we actually tested in Tab. 1.
    We did not try to test $\alpha$ with the litterbag data because previous studies (e.g., Frolking et al. (2001)) suggest that litterbag data are too short and (currently) have too large errors to estimate $\alpha$ correctly and also our previous study (Teickner and Knorr, 2024) suggests this. We estimate $\alpha$ only to propagate an approximate error estimate for this parameter.
    Please note that the models include many more parameters than shown in Tab. 1. These parameters are now defined throughout the text and will be listed in supporting Tab. S1 in the updated version of the supporting information.

12. **Q:** It would be great to have a schematic picture that illustrates the relationship between the different models and data that you use. If possible, it could include some more general information of the HPM. There apparently are also other parts in the model, as you focus only on a small part of it (the decomposition module). Are there other PFT's than Sphagnum?

    **A:** We think that this is a good idea and we suggest to include the picture below as new Fig. 1 in our manuscript. Yes, the HPM includes additional PFT: brown mosses, feather mosses, minerotrophic and ombrotrophic sedges, shrubs, and forbs, but the Peatland Decomposition Database currently focuses on *Sphagnum.*

Figure 1: Conceptual representation of the modeling approach. Arrows represent flows of information. Litterbag data that have information on water table depths (WTD) and incubation depths are used to estimate average decomposition rates with the HPM decomposition module ($\mu_k$). The HPM decomposition module needs plant functional type identity, peat degree of saturation, WTD, and incubation depth to predict decomposition rates. The modified Granberg model is used to estimate peat degree of saturation at incubation depths from WTD, minimum water content at the surface, and porosity, of which the latter two are estimated from the remaining masses. The litterbag decomposition model is used to estimate decomposition rates ($k_0$) for all litterbag studies, including those that have information on WTD and those that have not. A gamma distribution with $\mu_k$ as average is used as prior distribution for $k_0$ for the litterbag experiments that have information on WTD (curved arrow). This helps to constrain initial leaching loss and decomposition rate estimates for studies that can be predicted with the HPM decomposition module. The Litterbag decomposition module also estimates initial leaching losses ($l_0$) for all litterbag experiments. The equation at the bottom uses these to estimate remaining masses in the litterbag experiments. The litterbag decomposition model is described in more detail in section 2.2.1. See the text for further details.

13. **Q:** Please add a section describing the LBM, there's no information about it now.

    **A:** We suggest to modify section 2.2.2. to provide a more detailed description of our modeling approach. Please see the attached version of the manuscript to see how the text would look like with the suggested changes.

14. **Q:** l. 99-105: Please divide this sentence into smaller pieces, it is difficult to read. I do not completely understand what you did and for what you used only the data of the latter reference list. Did you compare the results to the same data into which you had fitted the models?

    **A:** We suggest to split the sentence as follows:

    "To test the HPM decomposition module against litterbag data, we used the Peatland Decomposition Database (Teickner and Knorr, 2024). In this study, we use data from Bartsch and Moore (1985), Vitt (1990), Johnson and Damman (1991), Szumigalski and Bayley (1996), Prevost et al. (1997), Scheffer et al. (2001), Thormann et al. (2001), Asada and Warner (2005), Trinder et al. (2008), Breeuwer et al. (2008), Straková et al. (2010), Hagemann and Moroni (2015), Golovatskaya and Nikonova (2017), and Mäkilä et al. (2018) to estimate $k_0$ using the litterbag decomposition model. Data from Johnson and Damman (1991), Szumigalski and Bayley (1996), Prevost et al. (1997), Straková et al. (2010), Golovatskaya and Nikonova (2017), and Mäkilä et al. (2018) reported water table depths and therefore only these data were used to predict $k_0$ also with the HPM decomposition module."

    We hope that the conceptual figure (Fig. 1) and our reply to comment 13 of reviewer 2 clarify how we used the litterbag data in our modeling approach.

15. **Q:** l. 109-111: Is this now the same list of input as in the Introduction l. 38-39? Porosity is missing here. Also, is "the fraction of mass already lost" the same as "initial leaching loss"?

    **A:** Yes, this is supposed to be the same list as in the Introduction l(l. 38-39). The reason why we did not mention peat porosity here is that the degree of saturation can be computed from the water content (mentioned in ll. 38-39) and peat porosity and the HPM decomposition module only needs the degree of saturation to make predictions. We also missed "the depth of the litter below the peat surface" in the list in the Introduction. To avoid confusion, we suggest to change ll. 38-39 from:

    "For example, in the Holocene Peatland Model (HPM) (Frolking et al., 2010), we only need to know litter species, peat water content, peat porosity, water table depth, and only five parameters to predict decomposition rates."

    to

    "For example, in the Holocene Peatland Model (HPM) (Frolking et al., 2010), we only need to know litter species, peat degree of saturation, the depth of the litter below the peat surface, water table depth, and only five parameters to predict decomposition rates."

    "the fraction of mass already lost" is one minus the fraction of initial mass remaining $(1 - \frac{m(t)}{m_0})$, i.e., the fraction of the mass of the litterbag before the incubation that has

been lost during the incubation. It is not the "initial leaching loss". The HPM assumes that the decomposition rate slows down as mass gets lost during the decomposition. Equation (2) (in the attached version of the manuscript) already considers this slow down of the decomposition rate, but assumes that the litter has not been decomposed yet. If this is not the case, $k_0$ needs to be adjusted for the fraction of initial mass already lost, as described in equations (3) and (6) in Frolking et al. (2001). We mention this here because samples from Prevost et al. (1997) are already decomposed *Sphagnum* samples. However, the fraction of initial mass already lost is not known which is why we assign them to dummy PFT and therefore estimate their initial decomposition rate ($k_{0,i}$) separately to account for this unknown fraction of initial mass lost.

16. **Q:** l. 112-115: Please divide this sentence into two, it's slightly complicated.

    **A:** We thank the reviewer for this suggestion. We suggest to change the sentence from

    "Predicting decomposition rates for the available litterbag data is not straightforward because the HPM decomposition module does not consider specific features of litterbag experiments, because it does not specify how to assign species to plant functional types, and because required variables such as the degree of saturation are not reported in the litterbag studies and therefore need to be estimated."

    to

    "Predicting decomposition rates for the available litterbag data is not straightforward because the HPM decomposition module does not consider specific features of available litterbag experiments. The HPM does not specify how to assign species to plant functional types. Moreover, none of the available litterbag studies reported the degree of saturation which therefore needs to be estimated in order to make predictions with the HPM decomposition module."

17. **Q:** l. 115-116: The logic of this sentence is not clear. Do you mean that one can't directly use litterbag data as a comparison for the HPM, because HPM doesn't predict masses but decomposition rates?

    **A:** We agree that this sentence is confusing at this place and therefore suggest to remove it. We hope that the new Fig. 1 and our reply to comment 13 of reviewer 2 explain how we link litterbag data to the HPM decomposition module.

18. **Q:** Section 2.2.2: Please report how many replicates/data points there were for each species, niches, etc. It is relevant information but not clearly mentioned anywhere.

    **A:** We agree that this is useful information. We suggest to include a new table at the end of section 2.1. and make appropriate references throughout the text. The table is:

Table 1: Overview on litterbag experiments included for each *Sphagnum* taxon in this study. "HPM microhabitat" is the HPM microhabitat assigned to each taxon. Taxa without value are not considered in Johnson et al. (2015) (see section 2.2.2). "Number of experiments" is the number of litterbag experiments available from the Peatland Decomposition Database (these are either individual replicates or average values of replicates, depending on what data were reported in the studies). "Number of experiments with WTD data" is the number of litterbag experiments that also report water table depths and for which we therefore could make predictions with the HPM decomposition module. "Depth range" are the maximum and minimum depth below the peat surface at which litterbags were placed [cm]. Missing values mean that no study reported depths.

| Taxon | HPM microhabitat | Number of studies | Number of experiments | Number of experiments with WTD data | Depth range |
|---|---|---|---|---|---|
| *Sphagnum* spec. | | 2 | 16 | 10 | 10, 30 |
| *S. angustifolium* | Hummock | 4 | 14 | 8 | 1, 30 |
| *S. auriculatum* | | 1 | 3 | 0 | 0, 6 |
| *S. balticum* | Lawn | 3 | 12 | 3 | 1, 30 |
| *S. cuspidatum* | Hollow | 1 | 5 | 5 | 10, 50 |
| *S. fallax* | Lawn | 1 | 4 | 1 | 1, 1 |
| *S. fuscum* | Hummock | 9 | 32 | 13 | 1, 50 |
| *S. lindbergii* | Lawn | 1 | 2 | 0 | |
| *S. magellanicum aggr.* | Hummock | 3 | 7 | 5 | 1, 50 |
| *S. majus* | Hollow | 1 | 2 | 2 | 10, 30 |
| *S. papillosum* | Lawn | 2 | 6 | 1 | 0, 1 |
| *S. rubellum* | Hummock | 1 | 2 | 2 | 10, 30 |
| *S. russowii* | Hummock | 1 | 3 | 2 | 1, 1 |
| *S. russowii* and *capillifolium* | | 1 | 18 | 0 | 5, 5 |
| *S. squarrosum* | Lawn | 1 | 2 | 0 | 0, 0 |
| *S. teres* | Lawn | 1 | 1 | 1 | 2, 2 |

20. **Q:** l. 145: Why did you estimate ki,0 only in these two set-ups?

   **A:** The reviewer is correct that we omitted model HPMe-LE-peat-l0-outlier (new model name: HPM-outlier) from this list. We suggest to correct this. For all other model variants, all HPM decomposition module parameters were kept constant and we therefore did not estimate $k_{0,i}$ for them.

21. **Q:** Section 2.2.3: Where was this information (degree of saturation) needed, was it input for the HPM?

   **A:** Yes, the degree of saturation is an input of the HPM deocmposition module. We hope that our new Fig. 1 (see our reply to comment 12 of reviewer 2) and our reply to comment 13 of reviewer 2 address this issue.

22. **Q:** l. 154: Does this mean you set the mass loss to zero in the model?

   **A:** This means that $m_0 = 1$ (equation (2) (attached manuscript)), i.e. all of the mass when the *Sphagnum* plants died is still available at the start of the incubation. To avoid misunderstandings, we suggest to modify ll. 153 to 155 from

   "All litterbag data we use here, except samples from Prevost et al. (1997), are from *Sphagnum* samples collected from the surface of peatlands and therefore can be expected to have not experienced mass loss due to decomposition at the start of the experiments."

   to

   "All litterbag data we use here, except samples from Prevost et al. (1997), are from *Sphagnum* samples collected from the surface of peatlands and therefore can be expected to have not experienced mass loss due to decomposition at the start of the experiments $(m(t = 0) = 1$ in equation (2))."

23. **Q:** l. 163: "Minimum water content at the surface" appears here for the first time. Is it an additional input parameter for HPM?

   **A:** We thank the reviewer for this question. We forgot to mention that the minimum water content at the surface is an input required by the modified Granberg model. We suggest to add this in section 2.2.3. The section then reads (with other additions due to comment 27 of reviewer 3):

   "We estimated the degree of saturation with the modified Granberg model (ModGberg model) (Granberg et al., 1999; Kettridge and Baird, 2007) from minimum water content at the surface $(\theta_{0,\min})$, total porosity $(P)$, the water table depth below the peat surface $(z_{\mathrm{wt}})$, and the positions of the litterbags during the incubation $(z)$:

   $$
   \begin{aligned}
   \theta(z) = \ & \min\left(P, \theta_0 + (P - \theta_0)\left(\frac{z}{z_{\mathrm{wt}}}\right)^2\right) \\
   \theta_0 = \ & \max\left(\theta_{0,\min}, 0.15 z_{\mathrm{wt}}^{-0.28}\right),
   \end{aligned}
   \tag{1}
   $$

   where $\theta_0$ is the water content at the surface and $0.15 z_{\mathrm{wt}}^{-0.28}$ is an empirical relation of $\theta_0$ with the WTD (Kettridge and Baird, 2007).

The minimum water content at the surface was not reported in any study and we therefore assumed a minimum water content at the surface of 0.05 $\mathrm{L}_{\mathrm{water}}\,\mathrm{L}_{\mathrm{sample}}^{-1}$ with a standard deviation of 0.05 $\mathrm{L}_{\mathrm{water}}\,\mathrm{L}_{\mathrm{sample}}^{-1}$, based on measurements from Hayward and Clymo (1982). The total porosity was not reported in any study and therefore we assumed an average value of 80% with a standard deviation of 10%, roughly based on values reported for low-density *Sphagnum* peat (Liu and Lennartz, 2019). An improved test of the HPM decomposition module would require litterbag experiments with direct measurements of the degree of saturation at sufficient temporal resolution."

24. **Q:** l. 166-168: So how did you determine the litterbag decomposition rates in this case?

    **A:** We hope that our new Fig. 1 (see our reply to comment 12 of reviewer 1) and our reply to comment 13 of reviewer 2 address this issue. Specifically, in this case the standard parameter values of the HPM decomposition module were used to predict decomposition rates ($\mu_k$ in Fig. 1, this reply document), and decomposition rates were estimated from the litterbag experiments using only the litterbag decomposition model. We think that our text may be ambiguous here and we suggest a complete re-write of the Methods section, where we improve our description of the different models. Please see section 2.3.1 in the atached version of the manuscript with the suggested changes.

25. **Q:** l. 169-173: I was trying to think how to name the model versions so that one doesn't need to always check the differences from the table. Perhaps it would help if you explained here why you named the models like this. E.g. where does the LE part come from?

    **A:** We thank the reviewer for this suggestion. We suggest to change all model names as suggested in comment 7 of reviewer 3 and also update Tab. 2 to clarify how the models differ. In addition, we introduce the name "LDM-standard" for only the litterbag decomposition model, without the HPM decomposition module as prior. Please see the attached manuscript with the suggested changes.

26. **Q:** l. 172: Please show the formulas you used for l0.

    **A:** We suggest to add after l. 192:

    "Specifically, we use the following logistic regression model to describe an average initial leaching loss per sample, in dependency of the degree of saturation:

$$
\begin{aligned}
\mu_l =&\ \ \mathrm{logit}^{-1}(\beta_{l,1} + \beta_{l,2}W)\\
l_0 \sim&\ \ \mathrm{beta}(\mu_l\phi_l, (1-\mu_l)\phi_l),
\end{aligned}
\tag{2}
$$

    where $\mu_l$ is the average initial leaching loss for a sample, $\beta_{l,1}$ is the (hypothetical) average initial leaching loss at a degree of saturation 0 for each taxon, $\beta_{l,2}$ is the coefficient that describes the relation to the degree of saturation ($W$), and $\phi_l$ transforms $\mu_l$ and $(1-\mu_l)$ into the shape and rate parameters of a beta distribution. This beta distribution has the same function as the gamma distribution (equation 8) for $k_0$ (compare also with Fig. 1): it is a prior for $l_0$ estimated with the litterbag decomposition model, where the average of this prior is $\mu_l$.

27. **Q:** l. 174-178: Please explain this part more thoroughly. What did the Bayesian model do exactly? I assume you optimized different parameter sets of HPM, using the default parameters of HPM as priors. But what was the role of the LBM here? Did you simultaneously optimize parameters of the LBM, and if yes, which parameters were they? Why did you assume that the default parameters of HPM are better priors than something based on the litterbag data? Could you also have used some standard parameters of the LBM as the starting point?

**A:** We apologize that we have omitted these information. We hope that our new Fig. 1 (see our reply to comment 12 of reviewer 1) and our reply to comment 13 of reviewer 2 address all aspects of this comment.
To help clarify possible misunderstandings, here are brief replies to the questions:

- "Did you simultaneously optimize parameters of the LBM, and if yes, which parameters were they?" Yes, the LBM estimates $k_0$, $l_0$, and $\alpha$ for each litterbag experiment. The HPM decomposition module estimates $\mu_k$ which is assumed to be the average of the gamma distribution that also estimates $k_0$ (equation (8), in the updated section 2.2.1).

- "Why did you assume that the default parameters of HPM are better priors than something based on the litterbag data? Could you also have used some standard parameters of the LBM as the starting point?" In a Bayesian data analysis, you have to assume a prior to estimate a parameter. There are no prior estimates for the HPM decomposition module parameters based on litterbag data, except for the standard parameter values. We therefore constructed our priors based on these standard values.

28. **Q:** l. 180: How did you obtain the uncertainties?

**A:** The referenced sentence is: "If HPM parameter estimates from the combined model are not compatible with standard values used in the original model (Tab. 1) even if we adjust them to the HPM within the range allowed by the uncertainties, this is a discrepancy worth testing in future experiments."
The uncertainties we mean here are the uncertainties of the HPM decomposition module parameters implied by our priors and the litterbag data, as represented by the joint posterior distribution for each model. To avoid misunderstandings, we suggest to change this sentence to:

"If HPM decomposition module parameter estimates from the combined model are different from the standard values used in the original model (Tab. 1), even if we consider these uncertainties and use the HPM decomposition module as prior for the litterbag data, this is a discrepancy worth testing in future experiments."

29. **Q:** l. 182-183: How do you estimate the uncertain peat properties? Do you mean you assumed the decomposition rates from the LBM are correct and the HPM parameters are correct, and then analyzed what kind of peat properties were needed to achieve these?

**A:** By Bayes theorem, the posterior probability distribution of some parameter $\theta$ conditional on some data $y$ is proportional to the product of the likelihood and the prior distribution: $p(\theta|y) \propto p(y|\theta)p(\theta)$.

In this case, the data are the measured remaining masses in the litterbag experiments. This means that if we put a prior on a variable in a model, it can be estimated from the data conditional on the prior and the model. Not in all cases are the data informative with respect to a parameter, and, as expected, the marginal posterior distribution of the peat properties is dominated by the prior in our case (as mentioned in supporting information S3 (supporting information S2 in the updated version of the supporting information)).

If the reviewer thinks this should be mentioned more explicitly in the text, we would be grateful for further comments how we could do this most appropriately.

30. **Q:** l. 184: Only parameters or also the peat properties?

    **A:** Model HPMe-LE-peat (now HPM-all) estimates peat properties and HPM decomposition module parameters. We hope that our reply to comment 25 of reviewer 2 addresses this issue.

31. **Q:** l. 189-190: I would have thought this was vice versa: weren't the leaching losses estimated more independently in the l0 version?

    **A:** We apologize that we may have caused confusion by not fully explaining our modeling approach. In the $l_0$ version, we introduce an additional model for initial leaching losses in dependency of the degree of saturation (see our reply to comment 26 of reviewer 2) and this potentially constrains $l_0$ estimates because it is an additional prior. Without this model, $l_0$ is only estimated by the LBM which is simply estimates $l_0$ from each individual litterbag experiment without assuming any specific relation to the degree of saturation. We hope that our reply to comment 26 of reviewer 2 also addresses this question.

32. **Q:** l. 200-201: The message of this sentence is unclear.

    **A:** The sentence is "Results for HPMe-LE-peat-l0-outlier are shown in supporting information S10 and HPM parameter estimates agree with the other models where HPM parameters were estimated."

    To avoid misunderstandings, we suggest to change this sentence to: "Results for HPM-outlier are shown in supporting information S9 and HPM decomposition module parameter estimates agree with estimates of HPM-leaching and HPM-all."

    We would be thankful if the reviewer lets us know whether this clarifies the sentence.

33. **Q:** l. 202-203: Is it so that the Granberg model was not included in the Bayesian system, it had some fixed parameters? Is the minimum peat water content at the surface is from Granberg?

    **A:** Yes, this is correct. We did estimate only the minimum water content at the surface to keep the model computationally manageable. We hope that we emphasized sufficiently throughout the manuscript that not having measurements for the degree

of saturation is a limitation and that the modified Granberg model is only an approximation we use since such data are not available.

Our prior for the minimum water content at the surface was defined as described in our reply to comment 23 of reviewer 2.

In the updated version of the manuscript, we suggest to include a detailed description of the modified Granberg model (see section 2.2.3 in the attached version of the manuscript with the suggested changes).

34. **Q:** l. 209-210: Again I'm a little bit lost with what the Bayesian model did. Can you please explain if the estimates from HPM and LBM were somehow optimized together, or if they were separate. I suppose that estimation of the decomposition rates from the litterbag data was done with the LBM?

    **A:** We hope that our new Fig. 1 (see our reply to comment 12 of reviewer 1) and our reply to comment 13 of reviewer 2 addresses this comment. We also suggest to completely re-write the referenced part of the Methods section and hope that it is more understandable now.

35. **Q:** l. 211: Should it be "large probability indicates…"?

    **A:** We thank the reviewer for reporting this error. We suggest to correct it as suggested.

36. **Q:** l. 218-219: Please define what is a one-pool decomposition rate? What pool?

    **A:** We suggest to remove "one-pool". We mean $k_0$ here.

37. **Q:** l. 224-225: Is it possible to re-formulate this. E.g. it might be clearer to say that in the folds, you included only data of those species for which there was data from several sites.

    **A:** We agree that this sentence should be improved. We suggest to change

    "Each fold consists of the data from one study, except those values that were measured for *Sphagnum* species for which only this study had data (we want to estimate the predictive accuracy not for new species). Data for species with data from one study only were always used for model training and not part of the testing folds."

    to

    "Each fold represents the data from one study, but only if there were still data for the same *Sphagnum* species left in the remaining data (we want to estimate the predictive accuracy not for new species). Species with data from one study only were always used for model training and not part of the testing folds."

38. **Q:** l. 232: This kind of prior knowledge sounds relevant - were there many this kind of restrictions? Perhaps they need to be listed.

    **A:** In Bayesian data analysis one always includes some sort of prior knowledge. All prior distributions are listed in supporting Tab. S1, as mentioned at the end of the same paragraph (l. 238).

39. **Q:** l. 239: Where was this priorsense package?

    **A:** Unfortunately, we do not understand this question.

40. **Q:** l. 250 and Table 3: To me it looks like the RMSE_test is smaller for HPMe-LE-peat_l0.

    **A:** The average $\text{RMSE}_{\text{test}}$ is smaller for HPMe-LE-peat-l0, but our estimates have large errors (as in indicated in Tab. 3) and therefore one can neither conclude that HPM-leaching makes better, nor that it makes worse predictions. We currently think that the confidence intervals given in Tab. 3 make this clear.

41. **Q:** l. 257: Didn't you also adjust the peat parameters in HPMf-LE-peat?

    **A:** Yes, this is described in Tab. 2. However, the peat properties are no parameters of the HPM decomposition module. This confusion was probably caused by our omission of details of our modeling approach. We hope that our new Fig. 1 (see our reply to comment 12 of reviewer 1) and our reply to comment 13 of reviewer 2 address this comment.

42. **Q:** l. 263-264: Where are the estimates of initial leaching losses needed?

    **A:** As now shown in our new Fig. 1 and in equation (2) (in the new version of the manuscript), $l_0$ is needed to predict remaining masses in the litterbag experiment. Our point here is that mass losses in litterbag experiments can be explained both by initial leaching losses and by decomposition, but there is uncertainty about the relative magnitude of both processes (because the design of available litterbag experiments is not ideal, see Teickner et al. (2024)). For this reason, the model can — within the range of uncertainty implied by the posterior distribution — adjust the decomposition rate so that it agrees with the HPM decomposition module and then change initial leaching losses accordingly to fit the remaining mass data from the litterbag experiments.

    To clarify this point, we competely re-wrote the referenced part of the Results section. Please see the attached manuscript for how the suggested changes would look like (sections 3.1 and 3.2).

43. **Q:** l. 271-272: Could this result be affected by the priors? Also, to me it seems (Fig. 2) that especially for S. angustifolium, these estimates are very similar - but perhaps I misunderstood the plot.

    **A:** The referenced sentences (ll. 269 to 272) are: "HPMe-LE-peat estimated larger initial leaching losses and smaller decomposition rates than the litterbag decomposition model from Teickner et al. (2024) alone, similar to HPMf-LE-peat (Fig. 3). This is particularly the case for *S. angustifolium*, for which the separate litterbag decomposition model estimated much larger average decomposition rates and smaller initial leaching losses than the litterbag decomposition model in HPMe-LE-peat (Fig. 2)."
    We think that this is a misunderstanding. We state that average decomposition rate estimats for *S. angustifolium* by the LBM alone (without using the HPM decomposition module as prior) are larger than when the HPM decomposition module is used as prior for $k_0$. Thus, we here refer to the black line in the first column in Fig. 2

compared to the grey lines in the first three columns for *S. angustifolium.* The black line in the first column clearly is above the grey lines, indicating that when the HPM decomposition module is used as prior for $k_0$, $k_0$ estimates are smaller.

We now think that the figure may be more confusing than helpful. We therefore suggest to remove it and replace it by a better presentation of our results and two new figures that more appropriately illustrate the differences between HPM-standard and LDM-standard on the one hand and between HPM-standard and the modifications of the HPM decomposition module on the other hand. Please see section 3.4 in the attached version of the manuscript with the suggested changes.

Regarding "Could this result be affected by the priors?": In the companion manuscript, Teickner et al. (2024), we analyzed whether initial leaching losses and decomposition rates of different magnitude in simulated data can be recovered by the LBM and this was the case. Therefore, priors for parameters of the LBM are unlikely to strongly bias our results, but of course, they can have some influence, which is why we suggest future experiments to test our hypotheses.

As mentioned above, when the HPM is used as prior, estimates change and therefore, the smaller $k_0$ and larger $l_0$ estimates compared to the LBM alone are caused by the HPM decomposition module.

44. **Q:** l. 258: Relations of l0 and what?

    **A:** We think the reviewer refers to l. 286, where we omitted to mention the degree of saturation, and not l. 258. We thank the reviewer for this suggestion and correct the sentence as suggested.

45. **Q:** Figure 2: I don't understand why the "HPM=No" values are different for different model versions, if they were not predicted by the different versions of HPM but just estimated from the litterbag data. What are the error bars?

    **A:** We think this misunderstanding is caused because we did not appropriately describe our modeling approach. We hope that our new Fig. 1 (see our reply to comment 12 of reviewer 1) and our reply to comment 13 of reviewer 2 address this comment.

    Specifically, decomposition rate estimates shown for "HPM=No" are $k_0$ and for "HPM=No" are $\mu_k$ as shown in Fig. 1. Thus, $k_0$ values ("HPM=No") for different versions of the HPM decomposition module are different because the prior implied by the modified HPM is different in each case. We re-wrote the entire Results section and hope that this point is clarified now. Please see the attached version of the manuscript with the suggested changes.

46. **Q:** Figure 3: Why especially HPMf vs. the LBM is on the 1:1 line although these were not from the same Bayesian model? Why weren't the Hagemann and Moroni estimates tested against the HPM? What are the error bars?

    **A:** In HPMf, the HPM decomposition module is not used as prior for $k_0$ estimated by the LBM. Therefore, the LBM in HPMf is the same as the LBM without the HPM decomposition module.

    Litterbag experiments in Hagemann and Moroni (2015) were reported without WTD

and therefore no predictions of decompositin rates with the HPM decompostion module could be made for these samples. Therefore, data from these experiments contributed to the prior implied by the HPM decomposition module only indirectly.
Points represent average estimates and error bars 95% posterior intervals.
We thank the reviewer for these questions and agree that the figure is more confusing than helpful. We decided to remove the figure and replace it by a hopefully more understandable description. Please see sections 3.1 and 3.2 in the attached version of the manuscript with the suggested changes.

47. **Q:** Figure 4b: Are these results for one species or some kind of summary?

    **A:** These are marginal posterior distributions for parameters that are valid for all species. We also hope that the updated section 2.2.1 clarifies the role of these parameters in the HPM decomposition module (please also refer to our reply to comment 13 of reviewer 2).

48. **Q:** l. 315-316: How significant is it to estimate the initial leaching loss, considering the long-term HPM results about peat accumulation?

    **A:** As suggested in our companion manuscript (Teickner et al., 2024), neglecting initial leaching losses biases decomposition rate estimates and this bias amplifies when these decomposition rate estimates are used for long-term predictions. We suggest to completely re-write this part of the Discussion section and to summarize limitations in a separate section (section 4.4, point 4 in the attached version of the manuscript with the suggested changes).

49. **Q:** l. 340: I would think it is possible that a more accurate fitting of these parameters to litterbag data is not the only key to better understanding of the C accumulation, but there are other factors affecting the processes and differences between different litterbag experiments.

    **A:** The referenced sentence is: "Explaining the discrepancies and finding ways to test them more accurately than possible with available litterbag data should therefore improve our understanding of peat C accumulation."
    We therefore do not state that "more accurate fitting of these parameters to litterbag data is not the *only* key to better understanding of the C accumulation" [our emphasis] and agree that this is only one of the necessary steps.
    In the updated version of the manuscript, we suggest to remove this section because of the suggested re-structuration of the Results and Discussion sections.

50. **Q:** l. 343: How do you conclude this? Wopt was related only to aerobic respiration (Table 1).

    **A:** The referenced sentence is: "The discrepancies in $c_2$ and $W_{opt}$ together imply smaller aerobic and larger anaerobic decomposition rates and therefore a less steep decrease of decomposition rates from oxic to anoxic conditions (Fig. 2)."
    It is true that $W_{\mathrm{opt}}$ is the degree of saturation where the aerobic decomposition rate is maximal. However, in the HPM decomposition module, the anaerobic decomposition rate is computed by transforming the hypothetical aerobic decomposition rate

computed at a degree of saturation of 100% (see equations (5) and (6) in the attached version of the manuscript with the suggested changes). Therefore, the magnitude of anaerobic decomposition rates in the HPM decomposition module depends on $W_{opt}$. We hope that our more detailed description of the HPM decomposition module (see our reply to comment 13 of reviewer 2) addresses this issue, and the re-formulation of the Results and Discussion section, where we now hopefully describe better how the gradient in decomposition rates from oxic to anoxic conditions is controlled by the HPM decomposition parameters (section 3.4 in the attached version of the manuscript with the suggested changes)

51. **Q:** l. 344 and 347: there's perhaps a bit repetition.

    **A:** We did include this part on purpose because the differences in $W_{\mathrm{opt}}$ and $c_2$ only imply relative differences between aerobic and anaerobic decomposition rates, but it will depend on the value of $k_{0,i}$ whether absolute decomposition rates are larger or smaller than implied by the standard parameter values.
    In the updated version of the manuscript, this part is removed.

52. **Q:** l. 351-354: This explanation is complicated. Can you please write it more clearly. How many different runs did you do exactly?

    **A:** We suggest to change the text (ll. 351 to 353) from:

    "We predicted average $k_0$ of *S. fuscum* with HPMe-LE-peat-l0 ($k_{0,\mathrm{modified}}$(HPMe-LE-peat-l0)) and with HPMe-LE-peat-l0 setting either $c_1$, $W_{opt}$, $f_{min}$, or $c_2$ to the standard value ($k_{0,\mathrm{standard}}$(HPMe-LE-peat-l0)) and computed their differences."

    to

    "With these settings, we predicted five sets of average $k_0$: (1) with HPM-leaching ($k_{0,\mathrm{modified}}$(HPM-leaching)). The remaining four sets were also predicted with HPM-leaching, but each time setting one of the HPM decomposition module parameters to their standard value ($k_{0,\mathrm{standard}}$(HPM-leaching)): (2) $c_1$, (3) $W_{opt}$, (4) $f_{min}$, (5) $c_2$. We then computed the difference of $k_0$ from set (1) and (2) to analyze the effect of the new $c_1$ estimate, from set (1) and (3) to analyze the effect of the new $W_{opt}$ estimate, and so on for sets (4) and (5)."

    In addition, we suggest to move this part to the Methods section.

53. **Q:** Figure 6: Please explain the confidence levels.

    **A:** We suggest to add to the caption of Fig. 6 (and also Fig. 7):

    "Shaded areas are central confidence intervals with probabilities given in the figure legend."

54. **Q:** l. 370: What do you mean by the "same study"?

    **A.** The referenced phrase is: "The less pronounced gradient in measured decomposition rates above the water table depth is, however, also visible for *S. fuscum* replicates within the same study …"
    With "same study" we mean litterbag experiments that were described within the same

publication and often were performed with the same original litter material. In these cases, the differences could be explained by differences in sample pre-treatment, litter chemistry, etc. Currently, we think that it is clear from the previous paragraph what is meant by "same study", but if the reviewer disagrees here, we would be grateful for recommendations how to improve our text here.

55. **Q:** One question about Supporting information. You write:

"The litterbag decomposition model combined with each modification of the HPM is obtained from Teickner et al. (2024) (model 1-4, see the supporting information to Teickner et al. (2024) for details). Here, we describe the modules which were added to this model in the different modifications of the HPM"

- Do you mean that you added some modules in this study, which you didn't have in the other work? Please add also this piece of information in the main text, when describing the LBM.

**A:** We hope that the conceptual figure (Fig. 1) and our reply to comment 13 of reviewer 2 address this issue.

**2 Additional changes**

1. We suggest a complete re-write of large parts of the manuscript to address the reviewer comments. Specific aspects of this re-write are listed in the comments of the reviewers, others are too numerous for a list of them to be useful without knowing the context of these changes. Please see the attached manuscript with the suggested changes.

2. We suggest to include Quillet et al. (2015) as reference for studies estimating $c_2$ from peat cores. We suggest the following changes:

   We suggest to change l. 392 to 393 from

   "Larger and smaller $c_2$ than the standard value have been estimated for several permafrost peatland cores with a modified version of the HPM with monthly time step (Treat et al., 2021, 2022)."

   to

   "Larger and smaller $c_2$ than the standard value have been estimated for several peatland cores with the HPM and a modified version with monthly time step (Quillet et al., 2015; Treat et al., 2021, 2022)."

3. Frolking et al. (2010) also mention that peat accumulation as predicted by the HPM is sensitive to $c_2$ and a site-specific parameter. We therefore add Frolking et al. (2010) as reference at l. 47 and 422.

4. In l. 77 we will correct "decmposition" to "decomposition".

[revised manuscript text omitted]

---

## Author Comment (AC3)

**Reply to Anonymous Referee #3 for "Underestimation of Anaerobic Decomposition Rates in Sphagnum Litterbag Experiments by the Holocene Peatland Model Depends on Initial Leaching Losses"**

Henning Teickner[1,2,*]    Edzer Pebesma[2]    Klaus-Holger Knorr[1]

22 December, 2024

[1] ILÖK, Ecohydrology & Biogeochemistry Group, Institute of Landscape Ecology, University of Münster, 48149, Germany
[2] IfGI, Spatiotemporal Modelling Lab, Institute for Geoinformatics, University of Münster, 48149, Germany

[*] Correspondence: Henning Teickner <henning.teickner@uni-muenster.de>

Comments made by the reviewer start with a bold **Q** while our reply starts with a bold **A**. In section "Additional changes" we list additional changes we would like to incorporate in an updated version of the manuscript.

**1 Reply to comments**

**1.1 General comments**

1. **Q:** The authors tested whether the Holocene Peatland Model (HPM) can predict Sphagnum decomposition rates from litterbag experiments. They used a series of modifications of the model's decomposition module in combination with Bayesian methods to get posterior distributions of the model parameters. They focused on the decomposition module of the HPM as a strategy to reduce uncertainty and get more precise model predictions. As much as this study has relevance for the advancement of peatland biogeochemistry modelling, major restructuring of the manuscript should be done before publication.

   **A:** We thank the reviewer for their helpful comments. We agree that we should have described in more detail which equations of the HPM are considered as HPM decomposition module, how we modified them, and how we linked them to the litterbag decomposition model. We hope that our changes suggested below address the issues.

Since we did a complete re-write of large parts of the manuscript, we have attached the manuscript with the suggested changes because we think that this is easier than to list all changes we made here, disconnected from the rest of the text.

2. **Q:** Generally speaking, the writing needs to be improved before this study is published. Sentence construction is often confusing. Furthermore, sentences often do not connect well with each other, creating confusing paragraphs. This hinders the quality of the study and makes it hard to understand, which from the perspective of manuscript-revising is counter-productive. As a general recommendation, try writing shorter sentences with concise ideas (a lot of the sentences here could be split into 2 or even 3 sentences).

   **A:** We thank the reviewer for this suggestion and note that the other two reviewers made similar suggestions. We suggest to simplify sentences and split long sentences where useful.

3. **Q:** I don't find that the title reflects the contents of the paper. In the title they state that initial leaching losses are important to determine decomposition rates. However, in the discussion, this is only brought up briefly in the last paragraph. Most of the discussion revolves around the quality of the litterbag studies, and water table depth parameters c2 and Wopt. Thus, it is hard to see how they came to this conclusion, when the focus of the manuscript is not there. Additionally, it's not clear that HPM underestimates decomposition rates. They should either change the title accordingly, or restructure the manuscript to make sure their point is coming across.

   **A:** We agree that the title should better reflect the contents of the manuscript. We wanted to include initial leaching losses here to mention the limiting factor to our statement about anaerobic decomposition rates. We suggest to change the title to:

   "Peat Oxic and Anoxic Controls of *Sphagnum* Decomposition Rates in the Holocene Peatland Model Decomposition Module Estimated from Litterbag Data"

   In addition, we suggest to re-write large parts of the text. In the updated Results section, we now hope to better describe differences in predicted/estimated decompostion rates under oxic and anoxic conditions between the models and to more reduce the attention on initial leaching losses. Please see the attached version of the manuscript with the suggested changes.

4. **Q:** The abstract does not provide a complete summary of the study. Generally, this section lacks a clear definition of what are the original contributions of this work and what are precedents from literature. There should be a description of the aims of the work and methods in this section. I suggest that you structure this section the same way the paper is structured, covering the introduction, methods, results and discussion sections in the abstract.

   **A:** We thank the reviewer for this suggestion. We suggest a complete re-write of the abstract. Please see the attached version of the manuscript with the suggested changes.

5. **Q:** It is important that the authors clarify what is the difference between the current work and another work currently under revision ("A Synthesis of Sphagnum litterbag

Experiments: Initial Leaching Losses Bias Decomposition Rate Estimates"). The study is only briefly described in the Introduction section, but there are big interactions between that study and the current one. I suggest giving more details about that previous study in the introduction, methods and discussion sections. This will help the reader understand what makes the two studies different, how they were combined, and ultimately what are the original contributions of the current study.

**A:** We thank the reviewer for this suggestion. The other two reviewers made similar suggestions and we therefore suggest to extensively rewrite the introduction and give a detailed description of our modeling approach.

We suggest to introduce "initial leahing loss" already earlier by changing the part starting in l. 59 from

[revised manuscript text omitted]

We introduce a schematic representation of our modeling approach already in the intoduction in Fig. 1 at the end of our reply to this comment.

Finally, we suggest to modify section 2.2.1. to provide a more detailed description of our modeling approach. Please see the attached version of the manuscript with the suggested changes.

Figure 1: Conceptual representation of the modeling approach. Arrows represent flows of information. Litterbag data that have information on water table depths (WTD) and incubation depths are used to estimate average decomposition rates with the HPM decomposition module ($\mu_k$). The HPM decomposition module needs plant functional type identity, peat degree of saturation, WTD, and incubation depth to predict decomposition rates. The modified Granberg model is used to estimate peat degree of saturation at incubation depths from WTD, minimum water content at the surface, and porosity, of which the latter two are estimated from the remaining masses. The litterbag decomposition model is used to estimate decomposition rates ($k_0$) for all litterbag studies, including those that have information on WTD and those that have not. A gamma distribution with $\mu_k$ as average is used as prior distribution for $k_0$ for the litterbag experiments that have information on WTD (curved arrow). This helps to constrain initial leaching loss and decomposition rate estimates for studies that can be predicted with the HPM decomposition module. The Litterbag decomposition module also estimates initial leaching losses ($l_0$) for all litterbag experiments. The equation at the bottom uses these to estimate remaining masses in the litterbag experiments. The litterbag decomposition model is described in more detail in section 2.2.1. See the text for further details.

6. **Q:** One of the main confusions comes from the so-called HPM module and the decomposition model. They are both introduced in the Methods section, but it is not clear what the distinction between them is. Formulas for both models should be defined clearly in this section. And if they're reported on the supplementary material, instead, that should be stated earlier (not 205 lines into the text). Also, since the HPM module is in itself a decomposition model, it is confusing that the other model is just called decomposition model and not something else to distinguish them. Finally, there should be a clear description of the combination of both models. As it is now, the study is not very reproducible from the information presented in the manuscript. The information on the supplementary material (S1 and S2) is full of parameters and formulas, everything written with initials and numbers, but most of them lack any explanation or description of what they are.

   **A:** We hope that our reply to comment 4 of reviewer 3 addresses most parts of this issue.

   - Regarding "Also, since the HPM module is in itself a decomposition model, it is confusing that the other model is just called decomposition model and not something else to distinguish them.": We now clearly distinguish between the litterbag decomposition model and the HPM decomposition module in the text.
   - Regarding "The information on the supplementary material (S1 and S2) is full of parameters and formulas, everything written with initials and numbers, but most of them lack any explanation or description of what they are.": We suggest to remove supporting information S1 as all formulas are now described in the main text. We also changed the symbols used to make the formulas prettier.

7. **Q:** Just an idea, the names of the models are complicated to read and remember. So, I suggest you give them more friendly names like HPM-standard, HPM-peat, HPM-all HPM-leaching, and HPM-outlier, or maybe just use less initials. This is only a suggestion, but I encourage you to do it if you really want readers to remember the differences between each model without having to decipher long combinations of initials.

   **A:** We think that these suggestions are good and we suggest to include them in our study. Please see the attached version of the manuscript with the suggested changes.

8. **Q:** Another main point is that throughout the introduction, results and discussion sections they mention many times aerobic/anaerobic or oxic/anoxic decomposition rates. However, it is not declared in the Methods section how this is considered in the models. From context, I figured out it has to do with water table depth, but specifically how this is incorporated in calculations is not defined. Please, make sure this is clear since it seems to be a big point for your study.

   **A:** We hope that our reply to comment 4 of reviewer 3 addresses this issue.

9. **Q:** The Results section could be improved. As it is now, it's not very well structured, there are too many very specific sub-sections. Also, results in the figures are discussed in many sections without a clear order. So first, I suggest making fewer sub-sections.

Second, try making sub-sections that correspond to specific figures or tables. This will make it easier to read and to find the data in the manuscript.

**A:** We completely re-wrote the Results section and hope that the updated version addresses this comment. Please see the attached version of the manuscript with the suggested changes.

**1.2 Specific comments**

**1.2.1 Abstract**

11. **Q:** Line 4: This sentence is not clear. I suggest rephrasing it, maybe try splitting it into two shorter sentences. Also, how are the "large uncertainties" allowing the fitting of the model?

    **A:** We thank the reviewer for this suggestion. We tried to improve this part of the abstract. Please see our reply to comment 4 of reviewer 3.

12. **Q:** Line 14: The sentence starting with "Based on previous…" is not clear. Who did the previous analysis, the same group of researchers or others? Are the updated parameter estimates part of the results of this work?

    **A:** We refer here to the sensitivity analyses conducted in Frolking et al. (2010), Quillet et al. (2013a), and Quillet et al. (2013b), so work done by others several years ago. We think that the abstract states clearly that these sensitivity analyses are not part of the current study ("previous") and we think that, while reading the abstract, it is not relevant to know who did this work. In the text, we explicitly cite the studies that provide the sensitivity analyses and we therefore think that it is clear that we are not the same authors who conducted the sensitivity analyses.

**1.2.2 Introduction**

13. **Q:** Line 19: Is "litter that does not decompose fast even under more conditions facilitating microbial decomposition" supposed to refer to litter quality/traits? I suggest rephrasing this part, since the English is not quite correct, and the scientific concepts are not quite clear.

    **A:** Yes, this refers to litter chemical and physical properties that decrease decomposition rates. We think that litter recalcitrance is a known scientific concept, but we agree that this sentence should be clarified. We suggest to change it to:

    "Compared to other ecosystems, northern peatlands usually have small decomposition rates because of cold temperatures, high water table levels, acidic pH value, and litter that does not decompose fast even under environmental conditions favorable for decomposition (van Breemen, 1995; Rydin et al., 2013)."

14. **Q:** Line 29: The "first" problem as it is written is not clear. Who are "they" (citation?)? Maybe flip the order of the sentence to put the actual problem of the model in the focus.

    **A:** "They" refers to global tests, not studies or persons. We agree that this sentence can be improved. We suggest to change it from

    "First, they test entire peatland models against observed data and thus can identify the parameter values or model equations that cause observed discrepancies less reliably."

    to

    "First, they cannot reliably identify the parameter values or model equations that cause discrepancies between model predictions and measurements because they test entire peatland models against observed data."

15. **Q:** Line 41: Could you explain why this approach is only useful for short term predictions?

    **A:** Litterbag experiments have been conducted only for few years and mostly with fresh litter samples. Based on such experiments, the slow-down of decomposition rates over time that is assumed in the HPM cannot be reliably estimated (Frolking et al., 2001; Teickner et al., 2024).
    Within the framework of the HPM, one could use litter of already decomposed samples (e.g. *Sphagnum* peat samples from larger depths) and conduct litterbag experiments with these, but only if one would also know the degree of decomposition (fraction of initial mass the sample has already lost) to estimate the slow-down of decomposition rates. We are not aware of a reliable method to estimate the degree of decomposition for such samples and therefore such analyses are not possible yet, as far as we know. To briefly mention this in the text, we suggest to change the sentence to:

    "Admittedly, such a test is restricted to short time ranges and not representative for long-term decomposition rates which may differ from that of fresh litter (e.g., Frolking et al., 2001), but future tests with different scope and applications of the model will benefit from the reduced parameter uncertainties and can consider where the model fails already on short time scales."

16. **Q:** Line 48: How is this proposal different from what was done before as mentioned in line 45?

    **A:** To clarify this, we suggest to add before this sentence:

    "These sensitivity analyses used assumed parameter ranges that are not informed by litterbag experiments."

17. **Q:** Line 62: This sounds very similar to the scope of the current study. How are they different?

    **A:** We hope that our reply to comment 4 of reviewer 3 addresses how the scopes of the two studies are different. As mentioned there, we suggest to change this sentence to avoid confusions with our previous study. The suggested changes are:

"The recently published Peatland Decomposition Database (Teickner and Knorr, 2024b) contains raw data from available *Sphagnum* litterbag experiments and therefore allows to estimate parameters with any mass loss-based decomposition model and therefore also allows to consider initial leaching losses."

18. **Q:** Line 68: What are the 5 parameters?

    **A:** These are the five parameters listed in Tab. 1 (where we count $k_{0,i}$ as one parameter since only one of these parameters is used to predict a specific decomposition rate). We think that mentioning these already here would be a distraction, but if the reviewer has a good suggestion how to include this information here, we would be grateful for further comments.

19. **Q:** Line 84: Both hypotheses here seem more like predictions.

    **A:** We do not understand this comment. A hypothesis is a prediction made with the intention to test it against measurements or, more generally, other predictions. Therefore, it is not surprising that our hypotheses are predictions.
    We are aware that some sources distinguish between hypotheses and predictions (for example, a quick search returned https://www.trentu.ca/academicskills/how-guides/how-succeed-math-and-science/writing-lab-reports/understanding-hypotheses-and), however we would label these as "theory" and "prediction", respectively. We see that there is no uniform definition, but we do not think the hypotheses as stated in our manuscript cause too much confusion.
    However, since the hypotheses are rather trivial, we suggest to not mention them explicitly any more and therefore suggest to remove our list of hypotheses.

**1.2.3 Methods**

20. **Q:** Line 99: Could you specify if the samples were buried or on the surface?

    **A:** Reviewer 2 (comment 19) suggested to include a table with general information on the number of litterbag experiments per species. We suggest to include an additional column that gives the depth range over which litterbags were buried there. The table is as follows:

Table 1: Overview on litterbag experiments included for each *Sphagnum* taxon in this study. "HPM microhabitat" is the HPM microhabitat assigned to each taxon. Taxa without value are not considered in Johnson et al. (2015) (see section 2.2.2). "Number of experiments" is the number of litterbag experiments available from the Peatland Decomposition Database (these are either individual replicates or average values of replicates, depending on what data were reported in the studies). "Number of experiments with WTD data" is the number of litterbag experiments that also report water table depths and for which we therefore could make predictions with the HPM decomposition module. "Depth range" are the maximum and minimum depth below the peat surface at which litterbags were placed [cm]. Missing values mean that no study reported depths.

| Taxon | HPM microhabitat | Number of studies | Number of experiments | Number of experiments with WTD data | Depth range |
|---|---|---|---|---|---|
| *Sphagnum* spec. | | 2 | 16 | 10 | 10, 30 |
| *S. angustifolium* | Hummock | 4 | 14 | 8 | 1, 30 |
| *S. auriculatum* | | 1 | 3 | 0 | 0, 6 |
| *S. balticum* | Lawn | 3 | 12 | 3 | 1, 30 |
| *S. cuspidatum* | Hollow | 1 | 5 | 5 | 10, 50 |
| *S. fallax* | Lawn | 1 | 4 | 1 | 1, 1 |
| *S. fuscum* | Hummock | 9 | 32 | 13 | 1, 50 |
| *S. lindbergii* | Lawn | 1 | 2 | 0 | |
| *S. magellanicum aggr.* | Hummock | 3 | 7 | 5 | 1, 50 |
| *S. majus* | Hollow | 1 | 2 | 2 | 10, 30 |
| *S. papillosum* | Lawn | 2 | 6 | 1 | 0, 1 |
| *S. rubellum* | Hummock | 1 | 2 | 2 | 10, 30 |
| *S. russowii* | Hummock | 1 | 3 | 2 | 1, 1 |
| *S. russowii* and *capillifolium* | | 1 | 18 | 0 | 5, 5 |
| *S. squarrosum* | Lawn | 1 | 2 | 0 | 0, 0 |
| *S. teres* | Lawn | 1 | 1 | 1 | 2, 2 |

21. **Q:** Line 105: What were the criteria for including these studies in your work? Is this an exhaustive list? It's not clear in which part of the study did you include the papers with water table depth data and in which part did you include the rest of the studies. How and for what purpose did you test one set of k0 against the other set?

    **A:** We hope that our reply to comment 4 of reviewer 3 addresses this issue. The list of studies provided is an exhaustive list.

22. **Q:** Line 109: when you speak of "module", is this a part of the HPM? If so, elaborate on the model and its so-called modules.

    **A:** We hope that our reply to comment 4 of reviewer 3 addresses this issue.

23. **Q:** Line 128: what does PFT stand for?

    **A:** We thank the reviewer for catching this error. We suggest to remove the abbreviation from the subsection title and introduce the abbreviation in ll. 109 to 111. (old version of the manuscript) The changed sentence is:

    "To predict decomposition rates, the HPM decomposition module needs as inputs the litter type in terms of the HPM plant functional types (PFT), the fraction of mass already lost due to previous decomposition, the depth of the litter below the peat surface, the water table depth, and the peat degree of saturation (Frolking et al., 2010)."

24. **Q:** Line 136: can you elaborate on the criteria of species classification? How was this criterion decided upon?

    **A:** We are not sure whether we understand this question correctly. We think the reviewer means assignment to PFT when they refer to "species classification" (and not taxonomic classification), however we think that this section describes in detail the criteria by which we assigned species to PFT and ll. 128 to 132 mention how these criteria are based on the WTD niches the HPM assumes for the PFT.

25. **Q:** Line 140: hummock species decompose slowly, I suppose? If assumptions like this are made, they need to be stated clearly and with citations to back them up.

    **A:** Yes, this is a common assumption. We suggest to change this part of the text from

    "Litterbag data from Prevost et al. (1997) are incubations of peat samples where the species is unknown. Based on descriptions in the paper, it is likely that the peat was formed by hummock species. In addition, decomposition rate estimates for these samples are small. For these reasons, we assigned these samples to the hummock PFT of the HPM."

    to

    "Litterbag data from Prevost et al. (1997) are incubations of peat samples where the species is unknown. Based on descriptions in this study, it is likely that the peat was formed by hummock species. Hummock species are assumed to have the smallest decomposition rate among the three *Sphagnum* PFT in the HPM (Frolking et al., 2010) and this is in line with small decomposition rate estimates for these samples (Teickner

et al., 2024). For these reasons, we assigned these samples to the hummock PFT of the HPM."

26. **Q:** Line 145: here you bring up for the first time the names of two models, but you have not introduced them yet in the text. Maybe consider describing the models first.

    **A:** We think that it would not improve the clarity of the text if we would already here introduce the different model versions because some of these were only computed because of data gaps that are described in the previous subsections. To avoid confusion, we suggest to refer to the subsection where the models are described. The sentence then is:

    "We therefore estimated $k_{0,i}$ for individual *Sphagnum* species in models HPM-all, HPM-leaching, and HPM-outlier (see section 2.3.1) and evaluated the variability of these species-specific estimates compared to the standard $k_{0,i}$ values of the HPM *Sphagnum* PFT."

27. **Q:** Line 147: Describe the Granberg model's formula and parameters.

    **A:** We suggest to change this subsection to:

    "We estimated the degree of saturation with the modified Granberg model (ModGberg model) (Granberg et al., 1999; Kettridge and Baird, 2007) from minimum water content at the surface ($\theta_{0,\mathrm{min}}$), total porosity ($P$), the water table depth ($z_{\mathrm{wt}}$), and the depth of the litterbags below the peat surface during the incubation ($z$):

$$
\begin{aligned}
\theta(z) = & \quad \min\left(P, \theta_0 + (P - \theta_0)\left(\frac{z}{z_{\mathrm{wt}}}\right)^2\right) \\
\theta_0 = & \quad \max\left(\theta_{0,\mathrm{min}}, 0.15 z_{\mathrm{wt}}^{-0.28}\right),
\end{aligned}
\tag{1}
$$

    where $\theta_0$ is the water content at the surface and $0.15 z_{\mathrm{wt}}^{-0.28}$ is an empirical relation for $\theta_0$ in dependency of the WTD estimated in Kettridge and Baird (2007).
    The minimum water content at the surface was not reported in any study and we therefore assumed a minimum water content at the surface of $0.05\ \mathrm{L_{water}\ L_{sample}^{-1}}$ with a standard deviation of $0.05\ \mathrm{L_{sample}^{-1}}$, based on measurmeents from Hayward and Clymo (1982). The total porosity was not reported in any study and therefore we assumed an average value of 80% with a standard deviation of 10%, roughly based on values reported for low-density *Sphagnum* peat (Liu and Lennartz, 2019). An improved test of the HPM decomposition module would require litterbag experiments with direct measurements of the degree of saturation at sufficient temporal resolution."

28. **Q:** Line 163 and 168: what is the "decomposition model" here if not the HPM module?

    **A:** We hope that our reply to comment 4 of reviewer 3 addresses this issue.

29. **Q:** Line 210: decomposition rates predicted by HPM correspond to each version of the HPM model or just the standard one? And how exactly were the rates estimated from the litterbag data if not from the HPM model?

**A:** We hope that our reply to comment 4 of reviewer 3 addresses this issue. In addition, we completely re-wrote this part of the text. Please see the attached manuscript with the suggested changes.

30. **Q:** Line 211: Shouldn't a high probability indicate a misfit? If the difference is not significatively different from 0, then doesn't it mean that the HPM predicts well decomposition rates from the litterbag data?

    **A:** We thank the reviewer for reporting this error. We suggest to correct it as suggested. However, if the difference is not significatively different from 0, this does not imply that the HPM predicts well decomposition rates from the litterbag data because non-significance may also be caused by large errors in estimates. Non-significance then indicates that the test did not have sufficient capacity to detect a difference. What is a "large error in estimates" depends on what one considers as relevant difference and this will depend on the purpose of an analysis.

31. **Q:** Line 230: In the supplementary information S3 you mention R was used, but in this section, you only mention Stan. Please, give a detailed description of the software used, including which methods were done with each software.

    Line 239: For disclosure, I am familiar with Bayesian Methods, but not with cross-validation and power-scaling.

    **A:** Stan is a probabilistic programming language. All models were coded in Stan and estimated with R via the R interface to Stan (rstan) package. To clarify this, we suggest to change the sentence starting in l. 232 to:

    "Bayesian computations were performed using Markov Chain Monte Carlo (MCMC) sampling with Stan (2.32.2) (Stan Development Team, 2021b) in R (4.2.0) (R Core Team, 2022) via the rstan package (2.32.5) (Stan Development Team, 2021a) …"

**1.2.4 Results**

32. **Q:** Figure 1: What type of error is used in this figure? Also, shouldn't the values predicted from litterbag data always be the same?

    **A:** We hope that our reply to comment 4 of reviewer 3 addresses this issue. In particular, $k_0$ estimated from the litterbag data with the different models are different because the HPM deomposition module acts as prior for the $k_0$ estimates for litterbag experiments with reported WTD (Fig. 1, this reply document). Since this HPM deomposition module (or its inputs) are different between the models, the $k_0$ estimates can be different, too.
    We suggest to add to the caption of Fig. 1 to 3 (Fig. 2 in the new version):

    "Points represent average estimates and error bars 95% posterior intervals."

33. **Q:** Line 252: You say that all k were undersestimated for this species but in the Figure the points not predicted with HPM are present only in one of the graphs. Why is that?

**A:** The points predicted with the litterbag prediction model are also shown for the other panels, but not visible due to overlap. Moreover, the referenced paragraph refers only to HPMf, not to all model versions.

Please note that we suggest to completely re-write the Results section and that we suggest to replace the old Fig. 2 by two new figures that more appropriately illustrate the differences between HPM-standard and LDM-standard on the one hand and between HPM-standard and the modifications of the HPM decomposition module on the other hand. Please see section 3.4 in the attached version of the manuscript with the suggested changes.

34. **Q:** Line 263: why is it incompatible?

    **A:** The referenced sentence is: "In combination with the improved fit of HPMf-LE-peat, this indicates that uncertainties in the litterbag data are large enough to make the HPM decomposition module compatible with the litterbag decomposition rates by varying the magnitude of decomposition rates and initial leaching losses, even though the standard HPM decomposition module parameters are not necessarily (most) compatible with the data."

    Thus, we do not state that the parameter values indeed are compatible. The reason why we state that these parameter values are not necessarily (most) compatible with the data is that when $k_0$ and $l_0$ are estimated without using the HPM decomposition module as prior (HPMf), these estimates are different and they are again different when we also estimate the HPM deomposition parameters from the litterbag data (HPMe-LE-peat-l0). Moreover, we also estimated the parameters of the HPM deocmposition module from the litterbag data and found that the estimates differ from the standard values. This indicates some incompatibility between the litterbag data and the standard parameter values of the HPM decomposition module.

    We suggest to completely re-write the Results section and this sentence will then be removed. Please see the attached manuscript with the suggested changes.

35. **Q:** Figure 2: Why are the values for the k not predicted from HPM different in every frame? Shouldn't they be constant for each species and have only the values from the HPM vary with each model version? Also, what do negative and positive values of water table depth mean if the are relative to the litterbag?

    **A:** We think this misunderstanding is caused because we did not appropriately describe our modeling approach. We hope that our new Fig. 1 and our reply to comment 4 of reviewer 3 address this comment.

    Specifically, decomposition rate estimates from the litterbag decomposition model are different for each HPM decomposition module version because these different HPM decomposition module versions were used as prior for $k_0$ estimated by the litterbag decomposition model. Thus, since the priors are different, so are the estimates.

    To avoid confusions, we suggest a complete re-write of the Methods and Results sections that hopefully better describes the differences between HPM-standard and LDM-standard on the one hand and between HPM-standard and the modifications of the HPM decomposition module on the other hand. We also suggest to replace Fig. 2 by two new figures (Fig. 4 and 5 in the new version of the manuscript) to avoid such

misunderstanding.

The x-axis in Fig. 2 (in the old version of the manuscript) is the depth of the water table below the litterbag. Negative values occur whenever the litterbag was buried below the average annual WTD (as estimated from the information given in the studies). Thus, for some studies some litterbags were buried more than 25 cm below the average annual WTD. We suggest to replace Fig. 2 by two new figures and suggest to describe in detail in the caption what the x-axis values mean. Please see section 3.4 in the attached version of the manuscript with the suggested changes in the Results section.

36. **Q:** Figure 3: How do the data points in the litterbag model vs HPMf plots exactly show a linear relationship? From this it does not seem that the more complex models make better predictions. Also, what is the meaning of the "-" in "(mass-%)"?

    **A:** In HPMf, the HPM decomposition module is not used as prior for $k_0$ estimated by the litterbag decomposition model. Therefore, the litterbag decomposition model in HPMf is the same as the litterbag decomposition model without the HPM decomposition module.
    We think the figure is more confusing than helpful. We therefore suggest to remove it and replace it by a more clear description of the results in section 3.1 and 3.2 (in the attached version of the manuscript with the suggested changes).

    the "-" in "(mass-%)" is just part of the unit, which is the fraction of initial mass lost (multiplied by 100). In German you would write it with a "-", but perhaps not in English. We suggest to remove the "-" here and elsewhere.

37. **Q:** Line 274: But in Fig. 3 you see that the best predictions are achieved with the HPMf version?

    **A:** We hope that our reply to comment 36 addresses this issue.

38. **Q:** Line 278: this is not quite visible in Fig. 2.

    **A:** The referenced sentence is: "HPMe-LE-peat-l0 estimates a larger average maximum possible decomposition rate, particularly for *S. angustifolium*, than the other models (Fig. 2 and supporting Fig. S9)."
    We agree that this is difficult to see from Fig. 2. We suggest a complete re-write of the Results section, including Fig. 2. In particular, we split Fig. 2 into two figures (new Fig. 4 and Fig. 5) and Fig. 5 (also shown below) now shows the difference of $k_0$ predicted by the other model versions on the one hand and $k_0$ predicted by HPM-standard on the other hand. We hope that this figure better shows that $k_0$ predicted by HPM-leaching (previously HPMe-LE-peat-l0) for *S. angustifolium* is larger on average than for the other models. The difference in $k_0$ estimates is now described in section 3.2 (Please see the attached manuscript with the suggested changes).

[Figure]

Figure 2: $k_0$ predicted by HPM decomposition module modifications (either HPM-peat, HPM-all, or HPM-leaching) minus $k_0$ predicted by the HPM decomposition module with standard parameter values (HPM-standard) versus estimated average water table depths below the litterbags. Points represent average estimates and error bars 95% posterior intervals. *Sphagnum* spec. are samples which that been identified only to the genus level. Only data for species with at least three replicates are shown.

39. **Q:** Line 280: none of the results discussed in this paragraph are shown in Fig. 2 or S3.

    **A:** The paragraph is: "In contrast to estimates for $k_0$, $l_0$, and $k_{i,0}$, the other HPM decomposition module parameters had similar estimates for HPMe-LE-peat and HPMe-LE-peat-l0 and as a consequence relative differences of decomposition rates along the water table depth gradient are very similar between all models (Fig. 2). Estimates for $f_{min}$ did not differ much to the prior value and the power-scaling sensitivity analysis indicates a weak influence of the data (supporting information S3) and therefore that available litterbag data provide only little information about minimum decomposition rates under anoxic conditions."
    We state that "relative differences of decomposition rates along the water table depth gradient are very similar between all models (Fig. 2)". Fig. 2 shows $k_0$ ("Predicted with HPM? = No", see equation (8) in the new version of the manuscript) and $\mu_k$ estimates ("Predicted with HPM? = Yes", equation (5) in the new version of the manuscript) for the different models versus WTD relative to the litterbags and therefore allows to compare "relative differences of decomposition rates along the water table depth gradient". We agree that Fig. 2 is not be the best way to illustrate our results and we therefore suggest to replace it by two new figures, Fig. 4 and Fig. 5 (See our reply to comment 38 of reviewer 3 and the attached manuscript with the suggested changes).
    We also state that "Estimates for $f_{min}$ did not differ much to the prior value and the power-scaling sensitivity analysis indicates a weak influence of the data (supporting information S3)". It is true that supporting information S3 does not provide additional results to justify this statement, but it describes the power scaling analysis and this is all we wanted to reference here.
    We hope that the new Results sections (sections 3.3 and 3.4 in the attached manuscript with the suggested changes) avoids misunderstandings. If the reviewer still thinks that some of the results are prestend incompletely, we would be thankful for further comments.

40. **Q:** Figure 4: it is not possible to differentiate between the species in panel a. Use different colors. Remind the reader in the figure caption what are the other HPM parameters. Apply the same recommendations for the equivalent figures in the Supplementary material, since they are similar.

    **A:** We agree that Fig. 4 (a) is cluttered, but we do neither think that colors are useful here, nor that it is impossible to roughly differentiate the species, and we think that all the figure should show is a rough indication of the variation in $k_{0,i}$ estimates for individual species relative to the HPM standard values for the corresponding PFT. If a reader wants more information, they can access them from the 'hpmdpredict' R package (Teickner and Knorr, 2024a).
    We do not think that colors are useful because there are lines that link each species name to the average value of the marginal posterior distribution shown in the plot and this gives a rough indication of the average $k_{0,i}$ for each species. We do not expect that a reader wants to read off this graph more than this rough information. If more detailed information are of interest, these can be derived from the published data and code. In particular, species-specific values for $k_{0,i}$ are also available through the 'hpmdpredict' R package (Teickner and Knorr, 2024a).

To remind readers what the other HPM parameters are, we suggest to include a reference to Tab. 1 in the caption (and to add similar information to the corresponding supporting figures). In addition, we suggest to switch panels (a) and (b):

"Marginal posterior distributions of HPM decomposition module parameters (see Tab. 1). (a) Marginal posterior distributions for $c_1$, $W_{opt}$, $f_{min}$, and $c_2$. (b) Marginal posterior distributions for $k_{0,i}$ (maximum possible decomposition rate for species $i$). Species were assigned to HPM microhabitats as described in section 2.2.2. Vertical black lines are the standard parameter values from Frolking et al. (2010). *Sphagnum* spec. are samples that have been identified only to the genus level."

41. **Q:** Line 299: Could this be because the standard value is an average for all hummock species?

    **A:** The referenced sentence is: "Both models also estimate a large posterior probability ($> 95\%$) that *S. russowii* and *S. rubellum* have a larger, and that *S. cuspidatum* has a smaller maximum possible decomposition rate ($k_{0,i}$) than the standard values for the respective PFT (Fig. 4 and supporting Fig. S11)."
    We agree that one standard value for all hummock species may not appropriately represent the variability in $k_{0,i}$ when estimated from litterbag data and we suggest to discuss this in sections 4.1 and 4.4 of the new manuscript (Please see the attached manuscript with the suggested changes) (in the old version, this was discussed in section 4.6).

42. **Q:** Line 302: Please, rephrase this, it's not clear what you mean.

    **A:** We thank the reviewer for this suggestion. We suggest to change the sentence from

    "However, because of the larger variability of $k_{0,i}$ in the cross-validation (compare with the previous subsection), this discrepancy is probably more uncertain when new data would become available."

    to

    "However, estimates for $k_{0,i}$ were very variable for the same species when different subsets of the litterbag data were used to estimate the model in the cross-validation. This indicates that samples of the same species from different studies have a large variability in $k_{0,i}$ values."

43. **Q:** Figures 5, 6 and 7 are not mentioned in the results section whatsoever. The results therein should be described and discussed just as with the rest of the figures and tables.

    **A:** It is true that Fig. 5, 6 and 7 have not been mentioned in the Results section. We agree that it makes sense to re-write the Results and Discussion sections. We suggest to include Fig. 5, Fig. 6 and part of the Discussion section relevant here in the new sections 3.4 and 3.5. (Please see the attached manuscript with the suggested changes). Fig. 7 has a specific purpose in our discussion, namely to visualize that the model suggests largest deocmposition rates shortly above the annual average WTD, as has been observed in previous studies. It therefore illustrates a comparison of our

model with previous studies and should, in our opinion, be part of the Discussion section, since otherwise it would be disconnected from the part of the text where it is referenced.

**1.2.5 Discussion**

44. **Q:** Line 314: I would not say this is a result of your study. This is maybe a consequence of the results in the study. How would you describe these litterbag experiments that allow to estimate initial leaching losses more accurately?

    **A:** The referenced sentence is: "Therefore, an important result of our study is that stronger tests of the HPM decomposition module and other peatland models require litterbag experiments that allow to estimate initial leaching losses more accurately than is possible with available experiments."
    We suggest a complete re-write of the Discussion section and suggest to remove this part as a consequence. Limitations of our study are now discussed in section 4.4. The first part of the Discussion section would mention this aspect only briefly. Please see the attached manuscript with the suggested changes.

45. **Q:** Line 320: is the steep gradient in decomposition rates really a sign of bad model fit or could it be just that decomposition changes with environmental factors such as water table depth?

    **A:** The referenced paragraph is: "In the following sections, we discuss these discrepancies. In particular, we show that they imply a less steep gradient of decomposition rates from oxic to anoxic conditions than assumed by the standard HPM decomposition module. We discuss how reliable this pattern is, considering that the data are from heterogeneous studies, what processes may cause the less steep gradient, and how important the suggested differences in parameter values are for the predicted C accumulation."
    In this paragraph, we do not state that "the steep gradient in decomposition rates really [is] a sign of bad model fit". More specifically, in the mentioned subsections of our manuscript, we analyze how reliable the discrepancies are that we identified (section 4.3) and clearly state that (ll. 376 to 378) further litterbag experiments need to test how reliable the discrepancies are.
    Regarding "or could it be just that decomposition changes with environmental factors such as water table depth?": Yes, and the HPM decomposition module is supposed to accurately describe this gradient. If it does not, either the HPM decomposition module or the litterbag experiments need to be improved and this is what we intended to discuss in sections 4.3, 4.4, and 5.
    We suggest to re-write the Results and Discussion sections to hopefully avoid misunderstandings. In particular, the predicted gradient in decomposition rates is now described in the Results section (new section 3.4). We also hope that our new Discussion section makes clear that what we identify is a discrepancy that needs further experiments to rule out confounding factors also on the experimental side.

46. **Q:** Line 349: This should be in the Results section.

    **A:** As mentioned in our reply to comment 43 of reviewer 3, we moved this part to the Results section (see section 3.5 in the attached manuscript with the suggested changes).

47. **Q:** Lines 384-391: This should be in the Results section.

    **A:** As mentioned in our reply to comment 43 of reviewer 3, we think that we can better present our argument by illustrating it with additional simple applications of the model in the discussion section and we think that it would be more confusing to add this paragraph to the Results section. This paragraph is tightly linked to the argument we make here.

48. **Q:** Line 433: Since you're making inferences based on another paper (Quillet et al., 2013a), please give more details about this study.

    **A:** This is a good idea. We suggest to change the sentence in l. 429 from

    "Previous sensitivity analyses identified $c_2$ as influential for C accumulation in the HPM (Quillet et al., 2013a, b)."

    to

    "Previous global and local sensitivity analyses, where HPM parameter values were varied in broad ranges and environmental conditions were varied, identified $c_2$ as influential for C accumulation in the HPM (Quillet et al., 2013a, b)."

49. **Q:** Line 438: I don't understand how larger anaerobic decomposition may result in higher water table levels.

    **A:** Larger anaerobic decomposition may result in higher water table levels because long-term decomposition of peat reduces the thickness of the peatland which lets the water table increase relative to the peat surface (unless the weather becomes so dry that this effect is compensated by net water losses).
    We currently think that the interested reader can look up details in (Quillet et al., 2013a), but if the reviewer thinks this is critical information, we would be grateful for further comments.

50. **Q:** Line 482: I believe this is the first time you have brought up the leaching argument in the Discussion, but somehow leaching is mentioned in the title as a central result. The focus of the manuscript is not on leaching at this moment, but most of the attention is on c2 and Wopt. I suggest changing the title accordingly.

    **A:** We agree with the reviewer and hope that our reply to comments 3 and 5 of reviewer 3 address this issue.

51. **Q:** Line 483: Fig. 2 does not show clearly that the estimates of k are larger in the HPMe-LE-peat-l0 as you suggest. If anything, they look similar or even lower.

    **A:** We agree with the reviewer that this is not clearly visible from Fig. 2 and we also agree (based on supporting Fig. S9) that $k_{0,i}$ estimates are not systematically

larger for HPMe-LE-peat-l0. However, our results support that errors in $k_{0,i}$ can be decreased with more accurate estimates for $k_0$. We therefore suggest to remove the paragraph starting in l. 479 (in fact, we suggest to completely remove section 4.6 and integrate it into the other Discussion section and add a new section 4.4, where we discuss limitations and ways to improve our test). Please see the attached manuscript with the suggested changes. We thank the reviewer for bringing this issue to our attention.

52. **Q:** Line 484: How do you suggest better estimates of leaching losses can be achieved in future experiments?

    **A:** The answer to this question is addressed in our other manuscript (Teickner et al., 2024) and therefore outside the scope of this study. In brief what we found in this other study, based on a sensitivity analysis of the litterbag decomposition model, is that litterbag experiments should sample one batch of litterbags shortly (few days to weeks) after the start of the experiment. We now suggest to summarize limitations of our test and suggested improvements in section 4.4. Please see the attached manuscript with the suggested changes.

**1.2.6 Supporting information**

53. **Q:** S1: Most of the variables and parameters in this section are not defined. I suggest presenting this section in table form with 3 columns, where one of the columns has human-friendly names for each variable/parameter/model. Making the reader go read three other papers to understand the formulas is not very mindful.

    **A:** We thank the reviewer for this suggestion. We suggest to remove supporting section S1 since the listed equations would now be included in the main text. The prior distributions where this is not the case are also listed in supporting information S2 (now supporting information S1). We also suggest to expand this table by including more friendly parameter names for key parameters (the same names now also used in the equations in the main text) and by adding a column that references the equation where the parameter occurs. This table would look as follows:

Table 2: Prior distributions of all Bayesian models and their justifications. "Parameter name in code" is the name for the parameter as used in our Stan models. "Parameter name in text" is the name of the corresponding parameter we use in the main text and figures. "Equation in main text" reerences the equation in the main text where the parameter occurs. When there is no value for "Justification", the prior was chosen based on prior predictive checks against the data. This prior predictive check tests whether the models can produce distributions of measured variables we expect based on prior knowledge. The results of these prior predictve checks are shown in supporting section S3.

[revised manuscript text omitted]

54. **Q:** S2: same as S1. Also, S2 does not seem to be referenced in the main manuscript, but if it's relevant to the manuscript, it should be mentioned.

   **A:** It is true that supporting information S2 is not mentioned in the main text, but its content, supporting Tab. S1 is referenced in l. 238 (new: l. 307 and l. 375).

55. **Q:** S3: when you say "all other computations were done in R", which ones weren't? Please, detail all the software used in this study.

    **A:** All used software is listed. We suggest to remove "other" to make clear that all computations were made in R. Please also see our reply to comment 31 of reviewer 3.

56. **Q:** S4: This section does not seem to be referenced in the main manuscript nor do the individual figures in it, but if this is relevant to the manuscript, it should be mentioned.

    **A:** We thank the reviewer for this suggestion. We suggest to reference section S4 (new: S3) in the caption of Tab. S1:

    "The results of these prior predictve checks are shown in supporting section S3."

    We also suggest to reference section S4 (new: S3) in the main text at l. 238:

    "All models used the same priors for the same parameters and prior choices are listed and justified in supporting Tab. S1. Results of prior and posterior predictive checks are shown in supporting information S3."

57. **Q:** S5: In the caption of Fig. S9 you wrote twice HPMe-LE-peat-l0, but one of them should be HPMe-LE-peat. The axis titles are switched (k should be x axis and Species should be y axis). Also, I suggest separating vertically points corresponding to each model, and for the different depths of Sphagnum spec. as well. This way you will avoid the overlap.

    **A:** We thank the reviewer for pointing us to these errors and potentials for improvements. We will correct and include them as suggested.

58. **Q:** S6: I suggest differentiating species with colors in Figs. S10 and S11, and defining what parameters are in the captions of Figs. S10, S11 and S12.

    **A:** We hope that our reply to comment 40 by reviewer 3 addresses this issue.

59. **Q:** S7: If I understand correctly, data points on the positive end of the x axis are covered in water? If so, I would expect leaching to be higher under those conditions, but somehow it does not seem like it. I find this interesting, could you discuss this in the main manuscript?

    **A:** It is the other way around: positive values indicate that the literbag was placed above the water table level and negative values mean that they were covered with water. We suggest to include the following note to the captions of Fig. 2 (new: Fig. 4), and supporting Fig. S13 to S15 to clarify this:

    "(negative values represent litterbags placed below the water table, positive values represent litterbags placed above the water table in the unsaturated zone)"

    As mentioned in the manuscript (l. 286), both positive and negative relations of $l_0$ to the degree of saturation would be compatible with the data. We had expected, in line with Lind et al. (2022) that $l_0$ would be larger under more saturated conditions, however other factors may confound this pattern or the $l_0$ estimates may simply have too large errors to detect such a relation (Teickner et al., 2024) and we therefore do not discuss any patterns in more detail in the main text.

Please note that results on the relation of initial leaching losses t the degree of saturation are now described in section 3.6 and discussed in point 4 of section 4.4.

60. **Q:** S9: Which one of the two panels uses the standard value for $W_{opt}$ or the $W_{opt}$ value estimated by HPMe-LE-peat-l0?

    **A:** We thank the reviewer for poiting out this error: one of the labels was missing. We have corrected the figure.

61. **Q:** S11: You say decomposition was simulated either under a degree of saturation of 0.6 L/L or 20 cm below the water table. But in the figure, you included -20 cm and 10 cm, so please make sure the description coincides with the figure.

    **A:** We thank the reviewer for this suggestion. We chose some arbitrary position above the water table and fixed the degree of saturation to 0.6 L/L. We suggest to change the sentence to:

    "To illustrate that the HPM decomposition module implies large uncertainties if its parameters are estimated from available litterbag data, we simulate decomposition of *S. fallax* and *S. fuscum* litter during 50 years, either incubated at 10 cm depth under a degree of saturation of 0.6 $L_{\text{water}}$ $L_{\text{pores}}^{-1}$, or 20 cm below the water table."

62. **Q:** S12: This section does not seem to be referenced in the main manuscript, but if this is relevant to the manuscript, it should be mentioned.

    **A:** We thank the reviewer for this suggestion. We suggest to reference section S12 (new: S11) at the end of section 4.4. Please see the attached manuscript with the suggested changes.

**1.2.7 Technical corrections**

63. **Q:** Line 3: separate "conditions and".

    **A:** We will correct this typo as suggested.

64. **Q:** Throughout the text citation style is inconsistent. I suggest not using parentheses for the year if the citation is already in a parenthesis. Use a comma instead, like you have already in many parts of the text. Correct the citations accordingly in lines: 27, 32, 33, 66, 375, 421, 422, 434, 453.

    **A:** We thank the reviewer for this suggestion and will change the citations as suggested.

65. **Q:** Line 99: correct "use" for "used".

    **A:** We will change the text as suggested.

66. **Q:** Line 117: add "and" before "water table depths".

    **A:** We will change the text as suggested.

67. **Q:** Line 223: change "form" with "from".

    **A:** We will correct "form" to "from" as suggested.

68. **Q:** Line 328: It should be "as a consequence".

    **A:** We will change the text as suggested.

69. **Q:** Line 366: after ":" the next letter should be lowercase, otherwise change ":" for ".".

    **A:** We will change the text as suggested.

70. **Q:** Line 475: I think another word like consider, believe, think, etc. is a better alternative for expect here.

    **A:** We did not find "expect" in l. 475. Does the reviewer mean l. 375?

71. **Q:** Supplementary information S11: change the "," in S. fallax.

    **A:** We will change the text as suggested.

**2 Additional changes**

1. We suggest a complete re-write of large parts of the manuscript to address the reviewer comments. Specific aspects of this re-write are listed in the comments of the reviewers, others are too numerous for a list of them to be useful without knowing the context of these changes. Please see the attached manuscript with the suggested changes.

2. We suggest to include Quillet et al. (2015) as reference for studies estimating $c_2$ from peat cores. We suggest the following changes:

   We suggest to change l. 392 to 393 from

   "Larger and smaller $c_2$ than the standard value have been estimated for several permafrost peatland cores with a modified version of the HPM with monthly time step (Treat et al., 2021, 2022)."

   to

   "Larger and smaller $c_2$ than the standard value have been estimated for several peatland cores with the HPM and a modified version with monthly time step (Quillet et al., 2015; Treat et al., 2021, 2022)."

3. Frolking et al. (2010) also mention that peat accumulation as predicted by the HPM is sensitive to $c_2$ and a site-specific parameter. We therefore add Frolking et al. (2010) as reference at l. 47 and 422.

4. In l. 77 we will correct "decmposition" to "decomposition".

**References**

[revised manuscript text omitted]

---

## Author Response (AR3)

**Reply to Anonymous Referee #3 for "Peat Oxic and Anoxic Controls of Sphagnum Decomposition Rates in the Holocene Peatland Model Decomposition Module Estimated from Litterbag Data"**

Henning Teickner[1,2,*]       Edzer Pebesma[2]       Klaus-Holger Knorr[1]

05 March, 2025

[1] ILÖK, Ecohydrology & Biogeochemistry Group, Institute of Landscape Ecology, University of Münster, 48149, Germany
[2] IfGI, Spatiotemporal Modelling Lab, Institute for Geoinformatics, University of Münster, 48149, Germany

[*] Correspondence: Henning Teickner <henning.teickner@uni-muenster.de>

Comments made by the reviewer start with a bold **Q** while our reply starts with a bold **A**. In section "Additional changes" we list additional changes we would like to incorporate in an updated version of the manuscript.

**1 Reply to comments**

**1.1 General comments:**

1. **Q:** The authors clearly took into consideration the comments from the previous round of evaluation of the manuscript and improved it accordingly. They changed the title which now better reflects the content and relevance of the study. They also modified the abstract which now gives a better overview of the work. Generally, the main text is highly improved compared to the first version, although it still needs some minor improvements listed below. In this new version, the originality and aims of this study are stated more clearly. Additionally, the connections and differences of this study with other studies by the same authors are better described. The methods section has been improved considerably, as well as the supplement material. Considering all this, I suggest the manuscript should be accepted subject to minor revisions.

   **A:** We thank the reviewer for the useful suggestions and hope that we addressed all comments (see the following replies).

**1.2 Specific comments:**

**1.2.1 Introduction:**

2. **Q:** L41: can "the litter that does not decompose fast even under environmental conditions favorable for decomposition" be explained by litter traits that make litter less decomposable (e.g., high C:N or lignin:N ratios, high toughness, high concentration of tannins, etc.)? If so, I think you could replace that part of the sentence with a statement about peat litter chemical and physical quality and how that makes it less decomposable.

   **A:** We changed the sentence to (new: ll. 29 to 32): "Compared to other ecosystems, northern peatlands usually have small decomposition rates because of cold temperatures, high water table levels, acidic pH value, and litter that does not decompose fast because of chemical and physical litter properties (van Breemen, 1995; Rydin et al., 2013)."

3. **Q:** L43: in "during the Holocene and…", I suggest changing and for but. You're making a contrast, so I think it's more suitable.

   **A:** We changed the text as suggested.

4. **Q:** L55: what are the "both sides of the test"?

   **A:** This refers to the rest of the paragraph: the models (one side of the test) have large error sources (structure, parameter values) and the observed data (the other side of the test) have large measurement errors. We agree that this can be written more diretly.
   We suggest to change (old: ll. 42 to 43):

   "Second, there often are large uncertainties on both sides of the test"

   to:

   "Second, there often are large uncertainties both in the model being tested and the data used to test the model"

5. **Q:** L64: I suggest rephrasing this as: "Estimating values and uncertainties of parameters that directly control decomposition rates could be used to test the decomposition module of a peatland model".

   **A:** We thank the reviewer for this suggestion and changed the sentence accordingly (new: l. 49).

6. **Q:** L70: Why is this test restricted to short time ranges?

   **A:** Available *Sphagnum* litterbag data cover only short time periods (usually not more than 5 years) and therefore do not provide information on long-term decomposition, which also restricts the test to these short time periods. To briefly explain this, we suggest to change this sentence from (old: l. 55):

"Admittedly, such a test is restricted to short time ranges and not representative for long-term decomposition rates which may differ from that of fresh litter (e.g., Frolking et al., 2001), but future tests with different scope and applications of the model will benefit from the reduced parameter uncertainties and can consider where the model fails already on short time scales."

to (new: ll. 56 to 59):

"Admittedly, such a test is restricted to the time ranges covered by available litterbag experiments and therefore not representative for long-term decomposition rates which may differ from that of fresh litter (e.g., Frolking et al., 2001), but future tests with different scope and applications of the model will benefit from the reduced parameter uncertainties and can consider where the model fails already on short time scales."

**1.2.2 Methods:**

7. **Q:** L155: I think you should reference Fig. 1 in this section and try to follow the flow of that figure to help the reader follow the workflow.

   **A:** We added cross-references to Fig. 1 in section 2.1, 2.2, and 2.3 (see next sentence). In addition, we moved part of the former section 2.2.2 to a new section 2.2 ("Modeling remaining masses and decomposition rates with the litterbag decomposition model") to better separate the description of the litterbag decomposition model from the HPM decomposition module. The new section 2.3 (former section 2.2) now has the title "Prediction of litterbag decomposition rates with the Holocene Peatland Model decomposition module". We made some smaller adjustments to make the text fit the new section structure.

8. **Q:** L159 and L163: here you mentioned k0, but you have not defined it yet.

   **A:** We thank the reviewer for pointing this out. We suggest to change the text from (old: ll. 131):

   "… to estimate $k_0$ using the litterbag decomposition model."

   to:

   "… to estimate decomposition rates ($k_0$) using the litterbag decomposition model."

9. **Q:** L162: define WTD on the main text, please. You only defined it on the caption of Fig. 1.

   **A:** We thank the reviewer for pointing this out. We suggest to change the text from (old: ll. 133 to 134):

   "… reported WTD and therefore only these data were used to predict $k_0$ also with the HPM decomposition module."

   to (note: this sentence also includes changes due to comment 11 of referee 2):

"… include water table depths (WTD) and depths below the surface where litterbags were incubated, in addition to remaining masses and taxonomic information, and therefore only these datasets were used to predict $k_0$ also with the HPM decomposition module."

10. **Q:** L201: it should be "consider it as an unknown parameter".

    **A:** We corrected the text as suggested.

11. **Q:** L285: It is not clear to me in this section if this refers to the estimation of mass lost before the collection of the samples and before the start of the experiment. Can you clarify this? Also, can you explain why samples from the surface are not expected to have lost any mass previously? And how did you estimate mass lost before the experiment for the samples in Prevost et al. (1997)?

    **A:** We thank the reviewer for pointing out that this section is confusing. To answer the questions:

    - "It is not clear to me in this section if this refers to the estimation of mass lost before the collection of the samples and before the start of the experiment.": This section refers to mass lost before the collection of the samples and before the start of the experiment.

    - "Also, can you explain why samples from the surface are not expected to have lost any mass previously?" It is generally difficult to separate living from dead *Sphagnum* material because a *Sphagnum* plant grows continuously at the top and dies off continuously at lower parts. This has difficulty is mentioned in several studies [e.g.; Breeuwer.2008], but the litterbag studies included here that use *Sphagnum* material from the surface tried to collect material that has not experienced decomposition yet. Thus, even though separating living from dead parts is poorly standardized, we here assume that experiments that report to have used undecomposed material indeed used undecomposed material.

    - "And how did you estimate mass lost before the experiment for the samples in Prevost et al. (1997)?" This is a misunderstanding that may have been caused by the section title. Since the HPM deocmposition module assumes that decomposition rates decrease as mass is lost, one has to know how much mass is lost prior to the start of a litterbag experiment to correctly predict its decomposition rate with the HPM decomposition module. Prevost et al. (1997) clearly used material that has experienced decomposition before the start of the litterbag experiment, but the amount of mass lost is unknown because of the nature of the sample (material from a peat core). To avoid this problem, we did not estimate mass lost before the experiment, but we defined a dummy *Sphagnum* species for the samples from the same depth layer and the HPM decomposition module is used to estimate an initial decomposition rate for these dummy species. This way, we account for the prior decomposition loss, making the data compatible with the HPM decomposition module, but do not need to estimate the fraction of initial mass already lost.

To avoid confusion, we suggest the following changes:

- We suggest to change the section title from "Fraction of mass lost during previous decomposition" to "Accounting for mass loss before the start of the litterbag experiments".
- We suggest to re-write the section to provide more details that answer the questions of the reviewer:

"The HPM decomposition module assumes that decomposition rates decrease the more of the initial mass has already been decomposed (Frolking et al., 2001; Frolking et al., 2010). Thus, if litter lost some mass due to decomposition before the start of the litterbag experiment, one has to know the magnitude of this mass loss to correctly predict decomposition rates with the HPM decomposition module.
Because of the continuous growth of *Sphagnum* at the top and die-off below, it is difficult to separate living material, assumed to not have lost mass, from dead material, which may have already lost some mass. Based on a visual assessment, the studies that used *Sphagnum* material from the surface, assume that the material did not loose mass prior to the litterbag experiments and we follow this assumption ($m(t = 0) = 1$ in equation (2)).
Samples from Prevost et al. (1997) are *Sphagnum* peat collected from two different depth levels from the same location and these samples probably had already experienced some decomposition, however it is difficult to estimate how much. Apart from knowing the exact mass loss prior to the litterbag experiment, an alternative approach to allow predicting decomposition rates with the HPM with previous mass loss is to define a dummy species for a sample, such that the maximum possible decomposition rate for the sample ($k_{0,i}$) is estimated separately. We therefore estimated $k_{0,i}$ separately for each peat layer in Prevost et al. (1997), implicitly assuming that these are two different PFT with different maximum possible decomposition rates."

12. **Q:** L290: it should be 'rates'.

    **A:** We corrected the text as suggested.

13. **Q:** L335: Can you please rewrite this sentence without the long dashes, using connectors instead?

    **A:** We replaced each "—" by a ",".

14. **Q:** L380: it should be 'predictions'.

    **A:** We thank the reviewer for reporting this typo. We corrected the text as suggested.

15. **Q:** L438: I suggest rewriting this as: 'With these settings, we predicted five sets of average k0: (1) with HPM-leaching (k0,modified(HPM-leaching)), and the remaining with (2) c1, (3) Wopt, (4) fmin, and (5) c2 set to their standard value (k0,standard(HPM-leaching)).'

    **A:** We thank the reviewer for this suggestion. We changed the text as suggested (new: ll. 378 to 380).

**1.2.3 Results:**

16. **Q:** L458: instead of 'less well' use 'worse'.

    **A:** We changed the text as suggested.

17. **Q:** L621: Where are the results to support this paragraph?

    **A:** Unfortunately, it is not clear what the reviewer means. The paragraph is:

    "In model HPM-leaching, we included a logistic regression model that estimates the relation between $l_0$ and the degree of saturation. The parameter estimates suggest that both positive and negative relations of $l_0$ to the degree of saturation are compatible with available litterbag data (95% confidence intervals for the slope (logit scale): (-0.28,0.15)). Thus, available litterbag data do not allow to conclude whether $l_0$ are positively related to the degree of saturation or not."

    The result that we report is that the 95% confidence interval clearly contains 0 and a range of negative and positive estimates for the relation between $\mu_l$ and the degree of saturation and we do not think the paragraph makes any statement that would not be supported by this result.

**1.2.4 Discussion:**

18. **Q:** L788: I suggest rewriting this as 'Previous global and local sensitivity analyses, where HPM parameter values and environmental conditions were varied in broad ranges, ...'.

    **A:** We thank the reviewer for this suggestion. We changed the text accordingly (new: ll. 596 to 597).

19. **Q:** L814: Is Sphagna correct or is this a typing error? I do not think you can make scientific genera plural.

    **A:** We agree that it is more appropriate to write "*Sphagnum* species" instead. We changed the text accordingly.

20. **Q:** L839: I suggest rewriting this as 'We suggest the following steps to improve accuracy when estimating peatland decomposition module parameters:'.

    **A:** We thank the reviewer for this suggestion. We changed the text accordingly (new: l. 640).

21. **Q:** L879: I think this section needs at least a sentence that clearly states your suggestions on how to decrease errors in k0 and l0.

    **A:** This is the topic of Teickner et al. (2025). We agree that the sentence can be improved. We suggest to change (old: ll. 588 to 589):

    "Future litterbag experiments that aim to improve peatland models should reduce errors of $k_0$ and $l_0$ estimates (e.g., Teickner et al., 2025)"

to:

"Future litterbag experiments that aim to improve peatland models should reduce errors of $k_0$ and $l_0$ estimates. A first step would for example be to modify litterbag experiments as described in Teickner et al. (2025)."

22. **Q:** L913: Can you explain what you mean by 'Studies that systematically change litter chemistry within species'?

    **A:** We mean studies that change in litter chemistry by changing growth conditions, for example different N supply, $CO_2$ concentration, or moisture conditions (e.g., Siegenthaler et al., 2010; Straková et al., 2010) or that observe systematic differences in litter chemistry under different environmental conditions [e.g.; Bengtsson et al. (2018)]. We suggest to include "(e.g., Siegenthaler et al., 2010; Straková et al., 2010)" in this sentence (new: l. 680).

**1.2.5 Conclusions**

23. **Q:** L930: I suggest rewriting this as 'Based on the litterbag data, our estimates of the degree of saturation where decomposition is optimal (Wopt) and the anoxia scale length (c2) are significantly larger than the standard parameter values.'

    **A:** We thank the reviewer for the suggestion and changed the text accordingly (new: ll. 696 to 697).

**1.2.6 Supplement**

24. **Q:** S3: The x axis label of Figures S1 and S2 should be 'Remaining mass (%)'.

    **A:** We changed the axis labels as suggested.

25. **Q:** S5 and S8: The captions in Figured S10, S11, and S16 are backwards, (a) and (b) should be interchanged.

    **A:** We thank the reviewer for pointing out this mistake. We corrected the figure captions as suggested.

**2 Additional changes**

1. In Fig. S2 and S4, we changed the title of the first panel from "HPM-standard" to "LDM-standard".

2. In l. 513 (old), we changed "… and in addition similar across these (independent) studies …" to (new: l. 522) "… and in addition similar across independent studies …".

3. In l. 530 (old; new: l. 539), we added the panel label "(b)" to the figure cross-reference.

4. In l. 534 (old; new: l. 543), we corrected the figure cross-reference from "supporting Fig. S15" to "Fig. 7 (a)".

**References**

Bengtsson, F., Rydin, H., and Hájek, T.: Biochemical determinants of litter quality in 15 species of *Sphagnum*, Plant and Soil, 425, 161–176, https://doi.org/10.1007/s11104-018-3579-8, 2018.

Frolking, S., Roulet, N. T., Moore, T. R., Richard, P. J. H., Lavoie, M., and Muller, S. D.: Modeling northern peatland decomposition and peat accumulation, Ecosystems, 4, 479–498, https://doi.org/10.1007/s10021-001-0105-1, 2001.

Frolking, S., Roulet, N. T., Tuittila, E., Bubier, J. L., Quillet, A., Talbot, J., and Richard, P. J. H.: A new model of Holocene peatland net primary production, decomposition, water balance, and peat accumulation, Earth System Dynamics, 1, 1–21, https://doi.org/10.5194/esd-1-1-2010, 2010.

Prevost, M., Belleau, P., and Plamondon, A. P.: Substrate conditions in a treed peatland: Responses to drainage, Écoscience, 4, 543–554, https://doi.org/10.1080/11956860.1997.11682434, 1997.

Rydin, H., Jeglum, J. K., and Bennett, K. D.: The biology of peatlands, 2nd ed., Oxford University Press, Oxford, 2013.

Siegenthaler, A., Buttler, A., Bragazza, L., Heijden, E. van der, Grosvernier, P., Gobat, J.-M., and Mitchell, E. A. D.: Litter- and ecosystem-driven decomposition under elevated $CO_2$ and enhanced N deposition in a *Sphagnum* peatland, Soil Biology and Biochemistry, 42, 968–977, https://doi.org/10.1016/j.soilbio.2010.02.016, 2010.

Straková, P., Anttila, J., Spetz, P., Kitunen, V., Tapanila, T., and Laiho, R.: Litter quality and its response to water level drawdown in boreal peatlands at plant species and community level, Plant and Soil, 335, 501–520, https://doi.org/10.1007/s11104-010-0447-6, 2010.

Teickner, H., Pebesma, E., and Knorr, K.-H.: A synthesis of *Sphagnum* litterbag experiments: Initial leaching losses bias decomposition rate estimates, Biogeosciences, 22, 417–433, https://doi.org/10.5194/bg-22-417-2025, 2025.

van Breemen, N.: How *Sphagnum* bogs down other plants, 6, 1995.

**Reply to Anonymous Referee #2 for "Peat Oxic and Anoxic Controls of Sphagnum Decomposition Rates in the Holocene Peatland Model Decomposition Module Estimated from Litterbag Data"**

Henning Teickner[1,2,*]    Edzer Pebesma[2]    Klaus-Holger Knorr[1]

05 March, 2025

[1] ILÖK, Ecohydrology & Biogeochemistry Group, Institute of Landscape Ecology, University of Münster, 48149, Germany
[2] IfGI, Spatiotemporal Modelling Lab, Institute for Geoinformatics, University of Münster, 48149, Germany

[*] Correspondence: Henning Teickner <henning.teickner@uni-muenster.de>

Comments made by the reviewer start with a bold **Q** while our reply starts with a bold **A**. In section "Additional changes" we list additional changes we would like to incorporate in an updated version of the manuscript.

**1   Reply to comments**

1. **Q:** Dear Authors,

   Thanks for your response and the modified manuscript. I think I now understand better what you did and what the litterbag model is, and therefore, also the text in general is clearer. However, I find especially the Method part of the text should still be clarified regarding different models. It was still difficult to follow which model you talk about and what are the differences between different models. Perhaps one of the reasons for the confusion was that you mention separately the HPM decomposition model and the litterbag decomposition model LDM, and in addition there are four different versions of models that combine those two - but their names only include 'HPM'. I would understand this if you only used their HPM decomposition module, with the new parameters, to predict decomposition rates, but that's not the case. For example, Fig.2a has titles HPM-peat, HPM-all and HPM-leaching, but the caption says the remaining masses are predicted by the litterbag decomposition model, not

HPM. I found this confusing. So, I hope you find a way to differentiate still more clearly the different models.

**A:** We thank the reviewer for this suggestion. We changed the names of the different model versions as suggested by referee 3 (from the first round of reviews). In the first version of the manuscript, we had model names that contained more information, but we agreed with referee 3 that it is better to use simple model names that are easier to parse and remember instead of trying to encode all the information about the models in these abbreviations. We therefore do not think that it makes sense to make the model abbreviations more complex again.
We hope that Tab. 3 provides a concise summary of what the different models contain and how they differ and we think that the Methods section now describes the different models in a comprehensive way. We think that this is a good compromise between the need for a more complex modeling strategy to test the HPM decomposition module against heterogeneous data and making our analysis as easy to follow as possible. For example, regarding the example of Fig. 2 (a), the models HPM-peat, HPM-all and HPM-leaching are described in Tab. 3 and there, it is clearly described that they contain the litterbag decomposition model as prior and also what the litterbag decomposition model is.

2. **Q:** I don't go through point-by-point my previous comments since you re-wrote many parts of the manuscript. But here are more specific comments for this version: Around line 54: Please describe briefly what is a litterbag experiment and what kind of data it produces (litter masses etc..), it's not necessarily a familiar concept for all readers.

   **A:** We thank the reviewer for this suggestion. We suggest to change ll. 53 to 55 (old) from:

   "The predictions can be compared to decomposition rates estimated from litterbag data and therefore future litterbag studies can directly test whether discrepancies identified in such a test are replicable."

   to (new: ll. 52 to 56):

   "Decomposition rates can also be estimated from litterbag experiments, where a known initial mass of litter is filled into mesh bags, incubated in peat, excavated after some time, and re-weighed to estimate the mass loss due to decomposition. Therefore, predicted decomposition rates can be compared to decomposition rates estimated from litterbag experiments and replicability of any identified discrepancies can be directly tested in future litterbag studies."

3. **Q:** Line 60-62: I find it slightly inaccurate to write that a parameter can result in a doubling of accumulated C. Do you mean that a minor change in the parameter value can result in the doubling of accumulated C?

   **A:** We agree that it is more correct to state that a change in a parameter value causes changes in some target quantity. We suggest to change the text from (old: ll. 60 to 63):

"Previous sensitivity analyses of the HPM and applications to peat cores suggest that the anoxia scale length ($c_2$), the parameter controlling how anaerobic decomposition rates are limited by electron acceptor depletion and accumulation of decomposition products, can result in a doubling of accumulated C, depending on climate conditions (Frolking et al., 2010; Quillet et al., 2013; Kurnianto et al., 2015)."

to (new: ll. 61 to 64):

"Previous sensitivity analyses of the HPM and applications to peat cores suggest that relative small changes to the anoxia scale length ($c_2$), the parameter controlling how anaerobic decomposition rates are limited by electron acceptor depletion and accumulation of decomposition products, can result in a doubling of accumulated C, depending on climate conditions (Frolking et al., 2010; Quillet et al., 2013; Kurnianto et al., 2015)."

4. **Q:** Line 76: Please define here briefly what a litterbag decomposition model is. Is it e.g. any model that takes as input data from a litterbag experiment and outputs the mass loss of the litter as function of time? Can you add a reference/references to model(s) other than yours?

   **A:** We suggest to change the sentence from (old: ll. 75 to 76):

   "Since decomposition rates have been estimated with different litterbag decomposition models in previous studies, their values are not directly comparable."

   to (new: ll. 76 to 77):

   "Since decomposition rates in litterbag experiments have been estimated with different litterbag decomposition models in previous studies, their values are not directly comparable."

   We do not cite our litterbag decomposition model in the referenced sentence and we do not think that specific references are required here.

5. **Q:** Point 7. of my first review (and here line 83 onwards): The problem with the sentence maybe is that the text above seems to talk about peatland models in general, but here you refer specifically to "model compatible with the HPM". So I get confused about why in general, for all models, compatibility with the HPM is relevant. In addition, I think it would be good to shortly write here what it means that a litterbag decomposition model is compatible with HPM, like you explained in your response letter.

   I mean, would e.g. something like this be correct, line 83-: "Even though tests of only a part of the HPM are less uncertain than tests of the whole model, there still is a risk that they are dominated by uncertainties. Remaining masses in litterbag experiments are often very variable, even under controlled environmental conditions (e.g., Bengtsson et al., 2018). In addition, for many litterbag experiments, a range of decomposition rates may produce similar predictions for remaining masses (e.g., Yu et al., 2001), also when the litterbag decomposition model is compatible with HPM, i.e., uses the same equation as HPM to describe mass losses (Teickner et al., 2025a)."

**A:** We thank the reviewer for the clarification. We suggest to change the sentence from (old: ll. 84 to 87):

"Remaining masses in litterbag experiments are often very variable, even under controlled environmental conditions (e.g., Bengtsson et al., 2018), and for many litterbag experiments, a range of decomposition rates may produce similar predictions for remaining masses (e.g., Yu et al., 2001), also if a litterbag decomposition model compatible with the HPM is used (Teickner et al., 2025)."

to something similar as suggested by the reviewer (new: ll. 85 to 89):

"Remaining masses in litterbag experiments are often very variable, even under controlled environmental conditions (e.g., Bengtsson et al., 2018), and for many litterbag experiments, a range of decomposition rates may produce similar predictions for remaining masses (e.g., Yu et al., 2001), also if a litterbag decomposition model compatible with the HPM, i.e. that uses equation (7) in Frolking et al. (2010) to describe decomposition mass losses, is used (Teickner et al., 2025)."

6. **Q:** Line 91: Is it only to estimate uncertainties or also parameter values?

   **A:** We think that the reader knows that Bayesian methods can be used to estimate parameters. The referenced sentence and the previous sentence explicitly refer to parameter uncertainty (parameter errors) and therefore, we only mention parameter uncertainty here.

7. **Q:** Line 96: Larger than what?

   **A:** We suggest to change "larger" to "large". We did not want to make a comparison, but give a statement that the discrepancies exceed a threshold considered as ecologically significant.

8. **Q:** Line 112: Please specify what "these studies" are.

   **A:** We mean "the litterbag studies" and suggest to change the text accordingly.

9. **Q:** Lines 119-124: I'd rather put this into conclusions.

   **A:** The referenced paragraph is: "We only test the decomposition module of the HPM, but the decomposition modules of many other peatland models are also parameterized based on litterbag experiments and our modeling approach is flexible enough to be combined with other decomposition modules. Therefore, our test could serve as a blueprint for similar tests of other peatland model decomposition modules. Similarly, the parameter discrepancies identified here suggest future litterbag experiments that would provide novel insights into oxic and anoxic controls of *Sphagnum* decomposition rates and our study therefore suggests a strategy to improve decomposition modules in general."
   We think that it is useful to use the last paragraph in the Introduction to circle back to the broad topic or problem a paper tries to address in order to describe what the study contributed to this broad topic or problem.

10. **Q:** Figure 1 caption: Please move the (μ_k) after "average decomposition rates", and explain why there are the arrows from the bottom (from μ_m(t)) back to the top.

    **A:** We changed the position of "$(\mu_k)$" as suggested. We changed "The equation at the bottom uses these to estimate remaining masses in the litterbag experiments." to "The equation at the bottom uses these to estimate remaining masses as reported in the litterbag experiments." to clarify that the arrows indicate that estimates are conditional on observations.

11. **Q:** Line 133: Is it so that "these data" means Sphagnum PFT, WTD and depth of litter sample below water table from these studies? No litter masses? I suggest this is mentioned clearly here, because it helps to understand the difference between HPM and litterbag decomposition model.

    **A:** The referenced sentence is: "Data from Johnson and Damman (1991), Szumigalski and Bayley (1996), Prevost et al. (1997), Straková et al. (2010), Golovatskaya and Nikonova (2017), and Mäkilä et al. (2018) reported water table depths (WTD) and therefore only these datasets were used to predict $k_0$ also with the HPM decomposition module."
    As illustrated in Fig. 1, we used only data from litterbag studies where at least remaining masses and *Sphagnum* PFT were reported. These data were used to estimate the litterbag decomposition model. If a study in addition reported WTD and the depth of the litter sample below the peat surface, it was used to estimate $\mu_k$ with the HPM decomposition module (and hence parameters of the HPM decomposition module).
    To avoid confusions, we suggest to change the sentence to (new: ll. 133 to 137):

    "Data from Johnson and Damman (1991), Szumigalski and Bayley (1996), Prevost et al. (1997), Straková et al. (2010), Golovatskaya and Nikonova (2017), and Mäkilä et al. (2018) include water table depths (WTD) and depths below the surface where litterbags were incubated, in addition to remaining masses and taxonomic information, and therefore only these datasets were used to predict $k_0$ also with the HPM decomposition module …"

12. **Q:** Title of 2.2: It seems you also describe the litterbag decomposition model under this title, so perhaps remove "with the Holocene Peatland Model" or add "and the litterbag decomposition model"?

    **A:** We thank the reviewer for this suggestion. We moved part of the former section 2.2.2 to a new section 2.2 ("Modeling remaining masses and decomposition rates with the litterbag decomposition model") to better separate the description of the litterbag decomposition model from the HPM decomposition module. The new section 2.3 (former section 2.2) now has the title "Prediction of litterbag decomposition rates with the Holocene Peatland Model decomposition module". We made some smaller adjustments to make the text fit the new section structure.

13. **Q:** Lines 153-179: If this is now description of your litterbag decomposition model, please mention it clearly here. It seems that from line 180 onwards you describe the HPM decomposition module.

**A:** We thank the reviewer for this suggestion. We hope that our reply to the previous comment addresses this issue.

14. **Q:** Line 193: Please write what model is "our model" in this case, whether it is the litterbag decomposition model or HPM decomposition module.

    **A:** It is neither the litterbag decomposition model nor HPM decomposition module, but the combination of both, as illustrated in Fig. 1.
    To clarify this, we suggest to change (old: l. 193):

    "In our model, $k_0$ estimated from the litterbag data for each litterbag experiment …"

    to (new: l. 197):

    "In our model that combines the HPM decomposition module and the litterbag decomposition model, $k_0$ estimated from the litterbag data for each litterbag experiment …"

15. **Q:** Line 204: I don't understand what you mean. What is the difference between estimating HPM decomposition module parameters and adjusting decomposition rates to the HPM decomposition module?

    **A:** We tried to express the difference between how parameters are constrained by data (estimated) and by a prior (adjusted). Since we use the HPM decomposition module as hyper-prior (a prior that is estimated from data), both the HPM decomposition module parameters are constrained by the remaining masses of the litterbag experiments and the decomposition rates estimated with the litterbag decomposition module are constrained by the HPM decomposition module prior. We hope to clarify the sentence by changing it from:

    "Moreover, combining the litterbag decomposition model and the HPM decomposition module into one Bayesian model does not only estimate HPM decomposition module parameters from the litterbag data, but it also adjusts the decomposition rates estimated from litterbag data to the HPM decomposition module because the HPM decomposition module serves as prior in the combined model which therefore estimates what parameter values are compatible with the data and the combined model."

    to (new: ll. 206 to 210) (changing the word "adjusts" to "constrains"):

    "Moreover, combining the litterbag decomposition model and the HPM decomposition module into one Bayesian model does not only estimate HPM decomposition module parameters from the litterbag data, but it also constrains the decomposition rates estimated from litterbag data by the HPM decomposition module because the HPM decomposition module serves as prior in the combined model which therefore estimates what parameter values are compatible with the data and the combined model."

    We made similar changes in l. 353 (new: l. 363).

16. **Q:** Line 262: Can you say - for clarity - shortly that HPM-standard is the HPM with standard values, instead of that it corresponds to it?

    **A:** We thank the reviewer for this suggestion and changed the text accordingly.

17. **Q:** Line 264-265: I don't quite understand the logic of this sentence. Even though the parameters of HPM-standard were used as priors for the litterbag decomposition model, wouldn't it itself remain independent of litterbag data?

    **A:** For all model versions except HPM-standard, the HPM decomposition module is a prior for the litterbag decomposition rates which has parameters that are estimated from litterbag data. Therefore, the HPM decomposition module depends on the litterbag data when it is used as prior.

18. **Q:** Lines 282-285: Please re-formulate this sentence, it is too long.

    **A:** As suggested by referee 3, we replaced each "—" by a ",", and hope that the sentence is now better to understand.

19. **Q:** Line 285: Please define briefly, what tea bags.

    **A:** We do not think that "tea bag" needs to be defined. For details on the experiments, we refer to Lind et al. (2022).

20. **Q:** Line 329: I don't understand why HPM-standard can't predict remaining masses. Why do you need the litterbag decomposition model to get the prediction?

    **A:** The referenced sentence is: "To analyze how well the models fit remaining masses observed in the litterbag experiments, we plotted reported remaining masses versus remaining masses estimated by the litterbag decomposition model in HPM-peat, HPM-all, and HPM-leaching. HPM-standard is not linked to the litterbag decomposition model and therefore does not predict remaining masses."
    HPM-standard is the HPM decomposition module (without litterbag decomposition model). The HPM decomposition module as defined in our study does not predict remaining masses (see the new section 2.2.2). As described in section 2.2.1, the HPM (note: this is more than the HPM *decomposition module*) has an equation to predict remaining masses, but as also described there, this equation does not account for initial leaching losses which need to be considered in order to get unbiased decomposition rate estimates, and it does not account for remaining variations in remaining masses observed in litterbag experiments. Therefore, HPM-standard cannot predict remaining masses because it needs to be linked to a litterbag decomposition model, which is the purpose of HPM-peat (see Tab. 3).

21. **Q:** Line 331: Plot similar to what?

    **A:** The referenced sentence is: "To analyze how well all HPM decomposition module versions fit $k_0$ estimated by the respective litterbag decomposition model, we created a similar plot for $k_0$."

    "similar plot" refers to the plot mentioned directly above in:

    "To analyze how well the models fit remaining masses observed in the litterbag experiments, we plotted reported remaining masses versus remaining masses estimated by the litterbag decomposition model in HPM-peat, HPM-all, and HPM-leaching. HPM-standard is not linked to the litterbag decomposition model and therefore does not predict remaining masses."

22. **Q:** Lines 378-381: I think here is a contradiction. You seem to refer to all litterbag decomposition models, also LDM-standard, because according to Table 3, it was the only one that did not use HPM parameters as priors. However, LDM-standard is not in Fig. 2a.

    **A:** It is true that we refer to all litterbag decomposition models, also LDM-standard, and it is also true that predictions of LDM-standard are not shown in Fig. 2 (a). However, results for LDM-standard are shown in Fig. S2 which is also cross-referenced in the same sentence. We could have included a plot for LDM-standard in Fig. 2 (a), but we did not do so to avoid confusions with Fig. 2 (b), where estimates of LDM-standard are the x-axis value of the first panel. We therefore decided to refer to the supporting information here. If the editor thinks that it would be better to also include a panel for LDM-standard in Fig. 2 (a), we will do so.

23. **Q:** Line 382: You write that "when the HPM decomposition module is not used as prior". Again I'm slightly confused: I would have thought it's not relevant whether it was used as prior or not, but the relevant factor is that it used the default parameters. But perhaps I misunderstood this.

    **A:** We hope that our reply to comment 17 above addresses this issue. As stressed in sections 3.1 and 3.2, the worse fit of HPM-standard mentioned in the sentence referenced by the reviewer is not necessarily a cause of using HPM decomposition module standard parameters. Indeed, HPM-peat, better fits $k_0$ estimated with the litterbag decomposition module included in HPM-peat, but still sets HPM decomposition module parameters to their standard values and does not estimate them from data. Therefore, what is relevant here, is whether the HPM decomposition module is used to constrain estimates from the litterbag decomposition model (= is used as prior for the litterbag decomposition module) or not.

24. **Q:** Section 3.3: You don't explain results related to c_1.

    **A:** It is true that we only mention that estimates for $c_1$ in either model version are compatible with the standard parameter values (section 3.3) and that this parameter does not cause qualitative differences in how $k_0$ change in dependency of the WTD (section 3.5). For this reason, we decided not to discuss these results in more detail.

25. **Q:** Lines 483-487: I was wondering whether observational data about the initial leaching losses could be compared with your results or whether they could be used as a constraint for the l_0 results. Did you consider that?

    **A:** We thank the reviewer for this suggestion. If data on initial leaching losses were available, we would have used them to constrain $l_0$ estimates and this would have solved a lot of problems (in particular this would have made $k_0$ estimates by the litterbag decomposition model more accurate and therefore HPM decomposition module parmeters would have been estimated more accurately, too). However, it would have been necessary to have precise $l_0$ estimates for the litterbag experiments since the magnitude of $l_0$ probably depends a lot on properties of the litter samples used in the experiment and how they were preprocessed (Teickner et al., 2025). This is mentioned in point 4 of section 4.4 of our manuscript here.

26. **Q:** Line 528 and 532: ModGberg

    **A:** This abbreviation for the modified Granberg model was introduced in l. 234 (old; new: l. 238).

27. **Q:** Line 660 onwards: I think that yes, it would be good to have litterbags in constant anoxia, but deep depths as such are perhaps not so relevant for fresh Sphagnum litter since peat deep in the soil is probably quite old.

    **A:** As shown in Fig. 7 and described in Tab. 1, the HPM decomposition module implies that there is a gradual change in anaerobic decompostion rates with distance below the average annual WTD due to a gradual change in availability of terminal electron acceptors and concentration of decomposition end products and it implies that this change is independent of the litter quality.
    As stated in ll. 660 to 668, our suggestion does not aim to provide decomposition rate estimates that are representative for natural conditions (as the reviewer seems to suggest), but it describes how one can estimate the hypothesized gradient in anaerobic decompostion rates with distance below the average annual WTD as accurately as possible; a necessary condition for this is that the same litter type is used across the entire depth gradient, since different litter types may have different decomposition rates under the same environmental conditions.
    Of course, one could use already decomposed peat (*Sphagnum* litter) for this, but because this material has smaller decomposition rates even under aerobic conditions (due to a decreased litter quality due to previous decomposition), it would result in smaller mass losses and therefore differences in decomposition rates across the gradient cannot be detected as accurately as would be the case with litter with larger decomposition rate. Moreover, *Sphagnum* litter is much easier to standardize than already decomposed peat and therefore experiments with already decomposed peat are less replicable. Therefore, it makes more sense to use fresh *Sphagnum* material or any other litter with not too small decomposition rate (assuming the HPM decomposition module assumption that the gradient is independent of the litter type is true). Only then, when the gradient has been estimated more accurately, the gradient of anaerobic decomposition rates for older peat can be estimated by using $k_{0,i}$ estimates for older peat samples (see also equation (5)).

**2 Additional changes**

1. In Fig. S2 and S4, we changed the title of the first panel from "HPM-standard" to "LDM-standard".

2. In l. 513 (old), we changed "… and in addition similar across these (independent) studies …" to (new: l. 522) "… and in addition similar across independent studies …".

3. In l. 530 (old; new: l. 539), we added the panel label "(b)" to the figure cross-reference.

4. In l. 534 (old; new: l. 543), we corrected the figure cross-reference from "supporting Fig. S15" to "Fig. 7 (a)".

**References**

Bengtsson, F., Rydin, H., and Hájek, T.: Biochemical determinants of litter quality in 15 species of *Sphagnum*, Plant and Soil, 425, 161–176, https://doi.org/10.1007/s11104-018-3579-8, 2018.

Frolking, S., Roulet, N. T., Tuittila, E., Bubier, J. L., Quillet, A., Talbot, J., and Richard, P. J. H.: A new model of Holocene peatland net primary production, decomposition, water balance, and peat accumulation, Earth System Dynamics, 1, 1–21, https://doi.org/10.5194/esd-1-1-2010, 2010.

Golovatskaya, E. A. and Nikonova, L. G.: The influence of the bog water level on the transformation of sphagnum mosses in peat soils of oligotrophic bogs, Eurasian Soil Science, 50, 580–588, https://doi.org/10.1134/S1064229317030036, 2017.

Johnson, L. C. and Damman, A. W. H.: Species-controlled *Sphagnum* decay on a south Swedish raised bog, Oikos, 61, 234, https://doi.org/10.2307/3545341, 1991.

Kurnianto, S., Warren, M., Talbot, J., Kauffman, B., Murdiyarso, D., and Frolking, S.: Carbon accumulation of tropical peatlands over millennia: A modeling approach, Global Change Biology, 21, 431–444, https://doi.org/10.1111/gcb.12672, 2015.

Lind, L., Harbicht, A., Bergman, E., Edwartz, J., and Eckstein, R. L.: Effects of initial leaching for estimates of mass loss and microbial decomposition—Call for an increased nuance, Ecology and Evolution, 12, https://doi.org/10.1002/ece3.9118, 2022.

Mäkilä, M., Säävuori, H., Grundström, A., and Suomi, T.: *Sphagnum* decay patterns and bog microtopography in south-eastern Finland, Mires and Peat, 1–12, https://doi.org/10.19189/MaP.2017.OMB.283, 2018.

Prevost, M., Belleau, P., and Plamondon, A. P.: Substrate conditions in a treed peatland: Responses to drainage, Écoscience, 4, 543–554, https://doi.org/10.1080/11956860.1997.11682434, 1997.

Quillet, A., Garneau, M., and Frolking, S.: Sobol' sensitivity analysis of the Holocene Peat Model: What drives carbon accumulation in peatlands?, Journal of Geophysical Research: Biogeosciences, 118, 203–214, https://doi.org/10.1029/2012JG002092, 2013.

Straková, P., Anttila, J., Spetz, P., Kitunen, V., Tapanila, T., and Laiho, R.: Litter quality and its response to water level drawdown in boreal peatlands at plant species and community level, Plant and Soil, 335, 501–520, https://doi.org/10.1007/s11104-010-0447-6, 2010.

Szumigalski, A. R. and Bayley, S. E.: Decomposition along a bog to rich fen gradient in central Alberta, Canada, Canadian Journal of Botany, 74, 573–581, https://doi.org/10.1139/b96-073, 1996.

Teickner, H., Pebesma, E., and Knorr, K.-H.: A synthesis of *Sphagnum* litterbag experiments: Initial leaching losses bias decomposition rate estimates, Biogeosciences, 22, 417–433, https://doi.org/10.5194/bg-22-417-2025, 2025.

Yu, Z., Turetsky, M. R., Campbell, I. D., and Vitt, D. H.: Modelling long-term peatland dynamics. II. Processes and rates as inferred from litter and peat-core data, Ecological Modelling, 145, 159–173, https://doi.org/10.1016/S0304-3800(01)00387-8, 2001.